# On the Accuracy of Newton Step and Influence Function Data Attributions

**Ittai Rubinstein** [1]  **Samuel Hopkins** [1]

## Abstract

Data attribution estimates how a trained model would change if a subset of training points were removed, and is a central primitive for tasks such as interpretability, data valuation, and machine unlearning. Despite its widespread use, our theoretical understanding of key data attribution methods—Influence Functions (IF) and a single Newton Step (NS)—remains limited: existing guarantees heavily rely on *global* strong convexity and yield bounds with pessimistic dependence on the parameter dimension $d$ and the number of removed samples $k$. We give a new analysis of IF and NS for convex ERM that replaces global assumptions with *local* conditions: it suffices that the loss is strongly convex and sufficiently smooth only in a neighborhood of the first Newton step. As a concrete validation, we analyze logistic regression with Gaussian features and show that our bounds capture the correct scaling up to polylogarithmic factors, yielding matching upper and lower bounds and explaining observed regimes in which NS is markedly more accurate than IF, thereby resolving open questions raised by (Koh et al., 2019).

## 1. Introduction

Let $L = \ell_1 + \cdots + \ell_n : \Omega_\theta \to \mathbb{R}$ be an Empirical Risk Minimization (ERM) problem, where the contributions to the loss may represent the training samples of some supervised learning problem $\ell_i = \ell(f(\boldsymbol{\theta}, x_i), y_i)$. Define

$$\hat{\boldsymbol{\theta}} \stackrel{\text{def}}{=} \operatorname*{argmin}_{\boldsymbol{\theta} \in \Omega_\theta} \left\{ \sum_{i=1}^{n} \ell_i(\boldsymbol{\theta}) \right\}, \quad \hat{\boldsymbol{\theta}}_T \stackrel{\text{def}}{=} \operatorname*{argmin}_{\boldsymbol{\theta} \in \Omega_\theta} \left\{ \sum_{i \notin T} \ell_i(\boldsymbol{\theta}) \right\}$$

*Data attribution* seeks to explain the dependence of $\hat{\boldsymbol{\theta}}$ on

the training data by predicting how $\hat{\boldsymbol{\theta}}$ changes when removing a subset $T \subset [n]$ of the samples $\ell_i$ (Ilyas et al., 2022; Park et al., 2023)[1]. We can answer such a query by fully retraining the model $\hat{\boldsymbol{\theta}}$ with some samples removed, but this is often computationally intensive and lacks a closed form solution, which can be crucial for downstream tasks (Madry et al., 2024). Therefore, data attributions often utilize approximations, the most widespread of which are influence functions (IF), which use a first-order approximation to the effect of downweighting a set of samples, and single Newton steps (NS), which approximate the removal effect by taking a single step of Newton's algorithm from the point $\hat{\boldsymbol{\theta}}$ (see Section 1.1).

Data attribution is used in a wide variety of applications from cross validation (Rahnama Rad & Maleki, 2020; Wilson et al., 2020) and data valuation (Jia et al., 2019; Ghorbani & Zou, 2019) to privacy and unlearning (Sekhari et al., 2021; Neel et al., 2021; Suriyakumar & Wilson, 2022; Brown et al., 2024). Despite this, at present, our mathematical understanding of the performance of IF and NS is spotty at best, especially when $\hat{\boldsymbol{\theta}}$ is high dimensional. Thus in this paper we take a rigorous mathematical approach to the questions:

> *When do IF and NS give good approximations of the data-removal effect? How does the approximation error scale with the total number of samples $n$, the problem dimension $d$ and number of samples removed $k = |T|$?*

We analyze the accuracy of IF and NS in the analytically tractable setting of convex empirical risk minimization. Beyond the direct application for convex learning problems (such as last-layer fine-tuning through logistic regression), understanding the convex setting is crucial since leading data attributions for non-convex settings often rely on heuristic reductions to data attribution for convex problems (e.g., TRAK uses a neural tangent kernel assumption to approx-

---

[1]EECS and CSAIL, Massachusetts Institute of Technology, Cambridge, MA, USA. Correspondence to: Ittai Rubinstein <ittair@csail.mit.edu>, Samuel B. Hopkins <samhop@mit.edu>.

*Proceedings of the $43^{rd}$ International Conference on Machine Learning*, Seoul, South Korea. PMLR 306, 2026. Copyright 2026 by the author(s).

---

[1]This formulation is also known as predictive data attribution or datamodeling. For a broader overview of predictive, corroborative, and game-theoretic notions of data attribution, see (Madry et al., 2024). Some works further restrict the counterfactual predictor to be linear or additive in the samples, assigning individual contributions to model weights or predictions, but we use the subset-level formulation for maximal generality.

imate the attribution of a deep neural network with the attribution of a related logistic regression (Park et al., 2023)). This is one motivation for our focus on uncovering the scaling laws for the convex setting – we discuss several further motivations later.

Existing mathematical analyses of IF and NS data attributions (Rahnama Rad & Maleki, 2020; Giordano et al., 2019; Koh et al., 2019; Wilson et al., 2020) make progress towards a mathematical understanding of IF and NS, but remain unsatisfactory in at least two respects. First, they rely on an unrealistic *global* strong convexity assumption – that both the loss $\sum_{i=1}^{n} \ell_i$ and the leave-$T$-out loss $\sum_{i \notin T} \ell_i$ are strongly convex for all $\theta$. Even for convex learning problems, this level of strong convexity often does not hold in practice.[2] Second, the quantitative bounds on approximation error for IF and NS scale poorly, both in this reliance on global strong convexity, and as high-degree polynomials in the problem dimension $d$ and with the number of samples removed $k$ (Rubinstein & Hopkins, 2025b).

Empirical work, e.g. (Koh et al., 2019; Rubinstein & Hopkins, 2025b) suggests that IF and NS approximations are much more accurate than previously-proven bounds would suggest, and that NS often provides a much better approximation of removal effects than IF, but to the best of our knowledge, no previous quantitative analysis comes close to explaining this phenomenon.

In this work, we provide the first theoretical analysis whose guarantees are tight enough to explain *why* and *when* the NS data attribution outperforms IF.

## 1.1. Data Attribution Models

### 1.1.1. INFLUENCE FUNCTIONS (IF)

IF, also known as infinitesimal jackknife (IJ) (Jaeckel, 1972) utilizes a first order approximation to the effect of downweighting a sample on the model parameters. Defining $\theta_{(\cdot)} : \mathbb{R}^n \to \mathbb{R}^d$ as the function that takes a set of weights $\mathbf{w} \in \mathbb{R}^n$, and optimizes the model on the weighted samples $\theta_{\mathbf{w}} = \operatorname{argmin}_{\theta \in \Omega_\theta} \{\sum_{i=1}^{n} w_i \ell_i(\theta)\}$, IF employs a first order Taylor series in $\mathbf{w}$. The resulting estimated change in the model parameters is given by the following formula (see

---

[2]Some analyses require only that the loss is strongly convex in some subset of the optimization domain. Assumption 7 of (Rahnama Rad & Maleki, 2020) only require that the loss is strongly convex along the path between $\hat{\theta}$ and $\hat{\theta}_T$ and Assumption 1 of (Wilson et al., 2020) only require that the loss is strongly convex in a ball around $\hat{\theta}_T$. However, these papers do not give an argument for why the loss should be more convex in this area than anywhere else in the optimization domain. Therefore, without prior knowledge on $\hat{\theta}_T$, it is not clear how one could get a concrete guarantee beyond the one in Theorem 1.7.

e.g., (Hampel et al., 1986) for a derivation),

$$\theta_{\mathbf{w}}^{\mathrm{IF}} := \theta_{\mathbf{w}=\mathbf{1}} + \frac{\partial \theta_{\mathbf{w}}}{\partial \mathbf{w}}(\mathbf{w} - \mathbf{1}) = \hat{\theta} + \mathbf{H}^{-1} \sum_{i=1}^{n} (1 - w_i) \mathbf{g}_i \,,$$

where $\mathbf{H}$ is the Hessian of the loss $L(\theta) = \sum_{i=1}^{n} \ell_i(\theta)$ evaluated at $\hat{\theta} = \theta_{\mathbf{w}=\mathbf{1}}$, and $\mathbf{g}_i$ is the gradient of $\ell_i$ at $\hat{\theta}$.

Influence functions are ubiquitous due to their simplicity and applicability for downstream tasks (Giordano et al., 2019; Broderick et al., 2020) and were made more applicable to modern machine learning with fast algorithms for estimating them without explicitly inverting the Hessian (Koh & Liang, 2017; Guo et al., 2021; Park et al., 2023; Grosse et al., 2023).

However, while IF data attribution appears good at capturing the qualitative behavior of a model's change when dropping a data point, it often misses the scale of the change and significantly underestimates removal effects, especially in the high-dimensional regime (Koh et al., 2019).

### 1.1.2. SINGLE NEWTON STEP (NS)

NS data attribution, dating to (Pregibon, 1981) and explored recently by (Koh & Liang, 2017; Rahnama Rad & Maleki, 2020; Koh et al., 2019; Wilson et al., 2020; Huang et al., 2025) offers somewhat higher accuracy than the IF approach. Here we approximate $\hat{\theta}_T$ using a single step of the Newton algorithm initialized at $\hat{\theta}$:

$$\theta_{\mathbf{w}}^{\mathrm{NS}} = \hat{\theta} + \mathbf{H}_{\mathbf{w}}^{-1} \sum_{i=1}^{n} (1 - w_i) \mathbf{g}_i \,,$$

where $\mathbf{H}_{\mathbf{w}}$ is the Hessian of the weighted loss function $L_{\mathbf{w}} = \sum_{i \in [n]} w_i \ell_i$.

NS appears almost equivalent to the influence function, except for the key difference that it takes into account the change to the Hessian due to dropping the samples. Moreover, when the loss is a quadratic function of the model (e.g., in ordinary least squares regression), NS data attribution is precise (i.e., $\theta_{\mathbf{w}}^{\mathrm{NS}} = \hat{\theta}_{\mathbf{w}}$), since Newton iteration converges in a single step.

NS is rarely used at scale because it has much higher query complexity than IF: IF can reuse a single inverse Hessian across many drop-set queries, whereas NS must update the Hessian inverse for each query. This extra cost can be worthwhile in smaller applications – in their "Group Influence" paper, Koh et al. (2019) showed empirically that NS can be much more accurate than IF on real-world datasets. Due to the structural similarity between IF and NS, NS also serves as a useful proxy for analyzing IF accuracy (Rahnama Rad & Maleki, 2020; Giordano et al., 2019; Koh et al., 2019).

Despite this, prior to this work, our theoretical understanding of the accuracy of NS was very limited, and in particular

we did not have a good explanation for the empirical observation of Koh et al. (2019) that NS is more accurate than IF.

## 1.2. Our Results

We analyze the accuracy of IF and NS in three settings, with three main results. Our first result applies in the general setting of empirical risk minimization problems for any smooth loss $L$. We place assumptions on the *local* strong convexity and *local* (higher-order) Lipschitzness of $L$ only in a small neighborhood of the Newton step itself, and show that under those assumptions the NS data model is quantitatively accurate. We offer a quantitative statement and comparison to prior work later.

**Theorem 1.1** (NS Accuracy Under Local Conditions (Informal, see Theorem 1.11)). *If $L_T = \sum_{i \notin T} \ell_i$ is strongly convex in a neighborhood of the first Newton step starting from $\hat{\boldsymbol{\theta}}$, and the Hessian of $L_T$ is (mildly) Lipschitz along the first Newton step, then the output of the first Newton step $\hat{\boldsymbol{\theta}}_T^{\mathrm{NS}}$ is close to a local minimizer of the loss $L_T$.*

*In particular, if $L_T$ is convex everywhere, then this minimizer is also the global minimizer, meaning that the output of the first Newton step is close to the full retrain $\hat{\boldsymbol{\theta}}_T$.*

Assumptions like local strong convexity or local Lipschitzness allow great generality across different loss functions and datasets, but can be difficult to interpret quantitatively, obscuring the significant differences observed in practice among various data models. Our second main result addresses a much more concrete setting, where we can hope to prove sharp quantitative bounds on the accuracy of data models, using Theorem 1.1. We focus on logistic regression with Gaussian features as a deliberately simple toy model: the goal is not to model real data, but to obtain a concrete setting in which the abstract quantities in Theorem 1.1 can be evaluated and compared to prior bounds. In this model we show that:

**Theorem 1.2** (NS is more accurate than IF for both average–case and adversarial drop-sets). *Suppose $L$ is the empirical logistic loss corresponding to $n$ independent samples $(\mathbf{x}_1, y_1), \ldots, (\mathbf{x}_n, y_n) \in \mathbb{R}^d \times \{0, 1\}$ from a distribution $(\mathbf{x}, y)$ with $\mathbf{x} \sim N(0, I)$, with population loss minimizer $\boldsymbol{\theta}^*$ having $\|\boldsymbol{\theta}^*\| = \Theta(1)$. For $T \subseteq [n]$, let $\hat{\boldsymbol{\theta}}_T^{\mathrm{IF}}$ be the IF estimate of $\hat{\boldsymbol{\theta}}_T$ and let $\hat{\boldsymbol{\theta}}_T^{\mathrm{NS}}$ be the NS estimate of $\hat{\boldsymbol{\theta}}_T$. Finally, assume that $k, d \ll \sqrt{n}$.*

*Then, the upper bounds given by Theorem 1.1, are asymptotically tight up to $\mathrm{polylog}(n)$ factors. In particular, with high probability over the training data*

$$\mathbb{E}_{T \in \binom{[n]}{k}} \left[ \left\| \hat{\boldsymbol{\theta}}_T - \hat{\boldsymbol{\theta}}_T^{\mathrm{NS}} \right\|_2 \right] = \widetilde{\Theta}\left( \frac{kd}{n^2} \right),$$

$$\max_{T \in \binom{[n]}{k}} \left\{ \left\| \hat{\boldsymbol{\theta}}_T - \hat{\boldsymbol{\theta}}_T^{\mathrm{NS}} \right\|_2 \right\} = \widetilde{\Theta}\left( \frac{k^2 + kd}{n^2} \right).$$

*and*

$$\mathbb{E}_{T \in \binom{[n]}{k}} \left[ \left\| \hat{\boldsymbol{\theta}}_T^{\mathrm{NS}} - \hat{\boldsymbol{\theta}}_T^{\mathrm{IF}} \right\|_2 \right] = \widetilde{\Theta}\left( \frac{k^{3/2}d^{1/2} + k^{1/2}d^{3/2}}{n^2} \right),$$

$$\max_{T \in \binom{[n]}{k}} \left\{ \left\| \hat{\boldsymbol{\theta}}_T^{\mathrm{NS}} - \hat{\boldsymbol{\theta}}_T^{\mathrm{IF}} \right\|_2 \right\} = \widetilde{\Theta}\left( \frac{k^2 + k^{1/2}d^{3/2}}{n^2} \right).$$

Theorem 1.2 bounds the accuracy of the NS estimate and the difference between the NS and IF estimates. This naturally yields upper and lower bounds on the IF error itself via the triangle inequality

$$\left\| \hat{\boldsymbol{\theta}}_T^{\mathrm{IF}} - \hat{\boldsymbol{\theta}}_T \right\|_2 = \left\| \hat{\boldsymbol{\theta}}_T^{\mathrm{IF}} - \hat{\boldsymbol{\theta}}_T^{\mathrm{NS}} \right\|_2 \pm \left\| \hat{\boldsymbol{\theta}}_T^{\mathrm{NS}} - \hat{\boldsymbol{\theta}}_T \right\|_2.$$

In particular, whenever the NS error is smaller than the IF–NS gap, the IF error is essentially determined by the IF–NS gap and NS is genuinely closer to retraining. In particular, this occurs whenever $d \gg k$, implying that NS is significantly more accurate in this regime.

We work in the $n \gg k^2, d^2$ regime to simplify our analysis. We do not expect a phase transition to occur when this assumption is violated. For random drop-sets, we expect the same scalings to remain tight even for larger $k, d$, up to minor changes detailed in Appendix C.

Although Theorem 1.2 focuses on logistic regression for the sake of analytic tractability, little about our analysis is specialized to that setting. We thus believe that our results provide insight into the scaling laws we can expect for the error of IF and NS approximations for a wide range of ERM-based estimators, and perhaps even beyond, to deeper and non-convex models. Even restricted to the convex ERM setting, we expect that our new quantitative understanding of IF and NS approximation accuracy will lead to significant improvements in downstream applications like unlearning, differential privacy, and cross-validation, which rely on such quantitative error bounds.

## 1.3. Related Work and Previous State-of-the-Art

**History and Applications** Leave-1-out and leave-$k$-out models, and fast approximations thereof, have a long history in statistics, especially for robustness and sensitivity analysis and uncertainty quantification. The jackknife (Tukey, 1958), infinitesimal jackknife (Jaeckel, 1972), the bootstrap (Efron, 1992), PRESS statistics (Allen, 1974), and Cook's regression diagnostics (Cook, 1977) are important examples of classical techniques for understanding robustness and uncertainty for estimators and regressors using the effect of one or more data point removals.

IF and NS methods originated in early work on robust statistics (Hampel, 1974; Jaeckel, 1972; Pregibon, 1981), with precursors like the *von Mises expansion* dating to the 1940s (Mises, 1947). IFs have historically seen broad use as a proof technique in mathematical statistics, (Hampel, 1974; Pfanzagl, 1982; Bickel et al., 1993; van der Vaart, 1998), but in this paper we are focused instead on their use as an algorithmic tool for data attribution. In spite of this 60+ year history, no prior work analyzes the accuracy of IF or NS as data attributions in finite-sample (non-asymptotic) settings without unrealistic strong convexity assumptions; in this work we provide such an analysis, leading to the first quantitatively tight error bounds for IF and NS data attribution methods in a natural finite-sample learning problem.

Both IF and NS data attribution methods are now used in a wide variety of downstream applications. Both are used for data attribution in neural networks (Park et al., 2023; Basu et al., 2021; Bae et al., 2022; Engstrom et al., 2025; Choe et al., 2025; Mlodozeniec et al., 2025), albeit with significant questions remaining about their accuracy as a data attribution tool in the neural-net setting. Beyond deep learning, data attribution via IF and NS is a core technique in *machine unlearning* for convex risk minimization problems (Sekhari et al., 2021; Neel et al., 2021; Suriyakumar & Wilson, 2022), for robustness auditing/finding highly influential sets of samples (Broderick et al., 2020; Rubinstein & Hopkins, 2025a; Huang et al., 2025; Hu et al., 2024), approximate cross-validation (Rahnama Rad & Maleki, 2020; Wilson et al., 2020), and even evaluation of model fairness (Ghosh et al., 2023). IF is additive in the samples, so the IF-approximate effect of dropping a set is the sum of leave-1-out effects – this additivity property is crucial for some downstream applications. But IF is typically much less accurate than NS; recent works such as (Lev & Wilson, 2026; Rubinstein & Hopkins, 2025b; Zou et al., 2025b; Huang et al., 2025) propose refinements of IF and NS which can improve the tradeoffs between running time and approximation accuracy.

**State-of-the-Art: Accuracy Guarantees under Global Strong Convexity Assumptions**  Motivated by the extensive downstream applications, several recent works analyze the accuracy of IF and NS methods (Rahnama Rad & Maleki, 2020; Koh et al., 2019; Giordano et al., 2019; Wilson et al., 2020) for convex ERMs. Each of these works uses slightly different assumptions and notations, but roughly speaking they all prove variations of the same high-level statement:

**Theorem 1.3** (Existing Theoretical Guarantees (Informal)). *If the loss function is "sufficiently strongly convex" over the entire optimization domain $\Omega_{\boldsymbol{\theta}}$ and its Hessian is "sufficiently Lipschitz" in this domain, then the single Newton step is "close" to the global optimum.*

The details of Theorem 1.3 and its proof vary slightly from paper to paper, but mostly follow a similar thread. To make things concrete, we consider a specific instantiation of Theorem 1.3.

Denote the gradients of the individual samples by $\mathbf{g}_i = \nabla \ell_i|_{\boldsymbol{\theta}=\hat{\boldsymbol{\theta}}}$ and the Hessian of the loss by $\mathbf{H}_{\boldsymbol{\theta}} = \sum_{i \notin T} \nabla^2 \ell_i|_{\boldsymbol{\theta}}$. Using these notations, previous analyses typically make Assumptions 1.4, 1.5 and 1.6.

**Assumption 1.4** (Lipschitz Hessian). There exists a finite constant[3] $C_{\text{Lip}} < \infty$ such that

$$\forall \boldsymbol{\theta}, \boldsymbol{\theta}' \in \Omega_{\boldsymbol{\theta}} \quad \|\mathbf{H}_{\boldsymbol{\theta}} - \mathbf{H}_{\boldsymbol{\theta}'}\|_{\text{op}} \le C_{\text{Lip}} \|\boldsymbol{\theta} - \boldsymbol{\theta}'\|_2$$

**Assumption 1.5** (Bounded Individual Gradients).

$$C_{\ell} = \max_{i \in [n]} \{\|\mathbf{g}_i\|_2\}$$

**Assumption 1.6** (Global Strong Convexity). There exists a finite constant $C_{\text{op}} < \infty$ such that

$$\forall \boldsymbol{\theta} \in \Omega_{\boldsymbol{\theta}} \quad \left\| \left( \sum_{i \notin T} \nabla^2 \ell_i|_{\boldsymbol{\theta}} \right)^{-1} \right\|_{\text{op}} \le C_{\text{op}}$$

Assumptions 1.4 and 1.5 suffice to (loosely) bound the norm of the gradient after one Newton step, and Assumption 1.6 gives us a quantitative way of converting this into a bound in parameter space.

**Theorem 1.7** (Existing Theoretical Guarantees). *Under Assumptions 1.4, 1.5 and 1.6, the error of the single Newton step data attribution is bounded by*

$$\left\| \hat{\boldsymbol{\theta}}_T - \hat{\boldsymbol{\theta}}_T^{\text{NS}} \right\|_2 = O(C_{\text{Lip}} C_{\text{op}}^3 k^2 C_{\ell}^2).$$

For completeness, we prove Theorem 1.7 in Appendix A.

The biggest limitation of existing approaches is that they require a bound on the spectrum of the inverse Hessian on the entire optimization domain $\Omega_{\boldsymbol{\theta}}$ (Assumption 1.6). While some variants (e.g. (Rahnama Rad & Maleki, 2020)) relax this assumption to one about the spectrum of the inverse Hessian (after samples are removed) on a subset of the domain large enough to include all of $\hat{\boldsymbol{\theta}}, \hat{\boldsymbol{\theta}}^{\text{NS}},$ and $\hat{\boldsymbol{\theta}}_T$, this remains quite restrictive, unless one knows *a priori* that $\hat{\boldsymbol{\theta}}_T$ is close to $\hat{\boldsymbol{\theta}}$ in the first place, which is exactly what data attribution methods aim to discover.

As a simple running example, consider a logistic regression with feature vectors $\mathbf{x}_i \in \mathbb{R}^d$ and labels $y_i \in \{0, 1\}$. The Hessian of a logistic regression is given by $\mathbf{H} =$

---

[3]Keeping with existing nomenclature, we use the term "constant" to mean that $C_{\text{Lip}}$ does not depend on $\boldsymbol{\theta}$, but it may still depend on $n, k, d$.

$\sum_{i \in [n]} \beta_i \mathbf{x}_i \mathbf{x}_i^\mathsf{T}$, where $\beta_i(\boldsymbol{\theta}) = \hat{y}_i(1 - \hat{y}_i)$ are the variances of this sample's prediction $\hat{y}_i = \frac{\exp(\boldsymbol{\theta}^\mathsf{T}\mathbf{x}_i)}{1+\exp(\boldsymbol{\theta}^\mathsf{T}\mathbf{x}_i)}$.

In this case, $\left\| \mathbf{H}_{\boldsymbol{\theta}}^{-1} \right\|_{\mathrm{op}}$ grows rapidly with the norm of the model $\|\boldsymbol{\theta}\|_2$. This is because $\beta_i = O(\exp(-|\boldsymbol{\theta}^\mathsf{T}\mathbf{x}_i|))$, so fixing the direction of $\boldsymbol{\theta}$ and taking its norm to infinity would almost surely cause the Hessian to decay exponentially. Therefore, if we perform our optimization over the domain $\Omega_{\boldsymbol{\theta}} = \mathbb{R}^d$, then $\left\| \mathbf{H}_{\boldsymbol{\theta}}^{-1} \right\|_{\mathrm{op}}$ is not bounded and Assumption 1.6 does not hold.

**Global Strong Convexity via Regularization is Inadequate** One approach to justifying this assumption (Koh et al., 2019; Wilson et al., 2020) is by adding a $L_2$ regularization term to the loss, which gives a global lower bound on the spectrum of the Hessian. However, as Koh et al. note, the regularization coefficient used in practice tends to be very small, meaning that the resulting bound on $C_{\mathrm{op}}$ is enormous, rendering the bound in Theorem B.1 rather weak – in particular, too weak to explain why NS still outperforms IF. Quoting (Koh et al., 2019):

> (Koh et al., 2019): The constraint in Proposition 3 implies that up to $O(1/\lambda^3)$ terms, influence underestimates the Newton approximation. [...] However, $\lambda/\sigma_{\mathrm{max}}$ is quite small in our experiments [...] so the actual correlation of influence is better than predicted by this theory.

This "small $\lambda$" phenomenon is borne out in theory as well as experiments: it is a classical result in learning theory (Dobriban & Wager, 2018) that when optimizing for test-loss, the regularization coefficient $\lambda$ scales with $\lambda = \Theta(\sigma^2 d/n) = O(d/n)$, where $\sigma$ is the "signal-to-noise ratio" of the model. This introduces a "hidden" $n^3$-dependence into Theorem B.1 – even if we make favorable assumptions like $C_{\mathrm{Lip}}, C_\ell, k, d \le O(1)$ the bounds produced by theorems like Theorem B.1 would not even converge when $n \to \infty$:

$$\text{Prior Bounds} \simeq k^2 \times C_\ell^2 \times C_{\mathrm{Lip}} \times C_{\mathrm{op}}^3 \approx \frac{k^2 dn}{n^3\lambda^3} \approx \frac{k^2 n}{d^2}.$$

Finally, we note that this $1/\lambda^3$ scaling can have a significant impact on downstream applications. For instance, in their seminal "Remember What You Want to Forget" paper, in order to certifiably unlearn a set of points, Sekhari et al. must add enough noise to their model to mask the error of their data attribution algorithm (Sekhari et al., 2021)[Lemma 3 and Theorem 3] and (Zou et al., 2025a)[Theorem 11][4],

---

[4]The curvature dependence is suppressed in this theorem which states the one-step Newton neighborhood size as $\widetilde{O}(\sqrt{m^3/n})$, but expanding the constants in Lemma 26 and the control of the event

and as a result this added noise scales with $\lambda^{-3}$. This further highlights the importance of not just having good data attribution methods, but also a good analysis of this data attribution method.

### 1.4. Formal Version of Theorem 1.1

We turn to the formal version of our main theorem for general convex risk minimization (Theorem 1.11). Comparing its informal version, Theorem 1.1, to the existing theory (see Theorem 1.3), the biggest change is that we assume only that the Hessian is well-behaved *in a small neighborhood of the first Newton step* and not over the entire optimization domain (or a subset of the domain whose radius is unknown without computing $\hat{\boldsymbol{\theta}}_T$ itself). At first this might seem like a small difference, but previous proof techniques break down without the global assumption which, as we have shown in Section 1.3, often does not hold in practice and may be hard to verify.

The second difference between Theorem 1.1 and existing theory is that we will require a much milder Lipschitz assumption on the change in the Hessian. Previous analyses require that the Hessian be Lipschitz in operator (Koh et al., 2019; Wilson et al., 2020) or $L_1$ (Giordano et al., 2019) norms, resulting in bounds that scale poorly with $d$ and may be harder to certify algorithmically. For Theorem 1.1, it suffices to show that the Hessian is Lipschitz only in its change along a single direction, resulting in a tighter bound under a more easily verifiable assumption.

Let $T \subseteq [n]$ be a set of $k$ samples to be removed. Let $\mathbf{g}_{\boldsymbol{\theta}} = \sum_{i \notin T} \nabla \ell_i|_{\boldsymbol{\theta}}$ and $\mathbf{H}_{\boldsymbol{\theta}} = \sum_{i \notin T} \nabla^2 \ell_i|_{\boldsymbol{\theta}}$ be the gradient and Hessian of the loss of the retained samples when evaluated at $\boldsymbol{\theta}$. When $\boldsymbol{\theta}$ is not specified below, we will set $\boldsymbol{\theta} = \hat{\boldsymbol{\theta}}$ to be the global optimum of the loss $L$ before samples are removed ($\mathbf{g} = \mathbf{g}_{\boldsymbol{\theta}=\hat{\boldsymbol{\theta}}}, \mathbf{H} = \mathbf{H}_{\boldsymbol{\theta}=\hat{\boldsymbol{\theta}}}$).

Recall that the NS data attribution estimates that

$$\hat{\boldsymbol{\theta}}_T^{\mathrm{NS}} \overset{\text{def}}{=} \hat{\boldsymbol{\theta}} - \mathbf{H}^{-1}\mathbf{g} \approx \hat{\boldsymbol{\theta}}_T .$$

Our first assumption will be that the Hessian is lower-bounded within a neighborhood of $\hat{\boldsymbol{\theta}}_T^{\mathrm{NS}}$. We will allow this neighborhood to assume the shape of any ellipsoid.

More concretely, let $\boldsymbol{\Sigma}$ be any positive-definite matrix (natural choices could be the identity matrix $\boldsymbol{\Sigma} = \mathbf{I}$ or the Hessian $\boldsymbol{\Sigma} = \mathbf{H}$ at the original model $\boldsymbol{\theta} = \hat{\boldsymbol{\theta}}$), and define $\|\mathbf{v}\|_{\boldsymbol{\Sigma}} := \sqrt{\mathbf{v}^\mathsf{T}\boldsymbol{\Sigma}\mathbf{v}}$. Let $r > 0$ be any positive radius, and let $\mathcal{B}$ be the $\boldsymbol{\Sigma}$ ball of radius $r > 0$ around $\hat{\boldsymbol{\theta}}_T^{\mathrm{NS}}$

$$\mathcal{B} = \mathcal{B}_{\boldsymbol{\Sigma},r} \overset{\text{def}}{=} \left\{ \hat{\boldsymbol{\theta}}_T^{\mathrm{NS}} + \mathbf{e} \mid \|\mathbf{e}\|_{\boldsymbol{\Sigma}} \le r \right\} .$$

---

$F_5$ in Lemma 29 reveals inverse-Hessian/regularization factors, including dependence on the order of $(\lambda\nu)^{-3}$.

**Assumption 1.8** (Strong Convexity in $\mathcal{B}$).

$$\forall \boldsymbol{\theta} \in \mathcal{B} \quad \boldsymbol{\Sigma}^{-1/2} \mathbf{H}_{\boldsymbol{\theta}} \boldsymbol{\Sigma}^{-1/2} \succeq C_{\mathrm{op}}^{-1} \mathbf{I} \,.$$

Assumption 1.8 avoids the aforementioned issues with the previous analyses by limiting our evaluation of the Hessian to just a neighborhood of the Newton step (thus avoiding issues with potentially large changes to the Hessian at $\boldsymbol{\theta}$ far from $\hat{\boldsymbol{\theta}}$).

The next assumption tells us that Hessian changes slowly along the first Newton step:

**Assumption 1.9** (Mildly Lipschitz Hessian). $\forall t \in [0,1]$, $\boldsymbol{\theta} = t\hat{\boldsymbol{\theta}} + (1-t)\hat{\boldsymbol{\theta}}_T^{\mathrm{NS}}$ $\left\| (\mathbf{H}_{\boldsymbol{\theta}} - \mathbf{H}) \mathbf{H}^{-1} \mathbf{g} \right\|_{\boldsymbol{\Sigma}^{-1}} \leq C_h$

Finally, similar to some previous analyses (e.g., (Giordano et al., 2019)[Condition 1]), we also require a condition on the relationship between the parameters in our bound:

**Assumption 1.10.** $C_h C_{\mathrm{op}} < r$

Assumption 1.10 encapsulates a non-trivial tradeoff, since on the one hand, increasing $r$ makes the right-hand-side of this condition larger, helping us satisfy it, but on the other hand, this also increases the domain over which $C_{\mathrm{op}}$ is maximized.

Under these assumptions, we can bound the error of the Newton step approximation:

**Theorem 1.11** (Main Result). *Under Assumptions 1.8, 1.9 and 1.10, the retained loss $L_T$ has a local minimizer $\hat{\boldsymbol{\theta}}_T^{\mathrm{loc}}$ satisfying*

$$\left\| \hat{\boldsymbol{\theta}}_T^{\mathrm{loc}} - \hat{\boldsymbol{\theta}}_T^{\mathrm{NS}} \right\|_{\boldsymbol{\Sigma}} \leq C_h C_{\mathrm{op}} \,.$$

*In particular, if $L_T$ is globally convex, then $\hat{\boldsymbol{\theta}}_T^{\mathrm{loc}} = \hat{\boldsymbol{\theta}}_T$ is the global minimizer of $L_T$, and the same bound holds for $\hat{\boldsymbol{\theta}}_T$.*

We prove Theorem 1.11 in Section 3 by combining ideas from self-concordance theory (Bach, 2010; Hsu & Mazumdar, 2024) and the analyses of (Giordano et al., 2019; Wilson et al., 2020).

### 1.5. Additional Data Attribution Methods

A series of recent works (Hu et al., 2024; Huang et al., 2025; Rubinstein & Hopkins, 2025b; Zou et al., 2025b), propose a middle-ground approach to achieving some of the added accuracy of NS data attribution with only a small computational overhead over IF and preserving the additive structure of IF that makes it invaluable for many downstream applications. This approach, called *Rescaled Influence Functions* (RIF) in (Rubinstein & Hopkins, 2025b), works by summing over the NS estimates of the leave-one-out (LOO) effects:

$$\hat{\boldsymbol{\theta}}_T^{\mathrm{RIF}} = \hat{\boldsymbol{\theta}} + \sum_{i \in T} \left( \mathbf{H}_{\backslash\{i\}} \right)^{-1} \mathbf{g}_i \,.$$

The RIF update can be computed efficiently for generalized linear models such as logistic regressions and piecewise linear models such as neural nets with ReLU activation using the Sherman-Morrison formula, because rather than invert the Hessian after dropping all samples in $T$, we just need inverse Hessian for single sample drops. Empirically, this rescaling significantly improves the accuracy of the data attribution estimates compared to IF when tested using a wide variety of non-random sample dropping strategies (e.g., dropping only samples whose gradient has a positive inner product with a fixed direction), often achieving accuracy comparable with that of NS (Rubinstein & Hopkins, 2025b).

Our theoretical analysis has two contributions to this setting. First, we show that RIF is almost as accurate as NS for adversarial data drops (explaining a corresponding empirical observation by (Rubinstein & Hopkins, 2025b)). Second, we propose a mild correction to RIF which *additionally* rescales the influence function by the inverse of the fraction of samples retained. We call this method *Doubly Rescaled Influence Functions* (DRIF).

$$\hat{\boldsymbol{\theta}}_T^{\mathrm{DRIF}} = \hat{\boldsymbol{\theta}} + \frac{n}{n-k} \sum_{i \in T} \left( \mathbf{H}_{\backslash\{i\}} \right)^{-1} \mathbf{g}_i \,.$$

This additional rescaling can clearly be applied efficiently and preserves the additive structure of IF and RIF, which makes it useful for applications such as outlier detection. We show that this additional rescaling suffices to achieve accuracy comparable to NS in both the random and adversarial drop-set settings.

**Theorem 1.12** (RIF and DRIF are almost as accurate as NS for adversarial drop-sets). *Under the assumptions of Theorem 1.2, with high probability on the training samples*

$$\max_{T \in \binom{[n]}{k}} \left\{ \left\| \hat{\boldsymbol{\theta}}_T^{\mathrm{RIF}} - \hat{\boldsymbol{\theta}}_T^{\mathrm{NS}} \right\|_2 \right\} = \widetilde{O}\left( \frac{kd + k^2}{n^2} \right)$$

$$\max_{T \in \binom{[n]}{k}} \left\{ \left\| \hat{\boldsymbol{\theta}}_T^{\mathrm{DRIF}} - \hat{\boldsymbol{\theta}}_T^{\mathrm{NS}} \right\|_2 \right\} = \widetilde{O}\left( \frac{kd + k^2}{n^2} \right) \,.$$

**Theorem 1.13** (The second rescaling is needed for large random drop-sets). *Under the assumptions of Theorem 1.2, with very high probability over the training set*

$$\mathop{\mathbb{E}}_{T \in \binom{[n]}{k}} \left[ \left\| \hat{\boldsymbol{\theta}}_T^{\mathrm{DRIF}} - \hat{\boldsymbol{\theta}}_T^{\mathrm{NS}} \right\|_2 \right] = \widetilde{O}\left( \frac{kd}{n^2} \right)$$

$$\mathop{\mathbb{E}}_{T \in \binom{[n]}{k}} \left[ \left\| \hat{\boldsymbol{\theta}}_T^{\mathrm{RIF}} - \hat{\boldsymbol{\theta}}_T^{\mathrm{DRIF}} \right\|_2 \right] = \widetilde{\Theta}\left( \frac{kd + k^{3/2} d^{1/2}}{n^2} \right) \,.$$

Figure 1 gives an empirical validation of these scalings on synthetic and real data.

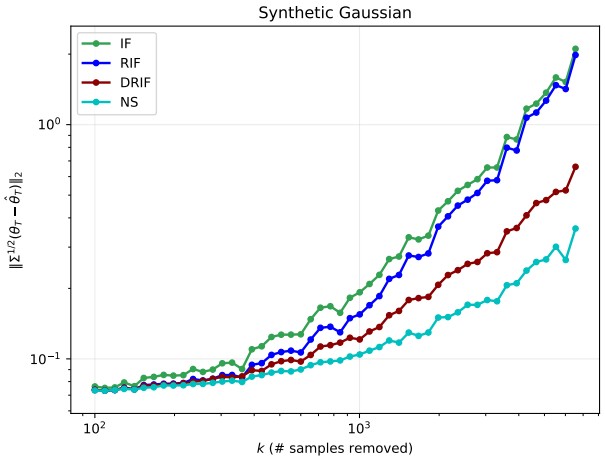

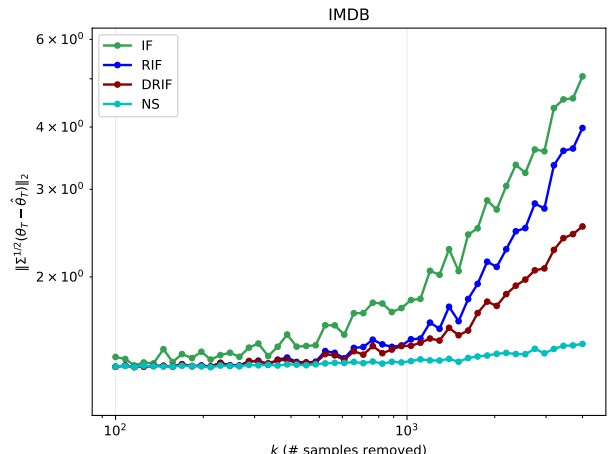

(a) Synthetic Gaussian logistic regression.    (b) IMDB sentiment analysis with frozen BERT embeddings.

*Figure 1.* Empirical validation of IF, NS, RIF, and DRIF. In both panels, the $x$-axis is the number $k$ of uniformly random training points removed, and the $y$-axis is error to full retraining in the Mahalanobis norm induced by the unnormalized second moment matrix $\boldsymbol{\Sigma} = \mathbf{X}^\top \mathbf{X}$. **Left:** synthetic logistic regression with $n = 2^{16}$ Gaussian covariates in dimension $d = 512$ and $\boldsymbol{\theta}^\star = e_1$. **Right:** logistic regression on frozen BERT embeddings of the binary IMDB movie-review sentiment dataset, generated as in (Rubinstein & Hopkins, 2025b). In both experiments, NS and DRIF are substantially more accurate than IF and RIF, especially for larger removals.

## 2. Notation

**General Notations** We use lower-case Greek and Latin letters $(a, b, c, \alpha, \beta, \gamma)$ to denote scalars and indices, bold lower-case letters $(\mathbf{a}, \mathbf{b}, \mathbf{c}, \boldsymbol{\alpha}, \boldsymbol{\beta}, \boldsymbol{\gamma})$ to denote vectors and bold upper-case letters $(\mathbf{A}, \mathbf{B}, \mathbf{C}, \boldsymbol{\Gamma}, \boldsymbol{\Pi})$ to denote matrices and higher order tensors.

Moreover, when working with higher order tensors, we often want to view their product with the tensor of several vectors. We will denote this by $\mathbf{T}(\mathbf{v}_1, \mathbf{v}_2, \ldots, \mathbf{v}_l)$, and sometimes use $\cdot$ to denote existing dimensions that are not fixed.

**Problem Settings** Let $L : \mathbb{R}^d \rightarrow \mathbb{R}$ be a smooth loss function with gradient $\mathbf{g}_{\boldsymbol{\theta}} = \nabla L|_{\boldsymbol{\theta}}$ and Hessian $\mathbf{H}_{\boldsymbol{\theta}} = \nabla^{\otimes 2} L|_{\boldsymbol{\theta}}$, and let $\boldsymbol{\theta}_0 \in \mathbb{R}^d$ be some initial point.

**Asymptotics** We use the notation $\widetilde{O}(\cdot)$, $\widetilde{\Theta}(\cdot)$, and $\widetilde{\Omega}(\cdot)$ to suppress polylogarithmic factors in $n$; that is, for a function $f(n)$, $\widetilde{O}\left(f(n)\right)$ denotes $O(f(n)\text{polylog}(n))$.

## 3. Proof of Theorem 1.11

In this section, we prove Theorem 1.11 which bounds the approximation error of NS data attributions. Recall that our assumptions for Theorem 1.11 were that the loss is strongly convex in a region $\mathcal{B}$ surrounding the NS data attribution $\hat{\boldsymbol{\theta}}_T^{\text{NS}}$ and that the Hessian did not change rapidly along the NS path $[\hat{\boldsymbol{\theta}}, \hat{\boldsymbol{\theta}}_T^{\text{NS}}]$, and our goal is to bound the distance between $\hat{\boldsymbol{\theta}}_T^{\text{NS}}$ and a local minimizer of $L_T$.

Our proof of Theorem 1.11 will combine ideas from exist-

ing analyses of NS data attributions with ideas from self-concordance theory. Our proof strategy will be to first bound the norm of the gradient at the end of the Newton step (Lemma 3.1), then use this to show $L_T$ has a local minimum in $\mathcal{B}$ and that this local minimum satisfies the claim of Theorem 1.11 (Lemma 3.2).

### 3.1. Bounded Gradient at $\hat{\boldsymbol{\theta}}_T^{\text{NS}}$

**Lemma 3.1** (Bounded Gradient at $\hat{\boldsymbol{\theta}}_T^{\text{NS}}$). *The loss gradient at $\hat{\boldsymbol{\theta}}_T^{\text{NS}}$ is bounded by*

$$\left\| \mathbf{g}_{\boldsymbol{\theta} = \hat{\boldsymbol{\theta}}_T^{\text{NS}}} \right\|_{\Sigma^{-1}} \leq C_h \, .$$

Lemma 3.1 and its proof are similar to analogous bounds in many of the previous analyses of NS data attribution:

*Proof of Lemma 3.1.* From the Fundamental Theorem of Calculus (FTC), we have

$$\begin{aligned}
\mathbf{g}_{\hat{\boldsymbol{\theta}}_T^{\text{NS}}} &= \mathbf{g}_{\hat{\boldsymbol{\theta}}} + \int_{\hat{\boldsymbol{\theta}}}^{\hat{\boldsymbol{\theta}}_T^{\text{NS}}} \mathbf{H}_{\boldsymbol{\theta}} \mathrm{d}\boldsymbol{\theta} \\
&= \int_0^1 \left( \mathbf{H}_{\boldsymbol{\theta} = t\hat{\boldsymbol{\theta}}_T^{\text{NS}} + (1-t)\hat{\boldsymbol{\theta}}} - \mathbf{H}_{\hat{\boldsymbol{\theta}}} \right) \left( \hat{\boldsymbol{\theta}}_T^{\text{NS}} - \hat{\boldsymbol{\theta}} \right) \mathrm{d}t \\
&\quad + \underbrace{\mathbf{g}_{\hat{\boldsymbol{\theta}}} + \mathbf{H}_{\hat{\boldsymbol{\theta}}} \left( \hat{\boldsymbol{\theta}}_T^{\text{NS}} - \hat{\boldsymbol{\theta}} \right)}_{=0} \\
&= \int_0^1 \left( \mathbf{H}_{\boldsymbol{\theta} = t\hat{\boldsymbol{\theta}}_T^{\text{NS}} + (1-t)\hat{\boldsymbol{\theta}}} - \mathbf{H}_{\hat{\boldsymbol{\theta}}} \right) \mathbf{H}_{\hat{\boldsymbol{\theta}}}^{-1} \mathbf{g} \mathrm{d}t .
\end{aligned}$$

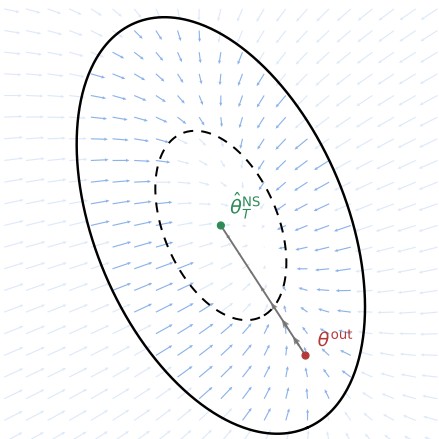

*Figure 2.* Diagrammatic illustration of the proof of Lemma 3.2. The outer ellipsoid represents the ball $\mathcal{B}$ centered at $\hat{\theta}_T^{\mathrm{NS}}$, and the dashed inner ellipsoid represents the smaller radius $C_{\mathrm{op}}C_h$. The point $\theta^{\mathrm{out}}$ lies in the shell between these two ellipsoids. The blue arrows depict the gradient-descent field, with arrow lengths reflecting the local gradient magnitude. The gray arrows along the segment from $\hat{\theta}_T^{\mathrm{NS}}$ to $\theta^{\mathrm{out}}$ show the projection of that field onto the segment direction along the entire segment, emphasizing the key step of the proof: the descent direction points inward along this radial direction, so a minimizer over $\overline{\mathcal{B}}$ cannot lie in the shell and must instead lie in the interior.

From Assumption 1.9, the integrand on the right hand side of this equation has bounded $\boldsymbol{\Sigma}^{-1}$ norm, so from the triangle inequality, we have

$$\left\| \int_0^1 \left( \mathbf{H}_{t\hat{\theta}_T^{\mathrm{NS}}+(1-t)\hat{\theta}} - \mathbf{H}_{\hat{\theta}} \right) \mathbf{H}_{\hat{\theta}}^{-1} \mathbf{g}\, \mathrm{d}t \right\|_{\boldsymbol{\Sigma}^{-1}} \leq C_h\,.$$

$\square$

### 3.2. $L_T$ has a Local Minimum $\theta_T^{\mathrm{loc}} \in \mathcal{B}$

**Lemma 3.2.** *Under the assumptions of Theorem 1.11, there exists a point $\theta_T^{\mathrm{loc}} \in \mathcal{B}$ such that*

- $\theta_T^{\mathrm{loc}}$ *is a local minimum of $L_T$, and*

- $\left\| \hat{\theta}_T^{\mathrm{NS}} - \theta_T^{\mathrm{loc}} \right\|_{\boldsymbol{\Sigma}} \leq C_h C_{\mathrm{op}}.$

Lemma 3.2 immediately implies Theorem 1.11.

*Proof of Lemma 3.2.* Consider the image of $L_T$ on the closed ball

$$I = \mathrm{Im}\left( L_T(\overline{\mathcal{B}}) \right) \subseteq \mathbb{R}\,.$$

From our assumption that $L_T$ is smooth (and in particular continuous) and the fact that $\overline{\mathcal{B}}$ is a compact set (it is bounded and closed in $\mathbb{R}^d$), we know that $I$ must be a closed

interval. In particular, $I$ has a minimum, implying that there exists at least one

$$\boldsymbol{\theta}_T^{\mathrm{loc}} \in \underset{\boldsymbol{\theta} \in \mathcal{B}}{\mathrm{argmin}} \left\{ L_T(\boldsymbol{\theta}) \right\}$$

that minimizes $L_T$ on that compact set.

If we can show that any $\boldsymbol{\theta}_T^{\mathrm{loc}}$ which minimizes the loss in $\overline{\mathcal{B}}$ lies in the interior $\boldsymbol{\theta}_T^{\mathrm{loc}} \notin \partial\mathcal{B}$, that would imply that $\boldsymbol{\theta}_T^{\mathrm{loc}}$ is a local minimum of $L_T$, yielding our first claim (this is because the interior of $\mathcal{B}$ is open so by definition it must contain a sufficiently small neighborhood of $\boldsymbol{\theta}_T^{\mathrm{loc}}$). To conclude the proof, it will therefore suffice to show that any $\boldsymbol{\theta}$ which minimizes $L_T$ on $\overline{\mathcal{B}}$ must lie within the stricter ball $\left\| \boldsymbol{\theta} - \hat{\boldsymbol{\theta}}_T^{\mathrm{NS}} \right\|_{\boldsymbol{\Sigma}} \leq C_h C_{\mathrm{op}} < r$.

Let $\boldsymbol{\theta}^{\mathrm{out}}$ be any point in the spherical shell $C_{\mathrm{op}}C_h < \left\| \boldsymbol{\theta}^{\mathrm{out}} - \hat{\boldsymbol{\theta}}_T^{\mathrm{NS}} \right\|_{\boldsymbol{\Sigma}} \leq r$, and let $\mathbf{g}^{\mathrm{out}}$ be the gradient of the loss at $\boldsymbol{\theta}^{\mathrm{out}}$. Applying FTC, we have

$$\mathbf{g}^{\mathrm{out}} - \mathbf{g}_{\hat{\theta}_T^{\mathrm{NS}}} = \int_{\hat{\theta}_T^{\mathrm{NS}}}^{\boldsymbol{\theta}^{\mathrm{out}}} \mathbf{H}_{\boldsymbol{\theta}}\, \mathrm{d}\boldsymbol{\theta}\,.$$

Taking an inner product with the parameter shift $\mathbf{v} := \boldsymbol{\theta}_T^{\mathrm{out}} - \hat{\boldsymbol{\theta}}_T^{\mathrm{NS}}$, we have

$$\langle \mathbf{v}, \mathbf{g}^{\mathrm{out}} - \mathbf{g}_{\hat{\theta}_T^{\mathrm{NS}}} \rangle = \int_0^1 \mathbf{v}^{\mathsf{T}} \mathbf{H}_{\boldsymbol{\theta}=(1-t)\hat{\theta}_T^{\mathrm{NS}}+t\boldsymbol{\theta}^{\mathrm{out}}} \mathbf{v}\, \mathrm{d}t\,.$$

From our strong convexity assumption (Assumption 1.8) and from our assumption that $\boldsymbol{\theta}_T^{\mathrm{out}}$ lives on the shell, we know that this integrand is bounded from below

$$
\begin{aligned}
\mathbf{v}^{\mathsf{T}} \mathbf{H}_{\boldsymbol{\theta}} \mathbf{v} &= \mathbf{v}^{\mathsf{T}} \boldsymbol{\Sigma}^{1/2} \left( \boldsymbol{\Sigma}^{-1/2} \mathbf{H}_{\boldsymbol{\theta}} \boldsymbol{\Sigma}^{-1/2} \right) \boldsymbol{\Sigma}^{1/2} \mathbf{v} \\
&\geq C_{\mathrm{op}}^{-1} \mathbf{v}^{\mathsf{T}} \boldsymbol{\Sigma} \mathbf{v} = C_{\mathrm{op}}^{-1} \|\mathbf{v}\|_{\boldsymbol{\Sigma}}^2 > C_h \|\mathbf{v}\|_{\boldsymbol{\Sigma}}\,.
\end{aligned}
\tag{1}
$$

On the other hand, from Lemma 3.1 and the Cauchy-Schwarz inequality, we know that

$$
\begin{aligned}
\langle \mathbf{v}, \mathbf{g}_{\hat{\theta}_T^{\mathrm{NS}}} \rangle &= \langle \boldsymbol{\Sigma}^{1/2}\mathbf{v}, \boldsymbol{\Sigma}^{-1/2}\mathbf{g}_{\hat{\theta}_T^{\mathrm{NS}}} \rangle \\
&\geq - \left\| \boldsymbol{\Sigma}^{-1/2} \mathbf{g}_{\hat{\theta}_T^{\mathrm{NS}}} \right\|_2 \|\mathbf{v}\|_{\boldsymbol{\Sigma}} \geq -C_h \|\mathbf{v}\|_{\boldsymbol{\Sigma}}
\end{aligned}
\tag{2}
$$

Combining equations (1) and (2) yields the lower bound

$$\langle \mathbf{v}, \mathbf{g}^{\mathrm{out}} \rangle \geq C_{\mathrm{op}}^{-1} \|\mathbf{v}\|_{\boldsymbol{\Sigma}}^2 - C_h \|\mathbf{v}\|_{\boldsymbol{\Sigma}} > 0\,.$$

Therefore, setting

$$\boldsymbol{\theta}' := \hat{\boldsymbol{\theta}}_T^{\mathrm{NS}} + \frac{C_{\mathrm{op}}C_h}{\|\mathbf{v}\|_{\boldsymbol{\Sigma}}} \mathbf{v}\,,$$

and applying FTC once more, we see that

$$L_T(\boldsymbol{\theta}') - L_T(\boldsymbol{\theta}^{\text{out}}) = -\int_{\frac{C_{\text{op}}C_h}{\|\mathbf{v}\|_{\boldsymbol{\Sigma}}}}^{1} \langle \mathbf{v}, \mathbf{g}_{\hat{\boldsymbol{\theta}}_T^{\text{NS}}+t\mathbf{v}} \rangle \mathrm{d}t < 0\,.$$

Therefore $\boldsymbol{\theta}^{\text{out}}$ cannot be the minimum of the loss $L_T$ on $\overline{\mathcal{B}}$ for any $\boldsymbol{\theta}^{\text{out}}$ in this shell, implying that $\boldsymbol{\theta}_T^{\text{loc}}$ must lie in the smaller ball $\left\|\boldsymbol{\theta}_T^{\text{loc}} - \hat{\boldsymbol{\theta}}_T^{\text{NS}}\right\|_{\boldsymbol{\Sigma}} \leq C_{\text{op}}C_h$ so it must be in the interior of $\mathcal{B}$, implying that it is indeed a local minimum of $L_T$ and yielding the desired result.

$\square$

## 4. Discussion and Limitations

Our work provides the first (to our knowledge) analysis of NS and IF data attribution methods which explains quantitatively the significant accuracy advantages of NS, in the context of a simple learning problem (such as logistic regression with Gaussian covariates). Under reasonable assumptions (see Theorem 1.2), this analysis even shows almost-matching upper and lower bounds on the scaling rate of the errors of these approximations with respect to $k, n$, and $d$. We do so via the more-general Theorem 1.11, which shows how to analyze the NS approximation without appeal to non-local strong convexity assumptions. Together, these results place IF and NS on firmer theoretical foundations.

**Future Directions – Machine Unlearning and Beyond** Quantitative accuracy bounds for IF and NS are of more than theoretical interest. One important application of IF and NS data attribution is machine unlearning, where the goal is to quickly "remove" samples from an already-learned model, e.g. to protect copyrighted material or respect "right to be forgotten" user requests (Sekhari et al., 2021; Neel et al., 2021; Suriyakumar & Wilson, 2022). The dominant technique used in machine unlearning methods with provable guarantees is to first use a data attribution method to approximate the data-dropped model, then add noise to the resulting estimate to obtain a differential-privacy-like indistinguishability guarantee. Crucially, the magnitude of this noise – and hence the utility of the resulting model – scales with the best available bound on the accuracy of the data attribution method used. The $1/\lambda^3$ scaling of the best bounds prior to our work (where $\lambda$ is a global $\ell_2$ regularization parameter) is a significant bottleneck in machine unlearning (Sekhari et al., 2021). So, we hope that the better bounds on accuracy of data attribution methods we derive here will lead to much better unlearning algorithms.

**Limitations** Our work is purely theoretical, aiming to explain a phenomenon observed in real-world data (Koh et al., 2019; Rubinstein & Hopkins, 2025b). We focus on relatively simple settings – convex empirical risk minimization, logistic regression, Gaussian data – to sharpen the focus on the core phenomenon we study. Although data attribution for convex models sometimes forms a core component of data attribution for neural nets, for non-convex losses, Theorem 1.11 is an existence result: it shows that a local minimum lies near the first Newton step, but it does not guarantee that a particular optimization algorithm will find this local minimum, and our theorems also do not allow for other "bells and whistles" such as non-smooth regularizers (Suriyakumar & Wilson, 2022). Moreover, our characterization of NS and IF error scalings for Gaussian logistic regression problems in Theorem 1.2 applies only in the large-$n$ regime, requiring $n \gg d^2, k^2$; characterizing the scaling rates allowing $n \geq d$ is an interesting direction for future work. Finally, our analysis focuses on the norms of the estimation errors, even though we may often also be interested in the direction of these errors.

## Impact Statement

This paper presents work whose goal is to advance the field of Machine Learning. There are many potential societal consequences of our work, none which we feel must be specifically highlighted here.

## Acknowledgments

We thank Tamara Broderick, David R. Burt, Tin D. Nguyen, Yunyi Shen, and Vishwak Srinivasan for conversations about data dropping approximations, Sam Park for feedback on the presentation of this work, and Kevin Lucca for suggesting a decoupling technique to simplify some proofs in this work.

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

# A. Proof Theorem 1.7

A common approach to proving Theorem 1.7 (Koh et al., 2019) is to bound the size of the loss gradient at the end of this first Newton step. Let

$$\mathbf{g} = \sum_{i \notin T} \nabla \ell_i|_{\boldsymbol{\theta}=\hat{\boldsymbol{\theta}}} = -\sum_{i \in T} \nabla \ell_i|_{\boldsymbol{\theta}=\hat{\boldsymbol{\theta}}}$$

be the gradient of the loss at the starting point $\hat{\boldsymbol{\theta}}$, and let

$$\mathbf{g}^{\mathrm{NS}} = \sum_{i \notin T} \nabla \ell_i|_{\boldsymbol{\theta}=\hat{\boldsymbol{\theta}}_T^{\mathrm{NS}}}$$

be the gradient of the loss at the end of the first Newton step. By design of the Newton step, as long as the Hessian does not change too much, we expect $\mathbf{g}^{\mathrm{NS}}$ to be small. More concretely, we have

$$\mathbf{g}^{\mathrm{NS}} = \mathbf{g} + \int_{\hat{\boldsymbol{\theta}}}^{\hat{\boldsymbol{\theta}}_T^{\mathrm{NS}}} \mathbf{H}_{\boldsymbol{\theta}} \mathrm{d}\boldsymbol{\theta} = \underbrace{\mathbf{g} + \int_{\hat{\boldsymbol{\theta}}}^{\hat{\boldsymbol{\theta}}_T^{\mathrm{NS}}} \mathbf{H}_{\hat{\boldsymbol{\theta}}} \mathrm{d}\boldsymbol{\theta}}_{=\mathbf{0}} + \int_{\hat{\boldsymbol{\theta}}}^{\hat{\boldsymbol{\theta}}_T^{\mathrm{NS}}} \left(\mathbf{H}_{\boldsymbol{\theta}} - \mathbf{H}_{\hat{\boldsymbol{\theta}}}\right) \mathrm{d}\boldsymbol{\theta},$$

where the first term cancels precisely by design of the Newton step, and the second term is bounded by

$$\left\| \int_{\hat{\boldsymbol{\theta}}}^{\hat{\boldsymbol{\theta}}_T^{\mathrm{NS}}} \left(\mathbf{H}_{\boldsymbol{\theta}} - \mathbf{H}_{\hat{\boldsymbol{\theta}}}\right) \mathrm{d}\boldsymbol{\theta} \right\|_2 = \left\| \int_0^1 \left(\mathbf{H}_{\boldsymbol{\theta}=(1-t)\hat{\boldsymbol{\theta}}+t\hat{\boldsymbol{\theta}}_T^{\mathrm{NS}}} - \mathbf{H}_{\hat{\boldsymbol{\theta}}}\right) \left(\hat{\boldsymbol{\theta}} - \hat{\boldsymbol{\theta}}_T^{\mathrm{NS}}\right) \mathrm{d}t \right\|_2 \leq \quad \text{(triangle inequality)}$$

$$\leq \int_0^1 \left\| \left(\mathbf{H}_{\boldsymbol{\theta}=(1-t)\hat{\boldsymbol{\theta}}+t\hat{\boldsymbol{\theta}}_T^{\mathrm{NS}}} - \mathbf{H}_{\hat{\boldsymbol{\theta}}}\right) \right\|_{\mathrm{op}} \left\| \hat{\boldsymbol{\theta}} - \hat{\boldsymbol{\theta}}_T^{\mathrm{NS}} \right\|_2 \mathrm{d}t \leq \quad \text{(Lipschitz assumption)}$$

$$\leq \int_0^1 C_{\mathrm{Lip}} t \left\| \hat{\boldsymbol{\theta}} - \hat{\boldsymbol{\theta}}_T^{\mathrm{NS}} \right\|_2^2 \mathrm{d}t = \frac{1}{2} C_{\mathrm{Lip}} \left\| \hat{\boldsymbol{\theta}} - \hat{\boldsymbol{\theta}}_T^{\mathrm{NS}} \right\|_2^2 = \frac{1}{2} C_{\mathrm{Lip}} \left\| \mathbf{H}^{-1} \mathbf{g} \right\|_2^2$$

Given a bound on the norm of the gradient at $\hat{\boldsymbol{\theta}}_T^{\mathrm{NS}}$, we can deduce that $\hat{\boldsymbol{\theta}}_T^{\mathrm{NS}}$ is close to the optimum by utilizing our global strong convexity assumption:

$$\left\| \hat{\boldsymbol{\theta}}_T - \hat{\boldsymbol{\theta}}_T^{\mathrm{NS}} \right\|_2 \leq \max_{\boldsymbol{\theta} \in \Omega_{\boldsymbol{\theta}}} \left\{ \left\| \mathbf{H}_{\boldsymbol{\theta}}^{-1} \right\|_{\mathrm{op}} \right\} \left\| \mathbf{g}^{\mathrm{NS}} \right\| \leq \frac{1}{2} C_{\mathrm{Lip}} C_{\mathrm{op}} \left\| \mathbf{H}^{-1} \mathbf{g} \right\|_2^2 \leq \frac{1}{2} C_{\mathrm{Lip}} C_{\mathrm{op}}^3 \left\| \mathbf{g} \right\|_2^2 \leq \frac{C_{\mathrm{Lip}} C_{\mathrm{op}}^3 k^2 C_\ell^2}{2} \tag{3}$$

where $C_\ell := \max_{i \in [n]} \left\{ \left\| \nabla \ell_i|_{\hat{\boldsymbol{\theta}}} \right\|_2 \right\}$ and the last two steps utilized the fact that $\left\| \mathbf{H}^{-1} \right\|_{\mathrm{op}} = \left\| \mathbf{H}_{\boldsymbol{\theta}=\hat{\boldsymbol{\theta}}^{-1}} \right\|_{\mathrm{op}} \leq C_{\mathrm{op}}$ and the triangle inequality.

# B. Asymptotic Analysis - Setting and Theorem Statements

Existing analyses of NS and IF are often parametrized as a function of complex quantities like $C_{\mathrm{op}}$ and $C_h$ above, making it difficult to compare their results. To get a sense for the asymptotic behavior of Theorem 1.11 and how it compares to previous results, we analyze their respective asymptotic behavior for well-behaved logistic regressions.

In particular, we analyze the existing bounds on the NS approximation error (Theorem 1.7), our bound (Theorem 1.11) and the distance between IF and NS, for $L_2$ regularized logistic regressions with loss

$$L(\boldsymbol{\theta}; \mathbf{X}, \mathbf{y}) = \sum_{i \in [n]} \ell(\langle \boldsymbol{\theta}, \mathbf{x}_i \rangle, y_i) + \frac{n\lambda}{2} \|\boldsymbol{\theta}\|_2^2,$$

where $\ell$ is the logistic loss. We perform this analysis for the case of "under-parametrized agnostic logistic regression with normally distributed features".

More concretely, we assume that the samples are drawn i.i.d from some underlying distribution $(\mathbf{x}, y) \sim \mathcal{D}$ over $\mathbb{R}^d \times \{0, 1\}$ such that:

- The problem is under-parametrized and the number of samples being removed is at relatively small $k, d \ll \sqrt{n}$.

- The features are iid normally distributed $\mathbf{x}_i \sim \mathcal{N}(\mathbf{0}, \mathbf{I})$.

- The optimal model for the distribution,

$$\boldsymbol{\theta}^\star \overset{\text{def}}{=} \operatorname*{argmin}_{\boldsymbol{\theta} \in \Omega_{\boldsymbol{\theta}}} \left\{ \mathbb{E}_{(\mathbf{x},y)\sim\mathcal{D}} [\ell(y, \langle \boldsymbol{\theta}, \mathbf{x}\rangle)] + \frac{\lambda}{2} \|\boldsymbol{\theta}\|_2^2 \right\},$$

has norm $\|\boldsymbol{\theta}^\star\|_2 = \Theta(1)$. We do not assume that the labels were generated by this model.

In the unregularized setting $\lambda = 0$ used in the proofs of Theorems B.2, B.3 and B.4, first-order optimality gives

$$\mathbb{E}_{(\mathbf{x},y)\sim\mathcal{D}} [(\sigma(\langle \boldsymbol{\theta}^\star, \mathbf{x}\rangle) - y)\,\mathbf{x}] = \mathbf{0}.$$

These assumptions are meant to provide a toy model where we can analyze the behavior of Theorem 1.11, including Assumption 1.10, and read off how the resulting bounds scale with $n, k$, and $d$.

Under the assumptions above, we can derive nearly matching upper and lower bounds for the existing bounds on NS accuracy and the distance between IF and NS estimates. Throughout this asymptotic analysis, we will say that an event happens *with very high probability* (wvhp) if it happens with all but negligible probability $n^{-\omega(1)}$, and we will use $\widetilde{\cdot}$ to suppress $\mathrm{poly}(\log(n), \log(d), \log(k))$ factors.

**Theorem B.1** (Asymptotic Analysis of Existing Bounds). *Under the assumptions above, with very high probability over the randomness of* $\mathbf{X}, \mathbf{y}$,

$$\mathbb{E}_{T\in\binom{[n]}{k}} [\text{Existing Bounds}] = \Omega\left(\frac{k^2 d}{n^2 \lambda^3}\right).$$

We prove Theorem B.1 in Section G.

The following theorems restate the asymptotic analyses from the main text. Theorems B.2 and B.3 imply Theorem 1.2, and Theorem B.4 implies Theorems 1.12 and 1.13. We prove them in the most challenging setting of $\lambda = 0$, though we expect the same scalings to hold for any $\lambda = O(1)$.

**Theorem B.2** (Asymptotic Analysis of Influence Functions). *Under the assumptions above, with very high probability over the randomness of* $\mathbf{X}, \mathbf{y}$,

$$\mathbb{E}_{T\in\binom{[n]}{k}} \left[\left\|\hat{\boldsymbol{\theta}}_T^{\mathrm{IF}} - \hat{\boldsymbol{\theta}}_T^{\mathrm{NS}}\right\|_2\right] = \widetilde{\Theta}\left(\frac{k^{3/2}d^{1/2} + k^{1/2}d^{3/2}}{n^2}\right),$$

*and the worst case is*

$$\max_{T\in\binom{[n]}{k}} \left\{\left\|\hat{\boldsymbol{\theta}}_T^{\mathrm{IF}} - \hat{\boldsymbol{\theta}}_T^{\mathrm{NS}}\right\|_2\right\} = \widetilde{\Theta}\left(\frac{k^2 + k^{1/2}d^{3/2}}{n^2}\right)$$

**Theorem B.3** (Asymptotic Analysis of Theorem 1.11). *Under the assumptions above, with very high probability over the randomness of* $\mathbf{X}, \mathbf{y}$, *the expected error of the Newton step estimate scales like*

$$\mathbb{E}_{T\in\binom{[n]}{k}} \left[\left\|\hat{\boldsymbol{\theta}}_T - \hat{\boldsymbol{\theta}}_T^{\mathrm{NS}}\right\|_2\right] = \widetilde{\Theta}\left(\frac{kd}{n^2}\right),$$

*and the worst-case error scales like*

$$\max_{T\in\binom{[n]}{k}} \left\{\left\|\hat{\boldsymbol{\theta}}_T - \hat{\boldsymbol{\theta}}_T^{\mathrm{NS}}\right\|_2\right\} = \widetilde{\Theta}\left(\frac{kd + k^2}{n^2}\right)$$

*where in both cases, the upper bound is certified by Theorem 1.11.*

**Theorem B.4** (Asymptotic Analysis of RIF and DRIF). *Under the assumptions above, with very high probability over the randomness of* $\mathbf{X}, \mathbf{y}$, *the expected error of the Newton step estimate scales like*

$$\mathbb{E}_{T\in\binom{[n]}{k}} \left[\left\|\hat{\boldsymbol{\theta}}_T^{\mathrm{DRIF}} - \hat{\boldsymbol{\theta}}_T^{\mathrm{NS}}\right\|_2\right] = \widetilde{O}\left(\frac{kd}{n^2}\right) \qquad \mathbb{E}_{T\in\binom{[n]}{k}} \left[\left\|\hat{\boldsymbol{\theta}}_T^{\mathrm{RIF}} - \hat{\boldsymbol{\theta}}_T^{\mathrm{DRIF}}\right\|_2\right] = \widetilde{\Theta}\left(\frac{kd + k^{3/2}d^{1/2}}{n^2}\right),$$

*and their worst-case error scales like*

$$\max_{T \in \binom{[n]}{k}} \left\{ \left\| \hat{\boldsymbol{\theta}}_T^{\text{RIF}} - \hat{\boldsymbol{\theta}}_T^{\text{NS}} \right\|_2 \right\}, \max_{T \in \binom{[n]}{k}} \left\{ \left\| \hat{\boldsymbol{\theta}}_T^{\text{DRIF}} - \hat{\boldsymbol{\theta}}_T^{\text{NS}} \right\|_2 \right\} = \widetilde{O}\left( \frac{kd + k^2}{n^2} \right)$$

*where in both cases, the upper bound is certified by Theorem 1.11.*

We prove Theorem B.2 in Section E, Theorem B.3 in Section D, and Theorem B.4 in Section F.

## C. Conjectured Behavior with Large $k$ and $d$

The assumption $k, d \ll \sqrt{n}$ is used to simplify concentration arguments. We do not expect the main phenomenon to break for larger $k, d$.

**Conjecture C.1** (Random drop-sets). *For random $T \in \binom{[n]}{k}$, the average-case scalings in Theorem 1.2 remain tight up to polylogarithmic factors for $k, d \leq cn$.*

**Conjecture C.2** (Adversarial drop-sets). *For adversarial $T \in \binom{[n]}{k}$, the NS error scales as*

$$\left\| \hat{\boldsymbol{\theta}}_T^{\text{NS}} - \hat{\boldsymbol{\theta}}_T \right\|_2 = \widetilde{\Theta}\left( \frac{kd + k^2}{n^2} + \frac{k^2 d^{3/2}}{n^3} \right).$$

Here is the heuristic. The NS error is driven by the second-order residual after the first Newton step, and hence by the third-order derivative tensor

$$\mathbf{T}_{\boldsymbol{\theta}} = \sum_i \gamma(\boldsymbol{\theta}, \mathbf{x}_i) \mathbf{x}_i^{\otimes 3}.$$

Unlike a matrix, this third-order tensor can have sample-level spikes. For a retained sample $\mathbf{x}_j$, contracting with $e^{\otimes 3}$ for $e = \mathbf{x}_j / \|\mathbf{x}_j\|$ gives a contribution of order $\|\mathbf{x}_j\|^3 = \Theta(d^{3/2})$. If the Newton direction has norm about $k/n$ and is aligned with such a spike, this creates a gradient residual of order $k^2 d^{3/2}/n^2$, and multiplying by the inverse Hessian contributes the additional parameter error

$$\frac{k^2 d^{3/2}}{n^3}.$$

We do not expect a random drop-set to correlate strongly with these spikes. For adversarial drop-sets, however, the adversary can adapt the construction used in our lower bound by dropping samples aligned with a fixed retained sample, thereby aligning the Newton direction with one of the tensor spikes. This preserves the high-level claim that NS is more accurate than IF for small adversarial drop-sets, although the condition for "small" becomes roughly $k \ll \min(d, n^{2/3})$ rather than $k \ll d$.

## D. Asymptotic Scaling of Newton Step Accuracy

In this appendix we prove Theorem B.3, which gives tight asymptotic bounds on the accuracy of the Newton step estimate under random or adversarial sample removal. Our goal is to establish that

$$\mathbb{E}_{T \in \binom{[n]}{k}} \left[ \| \hat{\boldsymbol{\theta}}_T - \hat{\boldsymbol{\theta}}_T^{\text{NS}} \|_2 \right] = \Omega\left( \frac{kd}{n^2} \right),$$

and that with very high probability over the removal set $T$, this bound is tight up to poly-logarithmic factors,

$$\Pr_{T \in \binom{[n]}{k}} \left[ \left\| \hat{\boldsymbol{\theta}}_T - \hat{\boldsymbol{\theta}}_T^{\text{NS}} \right\|_2 > \widetilde{O}\left( \frac{kd}{n^2} \right) \right] < O\left( \frac{1}{n^{\omega(1)}} \right),$$

under the assumptions stated in Section B.

Moreover, we will show that with very high probability

$$\max_{T \in \binom{[n]}{k}} \left\{ \left\| \hat{\boldsymbol{\theta}}_T - \hat{\boldsymbol{\theta}}_T^{\text{NS}} \right\|_2 \right\} = \widetilde{\Theta}\left( \frac{kd + k^2}{n^2} \right).$$

Throughout the following sections, we will need to analyze the behavior of the gradient and Hessian of the loss, when evaluated at different models $\boldsymbol{\theta}$ and over different subsets of the training set. For any given set $S \subseteq [n]$ and model $\boldsymbol{\theta}$, we denote the gradient and Hessian of the loss on this set of samples at this point by $\mathbf{g}_{\boldsymbol{\theta}}^{S} = \mathbf{g}_S(\boldsymbol{\theta})$ and $\mathbf{H}_{\boldsymbol{\theta}}^{S} = \mathbf{H}_S(\boldsymbol{\theta})$ respectively. When $\boldsymbol{\theta}$ is not specified, we will be referring to the case where $\boldsymbol{\theta} = \hat{\boldsymbol{\theta}}$ is the model trained on the whole training set, and when $S$ is not specified, we will be referring to the case where $S$ is the set of retained samples $S = [n] \setminus T$.

### D.1. Auxiliary bounds.

Our analysis will often use the fact that the following equations hold with very high probability over the randomness of the samples $(\mathbf{x}_i, y_i)$.

$$\|\hat{\boldsymbol{\theta}} - \boldsymbol{\theta}^{\star}\|_2 = \widetilde{O}\left(\sqrt{\frac{d}{n}}\right), \tag{4}$$

$$\left\|\mathbf{H}_{\boldsymbol{\theta}^{\star}}^{-1}\right\|_{\mathrm{op}} = O\left(\frac{1}{n}\right), \qquad \left\|\mathbf{H}_{\hat{\boldsymbol{\theta}}}^{-1}\right\|_{\mathrm{op}} = O\left(\frac{1}{n}\right), \tag{5}$$

$$\left\|\mathbf{H}_{\hat{\boldsymbol{\theta}}} - \mathbf{H}_{\boldsymbol{\theta}^{\star}}\right\|_{\mathrm{op}} \leq C_{\mathrm{Lip}} \cdot \|\hat{\boldsymbol{\theta}} - \boldsymbol{\theta}^{\star}\|_2 = \widetilde{O}\left(\sqrt{nd}\right), \tag{6}$$

$$\left\|\mathbf{H}_{\hat{\boldsymbol{\theta}}}^{-1} - \mathbf{H}_{\boldsymbol{\theta}^{\star}}^{-1}\right\|_{\mathrm{op}} \leq \left\|\mathbf{H}_{\hat{\boldsymbol{\theta}}}^{-1}\right\|_{\mathrm{op}} \left\|\mathbf{H}_{\hat{\boldsymbol{\theta}}} - \mathbf{H}_{\boldsymbol{\theta}^{\star}}\right\|_{\mathrm{op}} \left\|\mathbf{H}_{\boldsymbol{\theta}^{\star}}^{-1}\right\|_{\mathrm{op}} = \widetilde{O}\left(\frac{\sqrt{nd}}{n^2}\right), \tag{7}$$

$$\text{wvhp over the choice of } T \in \binom{[n]}{k} \qquad \|\mathbf{g}_{\hat{\boldsymbol{\theta}}}^{T}\|_2 = \widetilde{O}\left(\sqrt{kd}\right), \qquad \|\mathbf{g}_{\boldsymbol{\theta}^{\star}}^{T}\|_2 = \widetilde{O}\left(\sqrt{kd}\right), \tag{8}$$

$$\text{for all } T \in \binom{[n]}{k} \qquad \|\mathbf{g}_{\hat{\boldsymbol{\theta}}}^{T}\|_2 = \widetilde{O}\left(\sqrt{kd} + k\right), \qquad \|\mathbf{g}_{\boldsymbol{\theta}^{\star}}^{T}\|_2 = \widetilde{O}\left(\sqrt{kd} + k\right). \tag{9}$$

The observation that $\boldsymbol{\theta}^{\star}$ is close to $\hat{\boldsymbol{\theta}}$ (equation (4)) is due to Lemma H.7, the upper bound on the norm of $\mathbf{H}^{-1}$ (equation (5)) is due to Lemma H.9, the Hessian Lipschitzness (equation (6)) is due to Lemma H.13, and the bound on the difference between the Hessian inverses (equation (7)) follows from applying the matrix identity $\mathbf{A}^{-1} - \mathbf{B}^{-1} = \mathbf{A}^{-1}\left(\mathbf{B} - \mathbf{A}\right)\mathbf{B}^{-1}$. The high probability and worst-case bounds on the norm of $\mathbf{g}_{\boldsymbol{\theta}^{\star}}^{T}$ (equations (8) and (9)) are due to Lemma H.3.

Finally, we prove the bounds on $\left\|\mathbf{g}_{\hat{\boldsymbol{\theta}}}^{T}\right\|_2$ in the average and worst-case settings (equations (8) and (9)) in Lemma D.1.

### Bounding the Norm of the Gradient at $\hat{\boldsymbol{\theta}}$

**Lemma D.1** (Gradient difference for held-out subsets). *Let $C > 2 \|\boldsymbol{\theta}^{\star}\|$ be any finite constant. Then, under the assumptions of Section D.1, with very high probability over the randomness of the samples, for any subset $T \subseteq [n]$ of size $k$, we have*

$$\left\|\mathbf{g}_{\hat{\boldsymbol{\theta}}}^{T} - \mathbf{g}_{\boldsymbol{\theta}^{\star}}^{T}\right\|_2 \leq \sup_{\|\boldsymbol{\theta}\|_2 \leq C} \left\{\|\mathbf{H}_{\boldsymbol{\theta}}^{T}\|_{\mathrm{op}}\right\} \cdot \left\|\hat{\boldsymbol{\theta}} - \boldsymbol{\theta}^{\star}\right\|_2 = \widetilde{O}\left((k+d)\right) \cdot \widetilde{O}\left(\sqrt{\frac{d}{n}}\right) = o(\sqrt{kd}), \tag{10}$$

*where the final $o(\sqrt{kd})$ holds whenever $n \gg k^2 + d^2$.*

*Proof.* We first recall that $\mathbf{g}_{\hat{\boldsymbol{\theta}}}^{[n]} = \mathbf{0}$, hence $\mathbf{g}_{\hat{\boldsymbol{\theta}}}^{[n] \setminus T} = -\mathbf{g}_{\hat{\boldsymbol{\theta}}}^{T}$. Since $\boldsymbol{\theta}^{\star}$ and $\hat{\boldsymbol{\theta}}$ are close (from equation (4)), for any $T$ we write

$$\mathbf{g}_{\hat{\boldsymbol{\theta}}}^{T} - \mathbf{g}_{\boldsymbol{\theta}^{\star}}^{T} = \int_0^1 \mathbf{H}_{\boldsymbol{\theta}^{\star} + t(\hat{\boldsymbol{\theta}} - \boldsymbol{\theta}^{\star})}^{T}(\hat{\boldsymbol{\theta}} - \boldsymbol{\theta}^{\star}) \, dt.$$

Taking norms and using submultiplicativity + triangle inequality gives

$$\left\|\mathbf{g}_{\hat{\boldsymbol{\theta}}}^{T} - \mathbf{g}_{\boldsymbol{\theta}^{\star}}^{T}\right\|_2 \leq \sup_{\|\boldsymbol{\theta}\|_2 \leq C} \left\{\|\mathbf{H}_{\boldsymbol{\theta}}^{T}\|_{\mathrm{op}}\right\} \cdot \left\|\hat{\boldsymbol{\theta}} - \boldsymbol{\theta}^{\star}\right\|_2.$$

To bound $\left\|\mathbf{H}_{\boldsymbol{\theta}}^{T}\right\|_{\mathrm{op}}$ uniformly for any (even adversarial) $T$, consider the augmented set $T' = T \cup [n']$ with $n' = (k + d)\mathrm{polylog}(k + d)$. Apply Lemma H.9 to $\mathbf{H}_{\boldsymbol{\theta}}^{T'}$ to obtain

$$\left\|\mathbf{H}_{\boldsymbol{\theta}}^{T'}\right\|_{\mathrm{op}} \leq \widetilde{O}\left(k + d\right) \qquad \text{for all } \|\boldsymbol{\theta}\|_2 \leq C,$$

wvhp, even after a union bound over the $\binom{n}{k}$ choices of $T$.

Since $\mathbf{H}_{\boldsymbol{\theta}}^{T'} \succeq \mathbf{H}_{\boldsymbol{\theta}}^{T}$ (each per-sample Hessian is PSD and $T' \supseteq T$), it follows that

$$\left\| \mathbf{H}_{\boldsymbol{\theta}}^{T} \right\|_{\mathrm{op}} \leq \left\| \mathbf{H}_{\boldsymbol{\theta}}^{T'} \right\|_{\mathrm{op}} \leq \widetilde{O}\left( k + d \right).$$

Combining this with $\left\| \hat{\boldsymbol{\theta}} - \boldsymbol{\theta}^{\star} \right\|_{2} = \widetilde{O}\left( \sqrt{\frac{d}{n}} \right)$ (equation (4)) gives

$$\left\| \mathbf{g}_{\hat{\boldsymbol{\theta}}}^{T} - \mathbf{g}_{\boldsymbol{\theta}^{\star}}^{T} \right\|_{2} \leq \widetilde{O}\left( (k + d) \right) \cdot \widetilde{O}\left( \sqrt{\frac{d}{n}} \right).$$

Under $n \gg k^2 + d^2$, this is $o(\sqrt{kd})$, establishing (10). $\qquad \square$

### D.2. Upper Bounds

In this subsection, we prove the upper bounds of Theorem B.3. To prove these upper bounds, we will want to apply Theorem 1.11 with $r = 1$ and $\boldsymbol{\Sigma} = \mathbf{I}$. To do so, we first need to bound

$$C_h = \sup_{t \in [0,1]} \left\| \left( \mathbf{H}_{t\hat{\boldsymbol{\theta}}+(1-t)\hat{\boldsymbol{\theta}}_{T}^{\mathrm{NS}}} - \mathbf{H}_{\hat{\boldsymbol{\theta}}} \right) \mathbf{H}_{\hat{\boldsymbol{\theta}}}^{-1} \mathbf{g}_{\hat{\boldsymbol{\theta}}} \right\|_{\boldsymbol{\Sigma}^{-1}}.$$

We do so by applying the Fundamental Theorem of Calculus to express the change in the Hessian as an integral of the third-order derivative tensor $\mathbf{T}_{\boldsymbol{\theta}} = \nabla^{\otimes 3} L|_{\boldsymbol{\theta}}$:

$$\mathbf{H}_{\boldsymbol{\theta}} - \mathbf{H}_{\hat{\boldsymbol{\theta}}} = \int_0^1 \mathbf{T}_{\boldsymbol{\theta}'=(1-s)\hat{\boldsymbol{\theta}}+s\boldsymbol{\theta}} \left( \boldsymbol{\theta} - \hat{\boldsymbol{\theta}} \right) \mathrm{d}s = \int_0^1 t \mathbf{T}_{\boldsymbol{\theta}'=(1-s)\hat{\boldsymbol{\theta}}+s\boldsymbol{\theta}} \left( \mathbf{H}_{\hat{\boldsymbol{\theta}}}^{-1} \mathbf{g}_{\hat{\boldsymbol{\theta}}} \right) \mathrm{d}s,$$

where the last step holds for $\boldsymbol{\theta} = (1 - t)\hat{\boldsymbol{\theta}} + t\hat{\boldsymbol{\theta}}_T^{\mathrm{NS}}$.

Applying the triangle inequality and submultiplicativity, we have

$$C_h \leq \sup_{\boldsymbol{\theta} \in \mathrm{line}(\hat{\boldsymbol{\theta}}, \hat{\boldsymbol{\theta}}_T^{\mathrm{NS}})} \left\| \mathbf{T}_{\boldsymbol{\theta}} \left( \mathbf{H}_{\hat{\boldsymbol{\theta}}}^{-1} \mathbf{g}_{\hat{\boldsymbol{\theta}}}, \mathbf{H}_{\hat{\boldsymbol{\theta}}}^{-1} \mathbf{g}_{\hat{\boldsymbol{\theta}}} \right) \right\|_{\boldsymbol{\Sigma}^{-1}} \leq \sup_{\boldsymbol{\theta} \in \mathrm{line}(\hat{\boldsymbol{\theta}}, \hat{\boldsymbol{\theta}}_T^{\mathrm{NS}})} \left\| \mathbf{T}_{\boldsymbol{\theta}} \right\|_{\mathrm{op}} \cdot \left\| \mathbf{H}_{\hat{\boldsymbol{\theta}}}^{-1} \mathbf{g}_{\hat{\boldsymbol{\theta}}} \right\|_2^2.$$

From Lemma H.13, we know that, wvhp,

$$\left\| \mathbf{T}_{\boldsymbol{\theta}} - \mathbb{E}\left[ \mathbf{T}_{\boldsymbol{\theta}} \right] \right\|_{\mathrm{op}} = \widetilde{O}\left( \sqrt{nk + nd} + k^{3/2} + d^{3/2} \right) = o(n),$$

under our assumption that $n \gg k^2 + d^2$. Moreover, since the samples $\mathbf{x}_i$ are gaussian,

$$\left\| \mathbb{E}\left[ \mathbf{T}_{\boldsymbol{\theta}} \right] \right\|_{\mathrm{op}} = O(n).$$

Combining these, we obtain

$$\left\| \mathbf{T}_{\boldsymbol{\theta}} \right\|_{\mathrm{op}} = O(n), \qquad \text{for all } \boldsymbol{\theta} \text{ on the line segment between } \hat{\boldsymbol{\theta}} \text{ and } \hat{\boldsymbol{\theta}}_T^{\mathrm{NS}},$$

yielding the bound

$$\boxed{C_h = O\left( n \times \left\| \mathbf{H}_{\hat{\boldsymbol{\theta}}}^{-1} \mathbf{g}_{\hat{\boldsymbol{\theta}}} \right\|_2^2 \right).}$$

Applying equations (5), (8), and (9), as well as submultiplicativity, we have

$$\left\| \mathbf{H}_{\hat{\boldsymbol{\theta}}}^{-1} \mathbf{g}_{\hat{\boldsymbol{\theta}}} \right\|_2 \leq \left\| \mathbf{H}_{\hat{\boldsymbol{\theta}}}^{-1} \right\|_{\mathrm{op}} \cdot \left\| \mathbf{g}_{\hat{\boldsymbol{\theta}}} \right\|_2 = \widetilde{O}\left( \frac{\sqrt{kd}}{n} \right), \qquad \text{wvhp for fixed } T \in \binom{[n]}{k},$$

and

$$\left\| \mathbf{H}_{\hat{\boldsymbol{\theta}}}^{-1} \mathbf{g}_{\hat{\boldsymbol{\theta}}} \right\|_2 \leq \left\| \mathbf{H}_{\hat{\boldsymbol{\theta}}}^{-1} \right\|_{\mathrm{op}} \cdot \left\| \mathbf{g}_{\hat{\boldsymbol{\theta}}} \right\|_2 = \widetilde{O}\left( \frac{\sqrt{kd}+k}{n} \right), \qquad \text{for worst-case } T.$$

Therefore, with very high probability,

$$C_h = \widetilde{O}\left( \frac{kd}{n} \right) \quad \text{and} \quad C_h = \widetilde{O}\left( \frac{kd+k^2}{n} \right)$$

for the average-case and worst-case drop-sets $T$, respectively.

To apply Theorem 1.11, we also need to bound $C_{\mathrm{op}}$ and check Assumption 1.10. From Lemma H.9, we have $C_{\mathrm{op}} = O(n^{-1})$ in this regime, yielding

$$C_h C_{\mathrm{op}} = \widetilde{O}\left( \frac{kd}{n^2} \right) \ll r,$$

for the average-case, and

$$C_h C_{\mathrm{op}} = \widetilde{O}\left( \frac{kd+k^2}{n^2} \right) \ll r,$$

for the worst-case, allowing us to apply Theorem 1.11 and proving our claim.

### D.3. Lower Bounds

In this subsection, we prove the lower bounds in Theorem B.3.

**Lower Bound on the Gradient Implies a Lower Bound on NS Accuracy**

**Lemma D.2** (Newton step error lower bounded by gradient magnitude). *Let $L_T$ denote the drop-set objective with gradient $\mathbf{g}_{\boldsymbol{\theta}}$ and Hessian $\mathbf{H}_{\boldsymbol{\theta}}$. Let $\hat{\boldsymbol{\theta}}_T$ be a minimizer of $L_T$ so that $\mathbf{g}_{\hat{\boldsymbol{\theta}}_T} = \mathbf{0}$, and let the one-step Newton iterate be*

$$\hat{\boldsymbol{\theta}}_T^{\mathrm{NS}} \overset{\mathrm{def}}{=} \hat{\boldsymbol{\theta}} - \mathbf{H}_{\hat{\boldsymbol{\theta}}}^{-1} \mathbf{g}_{\hat{\boldsymbol{\theta}}}.$$

*Then, with very high probability over the samples,*

$$\sup_{\boldsymbol{\theta}} \|\mathbf{H}_{\boldsymbol{\theta}}\|_{\mathrm{op}} \leq O(n), \tag{11}$$

$$\text{and} \qquad \left\| \hat{\boldsymbol{\theta}}_T^{\mathrm{NS}} - \hat{\boldsymbol{\theta}}_T \right\|_2 \geq \frac{1}{\sup_{\boldsymbol{\theta}} \|\mathbf{H}_{\boldsymbol{\theta}}\|_{\mathrm{op}}} \left\| \mathbf{g}_{\hat{\boldsymbol{\theta}}_T^{\mathrm{NS}}} \right\|_2. \tag{12}$$

*In particular, $\left\| \hat{\boldsymbol{\theta}}_T^{\mathrm{NS}} - \hat{\boldsymbol{\theta}}_T \right\|_2 = \Omega\left( \left\| \mathbf{g}_{\hat{\boldsymbol{\theta}}_T^{\mathrm{NS}}} \right\|_2 / n \right).$*

*Proof. Global upper bound on the Hessian.* For logistic loss, $\mathbf{H}_{\boldsymbol{\theta}} = \sum_{i \notin T} \beta_i(\boldsymbol{\theta}) \mathbf{x}_i \mathbf{x}_i^\top$ with weights $\beta_i(\boldsymbol{\theta}) \in (0, 1/4]$. Since each summand is PSD and $\beta_i(\boldsymbol{\theta}) \leq 1/4$,

$$\mathbf{H}_{\boldsymbol{\theta}} \preceq \sum_{i \notin T} \mathbf{x}_i \mathbf{x}_i^\top \preceq \sum_{i=1}^{n} \mathbf{x}_i \mathbf{x}_i^\top.$$

By standard subgaussian covariance concentration, $\sum_{i=1}^{n} \mathbf{x}_i \mathbf{x}_i^\top = n\mathbf{I} \pm \widetilde{O}\left( \sqrt{nd} \right)$ wvhp, hence $\|\mathbf{H}_{\boldsymbol{\theta}}\|_{\mathrm{op}} \leq \left\| n\mathbf{I} \pm \widetilde{O}\left( \sqrt{nd} \right) \right\|_{\mathrm{op}} = O(n)$. This proves (11).

*FTC for the gradient along the segment from $\hat{\boldsymbol{\theta}}_T$ (where $\mathbf{g} = 0$) to $\hat{\boldsymbol{\theta}}_T^{\mathrm{NS}}$.* Define the path $\boldsymbol{\theta}(t) = \hat{\boldsymbol{\theta}}_T + t(\hat{\boldsymbol{\theta}}_T^{\mathrm{NS}} - \hat{\boldsymbol{\theta}}_T)$ for $t \in [0, 1]$. By the Fundamental Theorem of Calculus,

$$\mathbf{g}_{\hat{\boldsymbol{\theta}}_T^{\mathrm{NS}}} - \mathbf{g}_{\hat{\boldsymbol{\theta}}_T} = \int_0^1 \mathbf{H}_{\boldsymbol{\theta}(t)} (\hat{\boldsymbol{\theta}}_T^{\mathrm{NS}} - \hat{\boldsymbol{\theta}}_T) \, dt.$$

Since $\mathbf{g}_{\hat{\boldsymbol{\theta}}_T} = \mathbf{0}$, this becomes

$$\mathbf{g}_{\hat{\boldsymbol{\theta}}_T^{\mathrm{NS}}} = \underbrace{\left( \int_0^1 \mathbf{H}_{\boldsymbol{\theta}(t)} \, dt \right)}_{\mathbf{M}} (\hat{\boldsymbol{\theta}}_T^{\mathrm{NS}} - \hat{\boldsymbol{\theta}}_T).$$

Taking norms and using submultiplicativity,

$$\left\| \mathbf{g}_{\hat{\boldsymbol{\theta}}_T^{\mathrm{NS}}} \right\|_2 \leq \sup_{t \in [0,1]} \left\| \mathbf{H}_{\boldsymbol{\theta}(t)} \right\|_{\mathrm{op}} \cdot \left\| \hat{\boldsymbol{\theta}}_T^{\mathrm{NS}} - \hat{\boldsymbol{\theta}}_T \right\|_2 \leq O(n) \left\| \hat{\boldsymbol{\theta}}_T^{\mathrm{NS}} - \hat{\boldsymbol{\theta}}_T \right\|_2,$$

where the last inequality uses (11). Rearranging yields (12). $\qquad \square$

### Lower-Bounding the Gradient after NS

**Lemma D.3** (One-step Newton residual is quadratically large). *Let $L_T$ be the drop-set objective and define the one-step Newton iterate*

$$\hat{\boldsymbol{\theta}}_T^{\mathrm{NS}} \stackrel{\text{def}}{=} \hat{\boldsymbol{\theta}} - \mathbf{H}_{\hat{\boldsymbol{\theta}}}^{-1} \mathbf{g}_{\hat{\boldsymbol{\theta}}}, \qquad \Delta \stackrel{\text{def}}{=} \mathbf{H}_{\hat{\boldsymbol{\theta}}}^{-1} \mathbf{g}_{\hat{\boldsymbol{\theta}}}.$$

*Assume the conditions of Lemma H.2 and Section D.1 hold, $n \gg k^2 + d^2$, and that the step is sufficiently small:*

$$\|\Delta\|_2 \leq \frac{c_0}{\mathrm{polylog}(n)}$$

*for a sufficiently small absolute constant $c_0 > 0$. Then, with very high probability over the samples, for all $T \in \binom{[n]}{k}$,*

$$\langle \hat{\boldsymbol{\theta}}, \mathbf{g}_{\hat{\boldsymbol{\theta}}_T^{\mathrm{NS}}} \rangle \leq -c_1 \, n \, \|\Delta\|_2^2 \,,$$

*for an absolute constant $c_1 > 0$ depending only on the bounds $c < C$ from Lemma H.2. Consequently, using Cauchy–Schwarz and $\left\| \hat{\boldsymbol{\theta}} \right\|_2 = \Theta(1)$ (by Lemma H.7 and the assumption $\|\boldsymbol{\theta}^\star\|_2 = \Theta(1)$),*

$$\left\| \mathbf{g}_{\hat{\boldsymbol{\theta}}_T^{\mathrm{NS}}} \right\|_2 \geq \frac{1}{\left\| \hat{\boldsymbol{\theta}} \right\|_2} \left| \langle \hat{\boldsymbol{\theta}}, \mathbf{g}_{\hat{\boldsymbol{\theta}}_T^{\mathrm{NS}}} \rangle \right| = \Omega \left( n \, \|\Delta\|_2^2 \right).$$

*Proof.* By the Fundamental Theorem of Calculus applied to the gradient along the ray $\boldsymbol{\theta}(t) \stackrel{\text{def}}{=} \hat{\boldsymbol{\theta}} - t\Delta$,

$$\mathbf{g}_{\hat{\boldsymbol{\theta}}_T^{\mathrm{NS}}} - \mathbf{g}_{\hat{\boldsymbol{\theta}}} = \int_0^1 \mathbf{H}_{\boldsymbol{\theta}(t)} \, (-\Delta) \, dt = -\int_0^1 \left( \mathbf{H}_{\boldsymbol{\theta}(t)} - \mathbf{H}_{\hat{\boldsymbol{\theta}}} \right) \Delta \, dt - \underbrace{\left( \mathbf{H}_{\hat{\boldsymbol{\theta}}} \Delta \right)}_{= \mathbf{g}_{\hat{\boldsymbol{\theta}}}}.$$

Applying the Fundamental Theorem of Calculus again to the Hessian,

$$\mathbf{H}_{\boldsymbol{\theta}(t)} - \mathbf{H}_{\hat{\boldsymbol{\theta}}} = -\int_0^t \mathbf{T}_{\boldsymbol{\theta}(s)}(\Delta) \, ds, \qquad \text{where } \mathbf{T}_{\boldsymbol{\theta}} \stackrel{\text{def}}{=} \nabla^{\otimes 3} L_T |_{\boldsymbol{\theta}}.$$

Substituting and swapping the integrals yields the standard second-order remainder form for the gradient after one Newton step:

$$\mathbf{g}_{\hat{\boldsymbol{\theta}}_T^{\mathrm{NS}}} = \int_0^1 (1-s) \, \mathbf{T}_{\boldsymbol{\theta}(s)}(\Delta, \Delta) \, ds, \qquad \text{where } \boldsymbol{\theta}(s) = \hat{\boldsymbol{\theta}} - s\Delta.$$

Taking the inner product with $\hat{\boldsymbol{\theta}}$ and using linearity,

$$\langle \hat{\boldsymbol{\theta}}, \mathbf{g}_{\hat{\boldsymbol{\theta}}_T^{\mathrm{NS}}} \rangle = \int_0^1 (1-s) \, \langle \hat{\boldsymbol{\theta}}, \mathbf{T}_{\boldsymbol{\theta}(s)}(\Delta, \Delta) \rangle \, ds.$$

*Sign and size of the integrand.* By Lemma H.2, for each $s \in [0, 1]$,

$$\langle \boldsymbol{\theta}(s), \mathbb{E}\left[ \mathbf{T}_{\boldsymbol{\theta}(s)} \right] (\Delta, \Delta) \rangle = \mathbb{E}\left[ \mathbf{T}_{\boldsymbol{\theta}(s)} \right] (\Delta, \Delta, \boldsymbol{\theta}(s)) = -\Theta(n) \, \|\Delta\|_2^2 \, \|\boldsymbol{\theta}(s)\|_2^2 \,.$$

Since $\left\|\hat{\boldsymbol{\theta}}\right\|_2 = \Theta(1)$ and $\|\Delta\|_2 \leq c_0/\mathrm{polylog}(n)$, we have $\|\boldsymbol{\theta}(s)\|_2 = \Theta(1)$ uniformly in $s$, so

$$\langle \boldsymbol{\theta}(s), \mathbb{E}\left[\mathbf{T}_{\boldsymbol{\theta}(s)}\right](\Delta, \Delta)\rangle \leq -cn \|\Delta\|_2^2$$

for some absolute constant $c > 0$. We now relate the inner product with $\hat{\boldsymbol{\theta}}$ to that with $\boldsymbol{\theta}(s)$:

$$\left| \langle \hat{\boldsymbol{\theta}} - \boldsymbol{\theta}(s), \mathbb{E}\left[\mathbf{T}_{\boldsymbol{\theta}(s)}\right](\Delta, \Delta)\rangle \right| \leq \left\|\hat{\boldsymbol{\theta}} - \boldsymbol{\theta}(s)\right\|_2 \left\|\mathbb{E}\left[\mathbf{T}_{\boldsymbol{\theta}(s)}\right]\right\|_{\mathrm{op}} \|\Delta\|_2^2 \leq O(n)\, s\, \|\Delta\|_2^3,$$

using $\|\mathbb{E}\left[\mathbf{T}_{\boldsymbol{\theta}}\right]\|_{\mathrm{op}} = O(n)$ and $\left\|\hat{\boldsymbol{\theta}} - \boldsymbol{\theta}(s)\right\|_2 = s\|\Delta\|_2$. Thus,

$$\langle \hat{\boldsymbol{\theta}}, \mathbb{E}\left[\mathbf{T}_{\boldsymbol{\theta}(s)}\right](\Delta, \Delta)\rangle \leq -cn \|\Delta\|_2^2 + O(n)\|\Delta\|_2^3.$$

Next, by Lemma H.13 and $n \gg k^2 + d^2$,

$$\left\|\mathbf{T}_{\boldsymbol{\theta}(s)} - \mathbb{E}\left[\mathbf{T}_{\boldsymbol{\theta}(s)}\right]\right\|_{\mathrm{op}} = \widetilde{O}\left(\sqrt{nk + nd} + k^{3/2} + d^{3/2}\right) = o(n),$$

hence

$$\left| \langle \hat{\boldsymbol{\theta}}, (\mathbf{T}_{\boldsymbol{\theta}(s)} - \mathbb{E}\left[\mathbf{T}_{\boldsymbol{\theta}(s)}\right])(\Delta, \Delta)\rangle \right| \leq \left\|\hat{\boldsymbol{\theta}}\right\|_2 o(n) \|\Delta\|_2^2 = o(n)\|\Delta\|_2^2.$$

Combining the expectation term, the alignment correction, and the concentration error, for sufficiently small $c_0$ we obtain uniformly in $s$,

$$\langle \hat{\boldsymbol{\theta}}, \mathbf{T}_{\boldsymbol{\theta}(s)}(\Delta, \Delta)\rangle \leq -\frac{c}{2} n \|\Delta\|_2^2.$$

*Integrating along the path.* Therefore,

$$\langle \hat{\boldsymbol{\theta}}, \mathbf{g}_{\hat{\boldsymbol{\theta}}_T^{\mathrm{NS}}}\rangle = \int_0^1 (1-s)\,\langle \hat{\boldsymbol{\theta}}, \mathbf{T}_{\boldsymbol{\theta}(s)}(\Delta, \Delta)\rangle\, ds \leq -\frac{c}{2} n \|\Delta\|_2^2 \int_0^1 (1-s)\, ds = -\frac{c}{4} n \|\Delta\|_2^2.$$

Renaming constants yields the claimed bound with $c_1 = c/4$. Finally, applying Cauchy–Schwarz gives

$$\left\|\mathbf{g}_{\hat{\boldsymbol{\theta}}_T^{\mathrm{NS}}}\right\|_2 \geq \left\|\hat{\boldsymbol{\theta}}\right\|_2^{-1} \left| \langle \hat{\boldsymbol{\theta}}, \mathbf{g}_{\hat{\boldsymbol{\theta}}_T^{\mathrm{NS}}}\rangle \right| = \Omega\left(n \|\Delta\|_2^2\right),$$

since $\left\|\hat{\boldsymbol{\theta}}\right\|_2 = \Theta(1)$ by Lemma H.7. $\qquad \square$

**Lower-Bounding the Norm of the First Newton Step**

**Lemma D.4** (Average-case gradient norm at $\boldsymbol{\theta}^\star$). *Under the assumptions of Theorem B.3, with very high probability over the samples,*

$$\mathbb{E}_{T \in \binom{[n]}{k}}\left[\left\|\mathbf{g}_{\boldsymbol{\theta}^\star}^T\right\|_2^2\right] = \Omega(kd).$$

*Proof.* Let $\mathbf{u}_i = \mathbf{g}_{\boldsymbol{\theta}^\star}^i$ denote the sample gradients at $\boldsymbol{\theta}^\star$, and let $\mathbf{u} = \frac{1}{n} \sum_{i \in [n]} \mathbf{u}_i$ be their empirical mean.

Because of our definition of $\boldsymbol{\theta}^\star$ as the distribution optimum, we know that the expectation of $\mathbf{u}$ is $\mathbf{0}$, and that it is the sum over $n$ iid subgaussian random variables. Therefore, wvhp

$$\|\mathbf{u}\|_2 = \widetilde{O}\left(\sqrt{\frac{d}{n}}\right).$$

Let $A_{i,j} = \langle \mathbf{u}_i, \mathbf{u}_j \rangle$ be the pairwise inner products of the gradients. We have

$$\sum_{i,j \in [n]} A_{i,j} = \left\| \sum_{i \in [n]} \mathbf{u}_i \right\|_2^2 = \widetilde{O}(dn) .$$

Therefore,

$$\sum_{i \neq j} A_{i,j} = \widetilde{O}(dn) - \sum_{i \in [n]} A_{i,i} .$$

Applying this to our setting, we have

$$\mathop{\mathbb{E}}_{T \in \binom{[n]}{k}} \left[ \| \mathbf{g}_{\boldsymbol{\theta}^\star}^T \|_2^2 \right] = \mathop{\mathbb{E}}_{T \in \binom{[n]}{k}} \left[ \sum_{i,j \in T} A_{i,j} \right] =$$

$$= k \times \mathop{\mathbb{E}}_{i \in [n]} [A_{i,i}] + k(k-1) \times \mathop{\mathbb{E}}_{i \neq j} [A_{i,j}] =$$

$$= \left( 1 - \frac{k-1}{n-1} \right) k \times \mathop{\mathbb{E}}_{i \in [n]} [A_{i,i}] \pm \widetilde{O} \left( \frac{ndk^2}{n^2} \right) .$$

Finally, note that $\| \mathbf{u}_i \|_2^2 \geq \beta_i(\boldsymbol{\theta}^\star)^2 \| \mathbf{x}_i \|_2^2$, so we have

$$\mathop{\mathbb{E}}_{i \in [n]} [A_{i,i}] = \mathop{\mathbb{E}}_{i \in [n]} \left[ \| \mathbf{u}_i \|_2^2 \right] = \frac{1}{n} \operatorname{tr} \left[ \sum_{i \in [n]} \mathbf{g}_i \mathbf{g}_i^\intercal \right] \geq \frac{1}{n} \operatorname{tr} \left[ \sum_{i \in [n]} \beta_i(\boldsymbol{\theta}^\star)^2 \mathbf{x}_i \mathbf{x}_i^\intercal \right] \geq$$

$$\geq \frac{1}{n} \operatorname{tr} \left[ \sum_{i \in [n]} e^{-4} \mathbb{1}_{|\langle \boldsymbol{\theta}^\star, \mathbf{x}_i \rangle| \leq 1} \mathbf{x}_i \mathbf{x}_i^\intercal \right] = \Omega(d) .$$

Therefore,

$$\mathop{\mathbb{E}}_{T \in \binom{[n]}{k}} \left[ \| \mathbf{g}_{\boldsymbol{\theta}^\star}^T \|_2^2 \right] = \Omega(kd) \pm \widetilde{O} \left( \frac{k^2 d}{n} \right) = \Omega(kd) .$$

$\square$

**Lemma D.5** (Worst-case gradient norm at $\boldsymbol{\theta}^\star$). *Under the same assumptions as Lemma D.4, wvhp over the randomness of the samples, there exists (data-dependent) $T \in \binom{[n]}{k}$ such that,*

$$\| \mathbf{g}_{\boldsymbol{\theta}^\star}^T \|_2^2 = \Omega(k^2 + kd).$$

*Proof.* If $k \leq d$, then the claim follows trivially from Lemma D.4. Consider the cases where $k > d$.

Fix any unit vector $\mathbf{u} \in \mathbb{R}^d$ perpendicular to $\boldsymbol{\theta}^\star$. Write

$$r_i \overset{\text{def}}{=} \sigma(\langle \boldsymbol{\theta}^\star, \mathbf{x}_i \rangle) - y_i, \qquad \mathbf{g}_{\boldsymbol{\theta}^\star}^i = r_i \mathbf{x}_i.$$

By Gaussian anti-concentration and the fact that $\mathbf{u} \perp \boldsymbol{\theta}^\star$, there exist constants $c_x, C_\theta, p_0 > 0$ such that

$$\Pr \left( |\langle \mathbf{x}_i, \mathbf{u} \rangle| \geq c_x, \ |\langle \mathbf{x}_i, \boldsymbol{\theta}^\star \rangle| \leq C_\theta \right) \geq p_0.$$

On this event,

$$\sigma(\langle \boldsymbol{\theta}^\star, \mathbf{x}_i \rangle) \in [c_r, 1 - c_r]$$

for a constant $c_r > 0$. Since $y_i \in \{0, 1\}$, this implies

$$|r_i| = |\sigma(\langle \boldsymbol{\theta}^\star, \mathbf{x}_i \rangle) - y_i| \geq c_r.$$

Therefore, for every sample satisfying the event above,

$$\left|\langle \mathbf{g}_{\boldsymbol{\theta}^\star}^i, \mathbf{u}\rangle\right| = |r_i\langle \mathbf{x}_i, \mathbf{u}\rangle| \geq c_r c_x.$$

Let $\mathcal{I}$ be the set of samples satisfying the event above. By concentration, $|\mathcal{I}| = \Omega(n)$ wvhp. Among these samples, at least one of the two signs of $\langle \mathbf{g}_{\boldsymbol{\theta}^\star}^i, \mathbf{u}\rangle$ occurs on $\Omega(n)$ samples. Select $T$ to be any $k$ samples with this common sign. Then

$$\left|\langle \mathbf{g}_{\boldsymbol{\theta}^\star}^T, \mathbf{u}\rangle\right| = \left|\sum_{i \in T}\langle \mathbf{g}_{\boldsymbol{\theta}^\star}^i, \mathbf{u}\rangle\right| \geq k c_r c_x,$$

and hence $\left\|\mathbf{g}_{\boldsymbol{\theta}^\star}^T\right\|_2 = \Omega(k)$. $\qquad\square$

**Lemma D.6** (Newton step lower bounds: average and worst case). *Let $\Delta \overset{\text{def}}{=} \mathbf{H}^{-1}\mathbf{g}$ denote the Newton step for $L_T$ (the loss after dropping a set of samples $T$) at $\hat{\boldsymbol{\theta}}$. Under the assumptions of Section D.1,*

$$\|\Delta\|_2 \geq \frac{1}{O(n)}\left\|\mathbf{g}_{\hat{\boldsymbol{\theta}}}^T\right\|_2.$$

*Consequently, combining with Lemmas D.4 and D.5 and $\mathbf{g}_{\hat{\boldsymbol{\theta}}}^T = \mathbf{g}_{\boldsymbol{\theta}^\star}^T \pm o(\sqrt{kd})$ (Lemma D.1), we have wvhp:*

$$\text{(average-case random T)} \qquad \underset{T \in \binom{[n]}{k}}{\mathbb{E}}\left[\|\Delta\|_2^2\right] = \Omega\left(\frac{kd}{n^2}\right),$$

$$\text{(worst-case adversarial T)} \qquad \max_{T \in \binom{[n]}{k}}\left\{\|\Delta\|_2^2\right\} = \Omega\left(\frac{k^2 + kd}{n^2}\right).$$

*Proof.* From Lemma H.9, we know that wvhp

$$\mathbf{H}_{\boldsymbol{\theta}} = \sum_{i \notin T}\beta_i(\boldsymbol{\theta})\mathbf{x}_i\mathbf{x}_i^\top \preceq \sum_{i=1}^n \mathbf{x}_i\mathbf{x}_i^\top = n\mathbf{I} \pm \widetilde{O}\left(\sqrt{nd}\right).$$

Therefore,

$$\sup_{\boldsymbol{\theta}}\left\{\|\mathbf{H}_{\boldsymbol{\theta}}\|_{\text{op}}\right\} = O(n).$$

From submultiplicativity, for any vector $\mathbf{v}$,

$$\left\|\mathbf{H}^{-1}\mathbf{v}\right\|_2 \geq \|\mathbf{H}\|_{\text{op}}^{-1}\left\|\mathbf{H}\mathbf{H}^{-1}\mathbf{v}\right\|_2 = \|\mathbf{H}\|_{\text{op}}^{-1}\|\mathbf{v}\|_2 \geq \sup_{\boldsymbol{\theta}}\left\{\|\mathbf{H}_{\boldsymbol{\theta}}\|_{\text{op}}\right\}^{-1}\|\mathbf{v}\|_2 \geq \Omega\left(\|\mathbf{v}\|_2/n\right).$$

Applying this with $\mathbf{v} = \mathbf{g} = -\mathbf{g}_{\hat{\boldsymbol{\theta}}}^T$, we obtain the desired bound

$$\|\Delta\|_2 \geq \frac{1}{O(n)}\left\|\mathbf{g}_{\hat{\boldsymbol{\theta}}}^T\right\|_2.$$

From here, combining Lemma D.1 and Lemma D.4 yields the identity

$$\underset{T \in \binom{[n]}{k}}{\mathbb{E}}\left[\left\|\mathbf{g}_{\hat{\boldsymbol{\theta}}}^T\right\|_2^2\right] \geq \underset{T \in \binom{[n]}{k}}{\mathbb{E}}\left[\left\|\mathbf{g}_{\boldsymbol{\theta}^\star}^T\right\|_2^2\right] - 2\sqrt{\underset{T \in \binom{[n]}{k}}{\mathbb{E}}\left[\left\|\mathbf{g}_{\boldsymbol{\theta}^\star}^T\right\|_2^2\right] \times \underset{T \in \binom{[n]}{k}}{\mathbb{E}}\left[\left\|\mathbf{g}_{\hat{\boldsymbol{\theta}}}^T - \mathbf{g}_{\boldsymbol{\theta}^\star}^T\right\|^2\right]} = \Omega(kd).$$

Therefore

$$\underset{T \in \binom{[n]}{k}}{\mathbb{E}}\left[\|\Delta\|_2^2\right] = \Omega\left(\frac{kd}{n^2}\right).$$

Similarly for the worst-case, we have

$$\max_{T \in \binom{[n]}{k}}\left\{\|\Delta\|_2^2\right\} = \Omega\left(\frac{kd + k^2}{n^2}\right).$$

$\qquad\square$

D.3.1. PROOF OF LOWER-BOUNDS OF THEOREM B.3

We now combine Lemmas D.2, D.3 and D.6 to prove the lower bound of Theorem B.3.

*Proof of Lower Bounds of Theorem B.3.* From Lemma D.6, we know that wvhp over the samples, for average-case drop-sets, we have $\|\Delta\|_2 = \Omega\left(\frac{\sqrt{kd}}{n}\right)$ and for worst-case drop-sets, we have $\|\Delta\|_2 = \Omega\left(\frac{k+\sqrt{kd}}{n}\right)$. In both cases, we also know that $\|\Delta\|_2 \ll \frac{1}{\text{polylog}(n)}$ from equations (5) and (9).

From Lemma D.3, we know that the gradient at $\hat{\boldsymbol{\theta}}_T^{\text{NS}}$ is of norm at least $\Omega\left(n\|\Delta\|_2^2\right)$, and from Lemma D.2, we know that the error of the NS estimate is at least

$$\mathbb{E}_{T\in\binom{[n]}{k}}\left[\left\|\hat{\boldsymbol{\theta}}_T - \hat{\boldsymbol{\theta}}_T^{\text{NS}}\right\|_2\right] \geq \Omega\left(\mathbb{E}_{T\in\binom{[n]}{k}}\left[\frac{\left\|\mathbf{g}_{\hat{\theta}_T^{\text{NS}}}\right\|_2}{n}\right]\right) = \Omega\left(\mathbb{E}_{T\in\binom{[n]}{k}}\left[\|\Delta\|_2^2\right]\right) = \Omega\left(\frac{kd}{n^2}\right),$$

for average-case drop-sets and $\Omega\left(\frac{k^2+kd}{n^2}\right)$ for worst-case drop-sets.

$\square$

# E. Asymptotic Analysis of Influence Functions

In this section we will analyze the asymptotic behavior of the difference between the IF and NS estimates, proving Theorem B.2.

## E.1. Upper Bound

We begin by proving the upper bound that for any fixed $T \in \binom{[n]}{k}$, with very high probability

$$\left\|\hat{\boldsymbol{\theta}}_T^{\text{IF}} - \hat{\boldsymbol{\theta}}_T^{\text{NS}}\right\|_2 = \widetilde{O}\left(\frac{(k+d)\sqrt{kd}}{n^2}\right),$$

and that wvhp over the original dataset, the following slightly weaker bound holds uniformly for every $T \in \binom{[n]}{k}$,

$$\left\|\hat{\boldsymbol{\theta}}_T^{\text{IF}} - \hat{\boldsymbol{\theta}}_T^{\text{NS}}\right\|_2 = \widetilde{O}\left(\frac{(k+d)(\sqrt{kd}+k)}{n^2}\right).$$

*Proof.* This will follow from applying the CS inequality to the concentration bounds proven in Appendix H.

In particular, we note that

$$\hat{\boldsymbol{\theta}}_T^{\text{IF}} - \hat{\boldsymbol{\theta}}_T^{\text{NS}} = \left(\mathbf{H}^{[n]}\right)^{-1}\left(\mathbf{H}^T\right)\left(\mathbf{H}^{\backslash T}\right)^{-1}\mathbf{g}^T,$$

and as we showed in Appendix D.1, wvhp, for all drop-sets $\left\|\mathbf{H}^T\right\|_{\text{op}} = \widetilde{O}(k+d)$, and for all drop-sets $\|\mathbf{g}\|_2 = \widetilde{O}\left(\sqrt{kd}+k\right)$. Moreover, for any fixed drop set $T$, with very high-probability $\|\mathbf{g}\|_2 = \widetilde{O}\left(\sqrt{kd}\right)$. Moreover, from Lemma H.8, we know that wvhp $\left\|\left(\mathbf{H}^{[n]}\right)^{-1}\right\|_{\text{op}}, \left\|\left(\mathbf{H}^{\backslash T}\right)^{-1}\right\|_{\text{op}} = O(1/n)$.

Therefore, from the CS inequality,

$$\left\|\hat{\boldsymbol{\theta}}_T^{\text{IF}} - \hat{\boldsymbol{\theta}}_T^{\text{NS}}\right\|_2 = \widetilde{O}\left(\frac{(k+d)(\sqrt{kd}+k)}{n^2}\right),$$

for the worst-case drop-set $T$, and

$$\left\|\hat{\boldsymbol{\theta}}_T^{\text{IF}} - \hat{\boldsymbol{\theta}}_T^{\text{NS}}\right\|_2 = \widetilde{O}\left(\frac{(k+d)\sqrt{kd}}{n^2}\right),$$

wvhp over $T$.

In either case, this bound always dominates our corresponding bound on $\left\|\hat{\boldsymbol{\theta}}_T - \hat{\boldsymbol{\theta}}_T^{\text{NS}}\right\|_2$, yielding the upper bounds of Theorem B.2 by triangle inequality. $\qquad\square$

### E.2. Lower Bounds

We now proceed to prove the lower bound portion of Theorem B.2.

Here, our goal is to show that in expectation over random subsets of the train set, the IF estimate is at least $\widetilde{\Omega}\left(\frac{(k+d)\sqrt{kd}}{n^2}\right)$ from the NS estimate, and that for worst-case data drops, the error is at least $\widetilde{\Omega}\left(\frac{k^2+k^{1/2}d^{3/2}}{n^2}\right)$.

To show the expectation bound, we first prove a bound on the expectation of the error squared:

**Lemma E.1.** *Under the assumptions of Theorem B.2, wvhp over the samples:*

$$\mathbb{E}_{T\in\binom{[n]}{k}}\left[\left\|\hat{\boldsymbol{\theta}}_T^{\text{IF}} - \hat{\boldsymbol{\theta}}_T^{\text{NS}}\right\|_2^2\right] \geq \widetilde{\Omega}\left(\frac{k^3 d + d^3 k}{n^4}\right) .$$

We then conclude a bound on the first moment by showing that this second moment is not driven too much by outliers.

We prove Lemma E.1 by applying the following identity in 2 regimes

$$\hat{\boldsymbol{\theta}}_T^{\text{IF}} - \hat{\boldsymbol{\theta}}_T^{\text{NS}} = \left(\mathbf{H}^{\backslash T}\right)^{-1}\left(\mathbf{H}^T\right)\left(\mathbf{H}^{[n]}\right)^{-1}\mathbf{g}^T .$$

### E.2.1. $k \geq d \cdot \text{polylog}(n)$

By definition, we know that for all $T \in \binom{[n]}{k}$, it holds that $\mathbf{H}^{-1} \succeq \left(\mathbf{H}^{[n]}\right)^{-1}$. Moreover, clearly wvhp over the samples, we have

$$\left(\mathbf{H}^{[n]}\right)^{-1} \succeq 4\left(\sum_{i\in[n]}\mathbf{x}_i\mathbf{x}_i^{\intercal}\right)^{-1} \succeq \Omega\left(\frac{1}{n}\right)\mathbf{I}$$

Therefore,

$$\left\|\hat{\boldsymbol{\theta}}_T^{\text{IF}} - \hat{\boldsymbol{\theta}}_T^{\text{NS}}\right\|_2 \geq \Omega\left(\frac{1}{n^2}\right) \times \sigma_{\min}\left(\mathbf{H}^T\right) \times \left\|\mathbf{g}^T\right\|_2 .$$

**Average-Case Lower Bound** Lemmas D.4 and D.1, tell us that, wvhp over the randomness of the samples,

$$\mathbb{E}_{T\in\binom{[n]}{k}}\left[\left\|\mathbf{g}^T\right\|_2^2\right] = \Omega\left(kd\right) ,$$

and equations (8) and (9) tell us that this bound cannot be driven by outliers so should stay true when conditioning on events that hold wvhp.

In particular, Lemma H.9 implies that, wvhp, $\sigma(\mathbf{H}^T) = \Theta(k)$ when $k \gg d\text{polylog}(d)$ (here $\sigma(\cdot)$ denotes the entire spectrum of singular values), implying that $\sigma_{\min}\left(\mathbf{H}^T\right) = \Omega(k)$.

Therefore,

$$\mathbb{E}_{T\in\binom{[n]}{k}}\left[\left\|\hat{\boldsymbol{\theta}}_T^{\text{IF}} - \hat{\boldsymbol{\theta}}_T^{\text{NS}}\right\|_2^2\right] = \Omega\left(\frac{k^3 d}{n^4}\right) = \Omega\left(\frac{k^3 d + d^3 k}{n^4}\right) ,$$

proving our average-case bound for this regime.

**Worst-Case Lower Bound**  In this case, we build the set $T = T_1 \cup T_2$ by combining two smaller sets of size $\approx k/2$. For $T_1$, we select samples according to Lemma D.5, which states that we can select them in a way that will ensure that

$$\left\| \sum_{i \in T_1} \mathbf{g}^i \right\|_2^2 = \Omega\left(k^2 + kd\right) = \Omega(k^2).$$

We select the other samples at random from the remaining samples. From Lemma H.9, we know that wvhp

$$\sigma_{\min}\left(\mathbf{H}^{T_2}\right) = \Omega(k).$$

We combine these two sets, to get

$$\mathbf{H}^T \succeq \mathbf{H}^{T_2} \Rightarrow \sigma_{\min}\left(\mathbf{H}^T\right) = \Omega(k),$$

and using equation (8) which tells us that wvhp $\left\| \sum_{i \in T_2} \mathbf{g}^i \right\|_2 = \widetilde{O}\left(\sqrt{kd}\right) \ll k$, we know that from the triangle inequality,

$$\left\| \sum_{i \in T} \mathbf{g}^i \right\|_2 \geq \left\| \sum_{i \in T_1} \mathbf{g}^i \right\|_2 - \left\| \sum_{i \in T_2} \mathbf{g}^i \right\|_2 = \Omega(k).$$

Therefore, wvhp, this drop-set satisfies

$$\left\| \hat{\boldsymbol{\theta}}_T^{\mathrm{IF}} - \hat{\boldsymbol{\theta}}_T^{\mathrm{NS}} \right\|_2 = \Omega\left(\frac{k^2}{n^2}\right) = \Omega\left(\frac{k^2 + kd}{n^2}\right).$$

### E.2.2. $k = \widetilde{O}(d)$

Recall that our goal is to analyze the IF-NS error

$$\hat{\boldsymbol{\theta}}_T^{\mathrm{IF}} - \hat{\boldsymbol{\theta}}_T^{\mathrm{NS}} = \left(\mathbf{H}^{\backslash T}\right)^{-1} \left(\mathbf{H}^T\right) \left(\mathbf{H}^{[n]}\right)^{-1} \mathbf{g}^T.$$

Each of the Hessian inverses yields a $1/n$ factor, $\|\mathbf{g}\|_2 \approx \sqrt{kd}$ and $\left\|\mathbf{H}^T\right\|_{\mathrm{op}} \approx k + d$, so as long as we can show that $\mathbf{g}$ does not "miss" the $O(d)$ eigenvalues of $\mathbf{H}^T$, we expect to get at least an $\approx \sqrt{k}d^{3/2}/n^2$ lower bound on the norm of this error. Since we are focusing on the $k = \widetilde{O}(d)$ regime, this is the dominant term in the lower bound of Theorem B.2 which we are trying to prove. We begin with the simplest case of $k = 1$:

**Lemma E.2** (The $k = 1$ Case). *Under the assumptions of Theorem B.2, with very high probability over the samples*

$$\mathbb{E}_{i \in [n]}\left[\left\| \boldsymbol{\theta}_{\{i\}}^{\mathrm{NS}} - \boldsymbol{\theta}_{\{i\}}^{\mathrm{IF}} \right\|_2\right] = \Omega\left(\frac{d^{3/2}}{n^2}\right).$$

*In other words, the lower bound on the IF error holds for $k = 1$.*

Next, we show that directly summing over $k$ such LOO error vectors still yields a large error (i.e., that we do not have significant cancellations).

**Lemma E.3.** *Let $\mathbf{u}_1, \ldots, \mathbf{u}_n \in \mathbb{R}^d$ be any set of $n$ vectors and let $\mathbf{u}$ be their empirical mean*

$$\mathbf{u} \stackrel{\mathrm{def}}{=} \frac{1}{n} \sum_{i \in [n]} (\mathbf{u}_i).$$

*Then,*

$$\mathbb{E}_{T \in \binom{[n]}{k}}\left[\left\| \sum_{i \in T} (\mathbf{u}_i - \mathbf{u}) \right\|_2^2\right] = \left(1 - \frac{k-1}{n-1}\right) \mathbb{E}_{T \in \binom{[n]}{k}}\left[\sum_{i \in T} \|(\mathbf{u}_i - \mathbf{u})\|_2^2\right]$$

Extending this proof technique just a little bit further yields the desired scaling

**Lemma E.4.** *Define*

$$\mathbf{v}_{i,j} = \mathbf{H}^{[j]} \left( \mathbf{H}^{[n]} \right)^{-1} \mathbf{g}_i \, .$$

*Then,*

$$\mathbb{E}_{T \in \binom{[n]}{k}} \left[ \left\| \sum_{i,j \in T} \mathbf{v}_{i,j} \right\|_2^2 \right] = \left( 1 \pm O\left( \frac{k}{n} \right) \right) \mathbb{E}_{T \in \binom{[n]}{k}} \left[ \left\| \sum_{i \in T} \mathbf{v}_{i,i} \right\|_2^2 \right] .$$

*In particular,*

$$\mathbb{E}_{T \in \binom{[n]}{k}} \left[ \left\| \hat{\boldsymbol{\theta}}_T^{\mathrm{NS}} - \hat{\boldsymbol{\theta}}_T^{\mathrm{IF}} \right\|_2^2 \right] = \Omega\left( \frac{kd^3}{n^4} \right) .$$

**Proof of Lemma E.2**

*Proof of Lemma E.2.* Our proof of Lemma E.2 utilizes an analysis of terms of the form $\mathbf{H}^{-1} \mathbf{g}^T$. This form is somewhat difficult to analyze due to the dependence of the inverse Hessian.

From Lemmas H.13 and H.7, we know that with high probability the third order derivative of the loss is globally bounded and the learned model $\hat{\boldsymbol{\theta}}$ is close to population optimum $\boldsymbol{\theta}^\star$. Combined, these yield a bound on the difference between $\mathbf{H}_{\hat{\boldsymbol{\theta}}}$ and $\mathbf{H}_{\boldsymbol{\theta}^\star}$. Moreover, from Lemma H.9, we know that with high probability, the Hessian converges uniformly and from Lemma H.13, we know that with high probability the third order derivative has operator norm at most $O(n)$, so that

$$\mathbf{H} = \mathbf{H}_{\hat{\boldsymbol{\theta}}} \approx \mathbf{H}_{\boldsymbol{\theta}^\star} \approx \mathbf{H}_{\boldsymbol{\theta}^\star}^{[n]} \approx \mathbb{E}\left[ \mathbf{H}_{\boldsymbol{\theta}^\star}^{[n]} \right] = \Theta(n)\mathbf{I},$$

where each of these approximations yields an error that has operator norm at most $\widetilde{O}\left( \sqrt{n(d+k)} + k + d \right) = \widetilde{O}\left( \sqrt{n(d+k)} \right) = o(n)$. Therefore, $\left\| \mathbf{H}^{-1} - \mathbb{E}\left[ \mathbf{H}_{\boldsymbol{\theta}^\star}^{[n]} \right]^{-1} \right\|_{\mathrm{op}} = \widetilde{O}\left( \frac{\sqrt{n(d+k)}}{n^2} \right).$

For the case where $k = 1$ and $T = \{i\}$, we have

$$\left\| \sum_{i,j \in T} \beta_i \mathbf{x}_i \mathbf{x}_i^\mathsf{T} \mathbb{E}\left[ \mathbf{H}_{\boldsymbol{\theta}^\star}^{[n]} \right]^{-1} \mathbf{x}_j \alpha_j \right\|_2 \geq \left\| \mathbb{E}\left[ \mathbf{H}_{\boldsymbol{\theta}^\star}^{[n]} \right] \right\|_{\mathrm{op}}^{-1} |\alpha_i \beta_i| \left\| \mathbf{x}_i \right\|_2^3 \, .$$

It is easy to see that,

$$\left\| \mathbb{E}\left[ \mathbf{H}_{\boldsymbol{\theta}^\star}^{[n]} \right] \right\|_{\mathrm{op}} = n \left\| \mathbb{E}_{\mathbf{x} \sim \mathcal{N}(\mathbf{0}, \mathbf{I})} \left[ \mathbf{x}\mathbf{x}^\mathsf{T} \beta(\langle \boldsymbol{\theta}^\star, \mathbf{x} \rangle) \right] \right\|_{\mathrm{op}} \leq \frac{1}{4} n \left\| \mathbb{E}_{\mathbf{x} \sim \mathcal{N}(\mathbf{0}, \mathbf{I})} \left[ \mathbf{x}\mathbf{x}^\mathsf{T} \right] \right\|_{\mathrm{op}} = O(n) \, ,$$

that, wvhp over the samples, a constant fraction of the samples satisfy $|\langle \boldsymbol{\theta}^\star, \mathbf{x}_i \rangle| \leq C$ and $\|\mathbf{x}_i\|_2^2 = \Theta(d)$. For these samples, $\beta_i(\boldsymbol{\theta}^\star) = \Omega(1)$, and since $y_i \in \{0, 1\}$,

$$|\alpha_i| = |\sigma(\langle \boldsymbol{\theta}^\star, \mathbf{x}_i \rangle) - y_i| = \Omega(1).$$

Thus $|\alpha_i \beta_i| = \Omega(1)$ for a constant fraction of samples, yielding the bound for $k = 1$

$$\mathbb{E}_{i \in [n]} \left[ \left\| \mathbf{H}^{\{i\}} \mathbb{E}\left[ \mathbf{H}_{\boldsymbol{\theta}^\star}^{[n]} \right]^{-1} \mathbf{g}^{\{i\}} \right\|_2 \right] = \Omega\left( \frac{d^{3/2}}{n} \right) .$$

Finally, to conclude our claim, we note that

$$\mathbb{E}_{i \in [n]} \left[ \left\| \mathbf{H}^{\{i\}} \mathbf{H}^{-1} \mathbf{g}^{\{i\}} \right\|_2 \right] \geq \mathbb{E}_{i \in [n]} \left[ \left\| \mathbf{H}^{\{i\}} \mathbb{E}\left[ \mathbf{H}_{\boldsymbol{\theta}^\star}^{[n]} \right]^{-1} \mathbf{g}^{\{i\}} \right\|_2 \right] - \mathbb{E}_{i \in [n]} \left[ \left\| \mathbf{H}^{\{i\}} \left( \mathbb{E}\left[ \mathbf{H}_{\boldsymbol{\theta}^\star}^{[n]} \right]^{-1} - \mathbf{H}^{-1} \right) \mathbf{g}^{\{i\}} \right\|_2 \right] =$$

$$= \Omega\left( \frac{d^{3/2}}{n} \right) - \widetilde{O}\left( \frac{\sqrt{n(d+k)}}{n^2} \times d^{3/2} \right) = \Omega\left( \frac{d^{3/2}}{n} \right) - o\left( \frac{d^{3/2}}{n} \right) = \Omega\left( \frac{d^{3/2}}{n} \right) ,$$

where the first step used the triangle inequality, and the second step combined submultiplicativity, the fact that wvhp $\left\|\mathbf{H}^{\{i\}}\right\|_{\mathrm{op}} \leq \|\mathbf{x}_i\|_2^2 = \widetilde{O}\left(d\right)$, the fact that wvhp $\left\|\mathbf{g}^{\{i\}}\right\|_2 \leq \|\mathbf{x}_i\|_2 = \widetilde{O}\left(\sqrt{d}\right)$, and our bound on the operator norm of $\mathbb{E}\left[\mathbf{H}_{\boldsymbol{\theta}^\star}^{[n]}\right]^{-1} - \mathbf{H}^{-1}$. $\qquad\square$

**Proof of Lemma E.3**

*Proof of Lemma E.3.* Recall that we define $\mathbf{u}_i \in \mathbb{R}^d$ to be any set of vectors and $\mathbf{u} = \frac{1}{n}\sum_{i\in[n]}\mathbf{u}_i$. Let $\mathbf{w}_i = \mathbf{u}_i - \mathbf{u}$ be the centered version of these vectors. Finally, define $A_{i,j} = \langle\mathbf{w}_i, \mathbf{w}_j\rangle$ to be the inner products between these centered vectors.

Because the $\mathbf{w}$ vectors are centered, we have

$$\sum_{i,j\in[n]} A_{i,j} = 0\,.$$

Therefore,

$$\underset{\substack{i,j\in[n]\\i\neq j}}{\mathbb{E}}[A_{i,j}] = \frac{1}{n(n-1)}\sum_{\substack{i,j\in[n]\\i\neq j}} A_{i,j} = -\frac{1}{n(n-1)}\sum_{i\in[n]} A_{i,i} = -\frac{1}{n-1}\underset{i\in[n]}{\mathbb{E}}[A_{i,i}]\,.$$

Therefore,

$$\underset{T\in\binom{[n]}{k}}{\mathbb{E}}\left[\left\|\sum_{i\in T}\mathbf{w}_i\right\|_2^2\right] = \underset{T\in\binom{[n]}{k}}{\mathbb{E}}\left[\sum_{i,j\in T} A_{i,j}\right] = \tag{13}$$

$$= k \times \underset{i\in[n]}{\mathbb{E}}[A_{i,i}] + k(k-1)\times\underset{\substack{i,j\in[n]\\i\neq j}}{\mathbb{E}}[A_{i,j}] = \tag{14}$$

$$= \left(1 - \frac{k-1}{n-1}\right)\underset{T\in\binom{[n]}{k}}{\mathbb{E}}\left[\sum_{i\in T}\|\mathbf{w}_i\|_2^2\right] \tag{15}$$

$$\square$$

**Proof of Lemma E.4**

*Proof of Lemma E.4.* Recall that we defined

$$\mathbf{v}_{i,j} = \mathbf{H}^{\{j\}}\left(\mathbf{H}^{[n]}\right)^{-1}\mathbf{g}_i = \alpha_i\beta_j\mathbf{x}_j\mathbf{x}_j^\mathsf{T}\left(\mathbf{H}^{[n]}\right)^{-1}\mathbf{x}_i\,.$$

From our assumption that $\hat{\boldsymbol{\theta}}$ is the empirical risk minimizer, we have

$$\sum_{i\in[n]}\mathbf{g}_i = \mathbf{0} \Rightarrow \forall j\in[n]\ \sum_{i\in[n]}\mathbf{v}_{i,j} = \mathbf{0} \Rightarrow \sum_{i,j\in[n]}\mathbf{v}_{i,j} = \mathbf{0}\,.$$

We want to analyze the quantity

$$\left\|\sum_{i,j\in T}\mathbf{v}_{i,j}\right\|_2^2 = \sum_{i_1,i_2,j_1,j_2\in T} S_{i_1,i_2,j_1,j_2}\,,$$

where $S_{i_1,i_2,j_1,j_2} = \langle\mathbf{v}_{i_1,j_1}, \mathbf{v}_{i_2,j_2}\rangle$.

Because the $\mathbf{v}$ vectors are unbiased, we have

$$\sum_{i_1,j_1\in[n]} S_{i_1,i_2,j_1,j_2} = 0\,,$$

for any fixed index pair $i_2, j_2 \in [n]$.

Therefore,

$$\sum_{\substack{i_1,j_1\in[n]\\i_1\neq j_1}} S_{i_1,i_2,j_1,j_2} = -\sum_{\substack{i_1,j_1\in[n]\\i_1=j_1}} S_{i_1,i_2,j_1,j_2}.$$

Therefore,

$$\sum_{\substack{i_1,j_1,i_2,j_2\in[n]\\i_1\neq j_1\\i_2\neq j_2}} S_{i_1,i_2,j_1,j_2} = -\sum_{\substack{i_1,j_1,i_2,j_2\in[n]\\i_1=j_1\\i_2\neq j_2}} S_{i_1,i_2,j_1,j_2} = \sum_{\substack{i_1,j_1,i_2,j_2\in[n]\\i_1=j_1\\i_2=j_2}} S_{i_1,i_2,j_1,j_2} = -\sum_{\substack{i_1,j_1,i_2,j_2\in[n]\\i_1\neq j_1\\i_2=j_2}} S_{i_1,i_2,j_1,j_2}.$$

Applying these equalities to our summation of interest, we have

$$\mathbb{E}_{T\in\binom{[n]}{k}}\left[\sum_{i_1,i_2,j_1,j_2\in T} S_{i_1,i_2,j_1,j_2}\right] =$$

$$= \mathbb{E}_{T\in\binom{[n]}{k}}\left[\sum_{\substack{i_1,j_1,i_2,j_2\in T\\i_1\neq j_1\\i_2\neq j_2}} S_{i_1,i_2,j_1,j_2} + 2\sum_{\substack{i_1,j_1,i_2,j_2\in T\\i_1=j_1\\i_2\neq j_2}} S_{i_1,i_2,j_1,j_2} + \sum_{\substack{i_1,j_1,i_2,j_2\in[n]\\i_1=j_1\\i_2=j_2}} S_{i_1,i_2,j_1,j_2}\right] =$$

$$= \left(1-\frac{k-1}{n-1}\right)^2 \mathbb{E}_{T\in\binom{[n]}{k}}\left[\sum_{i,j\in T} S_{i,j,i,j}\right].$$

To understand this last term, we note that

$$\sum_{i,j\in T} S_{i,j,i,j} = \sum_{i,j\in T}\langle \mathbf{v}_{i,i}, \mathbf{v}_{j,j}\rangle = \left\|\sum_{i\in T}\mathbf{v}_{i,i}\right\|_2^2.$$

Therefore, we can complete the first portion of the proof:

$$\mathbb{E}_{T\in\binom{[n]}{k}}\left[\left\|\sum_{i,j\in T}\mathbf{v}_{i,j}\right\|_2^2\right] = \left(1-\frac{k-1}{n-1}\right)^2 \mathbb{E}_{T\in\binom{[n]}{k}}\left[\left\|\sum_{i\in T}\mathbf{v}_{i,i}\right\|_2^2\right] \tag{16}$$

To conclude the desired lower bound on the expected IF error, we first note that

$$\left\|\hat{\boldsymbol{\theta}}_T^{\mathrm{NS}} - \hat{\boldsymbol{\theta}}_T^{\mathrm{IF}}\right\|_2 = \left\|\mathbf{H}^{-1}\mathbf{H}^T\left(\mathbf{H}^{[n]}\right)^{-1}\mathbf{g}^T\right\|_2 \geq \|\mathbf{H}\|_{\mathrm{op}}^{-1}\left\|\mathbf{H}^T\left(\mathbf{H}^{[n]}\right)^{-1}\mathbf{g}^T\right\|_2 \geq$$

$$\geq \min_{T'\in\binom{[n]}{k}}\left\{\left\|\mathbf{H}^{[n]\setminus T'}\right\|_{\mathrm{op}}^{-1}\right\}\left\|\mathbf{H}^T\left(\mathbf{H}^{[n]}\right)^{-1}\mathbf{g}^T\right\|_2 = \Omega\left(\frac{1}{n}\right)\times\left\|\mathbf{H}^T\left(\mathbf{H}^{[n]}\right)^{-1}\mathbf{g}^T\right\|_2.$$

Therefore,

$$\mathbb{E}_{T\in\binom{[n]}{k}}\left[\left\|\hat{\boldsymbol{\theta}}_T^{\mathrm{NS}} - \hat{\boldsymbol{\theta}}_T^{\mathrm{IF}}\right\|_2^2\right] \geq \Omega\left(\frac{1}{n^2}\right)\times\mathbb{E}_{T\in\binom{[n]}{k}}\left[\left\|\mathbf{H}^T\left(\mathbf{H}^{[n]}\right)^{-1}\mathbf{g}^T\right\|_2^2\right] = \Omega\left(\frac{1}{n^2}\right)\mathbb{E}_{T\in\binom{[n]}{k}}\left[\left\|\sum_{i,j\in T}\mathbf{v}_{i,j}\right\|_2^2\right].$$

By utilizing equation (16), we have

$$\mathbb{E}_{T\in\binom{[n]}{k}}\left[\left\|\hat{\boldsymbol{\theta}}_T^{\mathrm{NS}} - \hat{\boldsymbol{\theta}}_T^{\mathrm{IF}}\right\|_2^2\right] \geq \left(1-\frac{k-1}{n-1}\right)^2\Omega\left(\frac{1}{n^2}\right)\mathbb{E}_{T\in\binom{[n]}{k}}\left[\left\|\sum_{i\in T}\mathbf{v}_{i,i}\right\|_2^2\right]$$

But now we can apply Lemma E.3 to $\mathbf{u}_i = \mathbf{v}_{i,i}/n$ to show that

$$\mathbb{E}_{T \in \binom{[n]}{k}} \left[ \left\| \sum_{i \in T} \mathbf{u}_i \right\|_2^2 \right] \geq \mathbb{E}_{T \in \binom{[n]}{k}} \left[ \left\| \sum_{i \in T} \mathbf{u}_i - \mathbf{u} \right\|_2^2 \right] = \left( 1 - \frac{k-1}{n-1} \right) \mathbb{E}_{T \in \binom{[n]}{k}} \left[ \sum_{i \in T} \| \mathbf{u}_i - \mathbf{u} \|_2^2 \right].$$

Applying a win-win argument to the norm of $\mathbf{u}$, we see that either

$$\mathbb{E}_{T \in \binom{[n]}{k}} \left[ \left\| \sum_{i \in T} \mathbf{u}_i \right\|_2^2 \right] \geq \left\| \mathbb{E}_{T \in \binom{[n]}{k}} \left[ \sum_{i \in T} \mathbf{u}_i \right] \right\|_2^2 = \| k\mathbf{u} \|_2^2 = \Omega \left( \frac{kd^3}{n^4} \right),$$

or

$$\mathbb{E}_{T \in \binom{[n]}{k}} \left[ \sum_{i \in T} \| \mathbf{u}_i - \mathbf{u} \|_2^2 \right] \geq k \mathbb{E}_{T \in \binom{[n]}{k}} \left[ \| \mathbf{u}_i \|_2^2 \right] - 2 \sqrt{k \mathbb{E}_{i \in [n]} \left[ \| \mathbf{u}_i - \mathbf{u} \|_2^2 \right]} \times \left( \| k\mathbf{u} \|_2 \right) = \Omega \left( \frac{kd^3}{n^4} \right),$$

from Lemma E.2.

In either case, we have

$$\mathbb{E}_{T \in \binom{[n]}{k}} \left[ \left\| \hat{\boldsymbol{\theta}}_T^{\mathrm{NS}} - \hat{\boldsymbol{\theta}}_T^{\mathrm{IF}} \right\|_2^2 \right] \geq \Omega \left( \frac{kd^3}{n^4} \right),$$

as desired.

$\square$

### E.2.3. CONCLUDING THE AVERAGE-CASE LOWER BOUND

Over the last two subsubsections, we proved Lemma E.1 which states that in both the $k \gg d$ and the $k \leq d\,\mathrm{polylog}(d)$ regimes we have

$$\mathbb{E}_{T \in \binom{[n]}{k}} \left[ \left\| \hat{\boldsymbol{\theta}}_T^{\mathrm{NS}} - \hat{\boldsymbol{\theta}}_T^{\mathrm{IF}} \right\|_2^2 \right] = \widetilde{\Omega} \left( \frac{k^3 d + d^3 k}{n^4} \right).$$

From the upper bounds in Section E.1, we can see that this expectation cannot be driven by outliers (since for all but $n^{-\omega(1)}$ fraction of the drop-sets this bound is tight and for the remaining we have a worst-case upper bound). Therefore,

$$\mathbb{E}_{T \in \binom{[n]}{k}} \left[ \left\| \hat{\boldsymbol{\theta}}_T^{\mathrm{NS}} - \hat{\boldsymbol{\theta}}_T^{\mathrm{IF}} \right\|_2 \right] = \widetilde{\Omega} \left( \frac{k^{3/2} d^{1/2} + d^{3/2} k^{1/2}}{n^2} \right),$$

completing our proof of Theorem B.2.

## F. Asymptotic Analysis of RIF and DRIF

Our guarantees on RIF and DRIF relate to either the distance between the DRIF estimate and the NS estimate or to the difference between the RIF and DRIF estimates.

### F.1. Bounding the Difference Between RIF and DRIF

In Section F.2, we will analyze the difference between DRIF and NS, showing that it is much smaller than the Newton step itself. Before delving into this more complex analysis, we show that it also implies our bounds on the difference between RIF and DRIF using to the fact that the two are linearly related.

From the definition of DRIF,

$$\hat{\boldsymbol{\theta}}_T^{\mathrm{DRIF}} - \hat{\boldsymbol{\theta}} = \frac{n}{n-k} \left( \hat{\boldsymbol{\theta}}_T^{\mathrm{RIF}} - \hat{\boldsymbol{\theta}} \right).$$

Therefore, from the triangle inequality

$$\left\| \hat{\boldsymbol{\theta}}_T^{\mathrm{DRIF}} - \hat{\boldsymbol{\theta}}_T^{\mathrm{RIF}} \right\|_2 = \frac{k}{n} \left\| \hat{\boldsymbol{\theta}}_T^{\mathrm{DRIF}} - \hat{\boldsymbol{\theta}} \right\|_2 = \frac{k}{n} \left( \left\| \hat{\boldsymbol{\theta}}_T^{\mathrm{NS}} - \hat{\boldsymbol{\theta}} \right\|_2 \pm \left\| \hat{\boldsymbol{\theta}}_T^{\mathrm{DRIF}} - \hat{\boldsymbol{\theta}}_T^{\mathrm{NS}} \right\|_2 \right),$$

where $\left\|\hat{\boldsymbol{\theta}}_T^{\text{DRIF}} - \hat{\boldsymbol{\theta}}_T^{\text{NS}}\right\|_2 \ll \left\|\hat{\boldsymbol{\theta}}_T^{\text{NS}} - \hat{\boldsymbol{\theta}}\right\|_2$ is analyzed in Section F.2, and $\left\|\hat{\boldsymbol{\theta}}_T^{\text{NS}} - \hat{\boldsymbol{\theta}}\right\|_2$ is bounded from above / below for both average and worst-case data drops (from equations (5) and (9), and Lemma D.6), yielding the desired bounds

$$\mathop{\mathbb{E}}_{T \in \binom{[n]}{k}} \left[\left\|\hat{\boldsymbol{\theta}}_T^{\text{DRIF}} - \hat{\boldsymbol{\theta}}_T^{\text{RIF}}\right\|_2\right] = \widetilde{\Theta}\left(\frac{k^{3/2} d^{1/2} + kd}{n^2}\right) \qquad \max_{T \in \binom{[n]}{k}} \left\{\left\|\hat{\boldsymbol{\theta}}_T^{\text{DRIF}} - \hat{\boldsymbol{\theta}}_T^{\text{RIF}}\right\|_2\right\} = \widetilde{\Theta}\left(\frac{k^2 + kd}{n^2}\right).$$

## F.2. Bounding the DRIF Error for Average-Case Sample Drops

In this section we prove the bound $\mathbb{E}_{T \in \binom{[n]}{k}}\left[\left\|\hat{\boldsymbol{\theta}}_T^{\text{DRIF}} - \hat{\boldsymbol{\theta}}_T^{\text{NS}}\right\|_2\right] = \widetilde{O}\left(\frac{kd}{n^2}\right)$ from Theorem B.4.

We will bound the approximation error of DRIF compared to NS by splitting it into 2 contributions

$$\hat{\boldsymbol{\theta}}_T^{\text{DRIF}} - \hat{\boldsymbol{\theta}}_T^{\text{NS}} = \sum_{i \in T} \left(\left(\frac{n-k}{n}\mathbf{H}^{\backslash\{i\}}\right)^{-1} - \left(\mathbf{H}^{\backslash T}\right)^{-1}\right)\mathbf{g}_i \tag{17}$$

$$= \frac{n}{n-k}\left(\mathbf{H}^{\backslash T}\right)^{-1}\sum_{i \in T}\left(p\mathbf{H}^{\backslash\{i\}} - \mathbf{H}^{T\backslash\{i\}}\right)\left(\mathbf{H}^{\backslash\{i\}}\right)^{-1}\mathbf{g}_i \tag{18}$$

$$= \underbrace{\frac{n}{n-k}\left(\mathbf{H}^{\backslash T}\right)^{-1}\sum_{i \in T}\left(p\mathbf{H}^{\backslash\{i\}} - \mathbf{H}^{T\backslash\{i\}}\right)\left(\mathbf{H}^{[n]}\right)^{-1}\mathbf{g}_i +}_{\text{Main Term}} \tag{19}$$

$$+ \underbrace{\frac{n}{n-k}\left(\mathbf{H}^{\backslash T}\right)^{-1}\sum_{i \in T}\frac{L_i}{1-L_i}\left(p\mathbf{H}^{\backslash\{i\}} - \mathbf{H}^{T\backslash\{i\}}\right)\left(\mathbf{H}^{[n]}\right)^{-1}\mathbf{g}_i,}_{\text{Higher Order Correction}} \tag{20}$$

where $p = \frac{k}{n}$ and $L_i = \beta_i \mathbf{x}_i^{\mathsf{T}}\left(\mathbf{H}^{[n]}\right)^{-1}\mathbf{x}_i$ is the $i$th leverage before any samples were dropped.

The key structure we try to preserve here is that both the expectation over $i$ contributions and the expectation over $j$ contributions in the main term, independently have mean $\mathbf{0}$ if the set $T$ is selected at random (this is because each $j$ contribution has a factor of $\mathbb{1}_{j \in T} - p$, which is a mean $0$ scalar, and because the $i$ contributions are a sum over a random subset of the gradients at $\hat{\boldsymbol{\theta}}$ and the sum over all gradients at $\hat{\boldsymbol{\theta}}$ is $\mathbf{0}$ by definition).

**Bounding the Higher Order Term**   The higher order correction can be naively bounded by

$$\|\text{Higher Order Correction}\|_2 = \widetilde{O}\left(\frac{d}{n} \times \frac{\widetilde{O}(k+d)}{n^2} \times k\sqrt{d}\right) = \widetilde{O}\left(\frac{kd\sqrt{d}(k+d)}{n^3}\right) = o\left(\frac{kd}{n^2}\right),$$

so long as $n \gg (k+d)\sqrt{d}$.

### F.2.1. BOUNDING THE MAIN TERM

Focusing on the main term, and opening the definitions of the Hessian and gradient of a logistic regression, we have

$$\|\text{Main Term}\|_2 = \frac{n}{n-k}\left\|\left(\mathbf{H}^{\backslash T}\right)^{-1}\sum_{i \neq j}\alpha_i\beta_j\mathbb{1}_{i \in T}\left(\mathbb{1}_{j \in T} - p\right)\mathbf{x}_j\mathbf{x}_j^{\mathsf{T}}\left(\mathbf{H}^{[n]}\right)^{-1}\mathbf{x}_i\right\|_2 \tag{21}$$

$$\leq \Theta\left(\frac{1}{n}\right)\left\|\sum_{i \neq j}\alpha_i\beta_j\mathbb{1}_{i \in T}\left(\mathbb{1}_{j \in T} - p\right)\mathbf{x}_j\mathbf{x}_j^{\mathsf{T}}\left(\mathbf{H}^{[n]}\right)^{-1}\mathbf{x}_i\right\|_2. \tag{22}$$

We will analyze this term by combining a decoupling argument and a Poissonization argument.

**Poissonization Step**   Our goal is to prove a bound of the form $\mathbb{E}_{T \in \binom{[n]}{k}}\left[\left\|\hat{\boldsymbol{\theta}}_T^{\text{NS}} - \hat{\boldsymbol{\theta}}_T^{\text{DRIF}}\right\|_2\right] \leq \tau$. To prove such a bound, we first note that with very high probability over the samples, this difference is polynomially bounded for all $T$ (e.g., using

the triangle inequality to show that both the NS and DRIF steps are bounded). Therefore, it suffices to show a bound of the form $\Pr_{T \in \binom{[n]}{k}} \left[ \left\| \hat{\boldsymbol{\theta}}_T^{\text{NS}} - \hat{\boldsymbol{\theta}}_T^{\text{DRIF}} \right\|_2 \geq \tau \right] \leq n^{-\omega(1)}$.

We do this using a Poissonization argument: instead of analyzing the case where $T \sim \binom{[n]}{k}$, we analyze the simpler case where $T$ is selected to keep each sample $i$ iid with probability $p = \frac{k}{n}$, and show that our bound holds with very high probability. Such a bound suffices for the case where $T \sim \binom{[n]}{k}$ is drawn only from sets of size $k$, since drawing from the Poissonized model and conditioning on having exactly $k$ samples produces the uniform distribution over $\binom{[n]}{k}$, while only requiring us to condition on an event with probability $1/\text{poly}(n)$, causing the failure probability to increase by at most a polynomial factor.

In other words, fixing $p = \frac{k}{n}$, for any threshold $\tau > 0$

$$\Pr_{T \in \binom{[n]}{k}} \left[ \left\| \hat{\boldsymbol{\theta}}_T^{\text{NS}} - \hat{\boldsymbol{\theta}}_T^{\text{DRIF}} \right\|_2 \geq \tau \right] = \Pr_{T \sim \text{Bern}(p)^{\otimes n}} \left[ \left\| \hat{\boldsymbol{\theta}}_T^{\text{NS}} - \hat{\boldsymbol{\theta}}_T^{\text{DRIF}} \right\|_2 \geq \tau \,\middle|\, |T| = k \right] \leq$$

$$\leq \Pr_{T \sim \text{Bern}(p)^{\otimes n}} \left[ \left\| \hat{\boldsymbol{\theta}}_T^{\text{NS}} - \hat{\boldsymbol{\theta}}_T^{\text{DRIF}} \right\|_2 \geq \tau \right] \times \text{poly}(n) \,.$$

**Decoupling Argument**

$$\sum_{i \neq j} A_{i,j} = 4 \, \mathbb{E}_{S \subseteq [n]} \left[ \sum_{\substack{i \in S \\ j \in \overline{S}}} A_{i,j} \right],$$

where

$$A_{i,j} = \alpha_i \beta_j 1_{i \in T} \left( 1_{j \in T} - p \right) \mathbf{x}_j \mathbf{x}_j^{\mathsf{T}} \left( \mathbf{H}^{[n]} \right)^{-1} \mathbf{x}_i \,,$$

yielding the equality

$$\sum_{i \neq j} \alpha_i \beta_j 1_{i \in T} \left( 1_{j \in T} - p \right) \mathbf{x}_j \mathbf{x}_j^{\mathsf{T}} \left( \mathbf{H}^{[n]} \right)^{-1} \mathbf{x}_i = 4 \, \mathbb{E}_{S \subseteq [n]} \left[ \left( \sum_{j \in \overline{S}} \left( \mathbb{1}_{j \in T} - p \right) \beta_j \mathbf{x}_j \mathbf{x}_j^{\mathsf{T}} \right) \left( \mathbf{H}^{[n]} \right)^{-1} \sum_{i \in S} \mathbf{g}_i \mathbb{1}_{i \in T} \right] \quad (23)$$

The key feature of this decoupling is that we are now trying to analyze the product between a Hessian-like matrix (that has expectation $\mathbf{0}$ if $T$ is random) and the gradient of a set of samples, making for a much simpler object to analyze than a summation over $n^2 - n$ terms.

### F.2.2. PROOF VIA VECTOR BERNSTEIN

We will bound the decoupled summation by viewing the randomness of the process in a specific order. We will show that with very high probability over the randomness of $S$, with very high probability over the randomness of $T \cap S$, with very high probability over $T \cap \overline{S}$, the decoupled summation has bounded $L_2$ norm.

We utilize the randomness of $\mathbb{1}_{T \cap \overline{S}}$ via a vector Bernstein inequality over this randomness. To apply vector Bernstein, we first need to show that each of the vectors in the summation is bounded:

**Lemma F.1** (Each $j$ Element in the Decoupled Summation is Bounded)**.** *For any $j \in \overline{S}$, with very high probability over the randomness of $T \cap S$ and over the samples,*

$$\left\| \mathbf{x}_j \mathbf{x}_j^{\mathsf{T}} \left( \mathbf{H}^{[n]} \right)^{-1} \sum_{i \in S} \mathbf{g}_i \mathbb{1}_{i \in T} \right\|_2 = \widetilde{O} \left( \frac{\sqrt{k} d}{n} \right)$$

We then show that the covariance of these contributions is also bounded:

**Lemma F.2** (Trace of Total Covariance is Bounded)**.** *With very high probability over the samples and over $T \cap S$, the trace of the covariance of the decoupled summation wrt $T \cap \overline{S}$ is at most*

$$\text{tr} \left[ \sum_{j \in \overline{S}} \beta_j^2 p(1-p) \mathbf{x}_j \mathbf{x}_j^{\mathsf{T}} \left\langle \mathbf{x}_j, \left( \mathbf{H}^{[n]} \right)^{-1} \sum_{i \in S} \mathbf{g}_i \mathbb{1}_{i \in T} \right\rangle^2 \right] = \widetilde{O} \left( \frac{k^2 d^2}{n^2} \right)$$

Combining Lemmas F.1 and F.2 with vector Bernstein inequality yields that wvhp

$$\|\text{Main Term}\|_2 = \Theta\left(\frac{1}{n}\right) \times \left\|\sum_{i \neq j} \alpha_i \beta_j \mathbb{1}_{i \in T} \left(\mathbb{1}_{j \in T} - p\right) \mathbf{x}_j \mathbf{x}_j^\intercal\right\|_2 = \widetilde{O}\left(\frac{kd}{n^2}\right),$$

proving $\mathbb{E}_{T \in \binom{[n]}{k}}\left[\left\|\hat{\boldsymbol{\theta}}_T^{\text{DRIF}} - \hat{\boldsymbol{\theta}}_T^{\text{NS}}\right\|_2\right] = \widetilde{O}\left(\frac{kd}{n^2}\right)$ from Theorem B.4. It remains to prove Lemmas F.1 and F.2.

**Proof of Lemma F.1**   The main step in proving Lemma F.1, is to prove that

**Lemma F.3** (Bounded Sample-Gradient Inner Product). *With very high probability over the randomness of the samples and* $T \cap S$, *for any* $j \in \overline{S}$, *we have*

$$\left|\mathbf{x}_j^\intercal \left(\mathbf{H}^{[n]}\right)^{-1} \sum_{i \in S} \mathbf{g}_i \mathbb{1}_{i \in T}\right| = \widetilde{O}\left(\frac{\sqrt{k}d}{n}\right).$$

*Lemma F.3 Implies Lemma F.1.*   By definition

$$\left\|\mathbf{x}_j \mathbf{x}_j^\intercal \left(\mathbf{H}^{[n]}\right)^{-1} \sum_{i \in S} \mathbf{g}_i \mathbb{1}_{i \in T}\right\|_2 = \|\mathbf{x}_j\|_2 \times \left|\mathbf{x}_j^\intercal \left(\mathbf{H}^{[n]}\right)^{-1} \sum_{i \in S} \mathbf{g}_i \mathbb{1}_{i \in T}\right| = \widetilde{O}\left(\sqrt{d}\right) \times \widetilde{O}\left(\frac{\sqrt{k}d}{n}\right),$$

where the last step utilized Lemma H.3 and Lemma F.3, yielding the lemma.

$\square$

**Proof of Lemma F.3**

*Proof of Lemma F.3.*   With very high probability $|T \cap S| = \widetilde{O}(k)$, and for a given size it has equal probability of selecting any subset of the samples of this size. So from Lemma D.1, wvhp $\left\|\mathbf{g}^{T \cap S}\right\|_2 = \widetilde{O}\left(\sqrt{kd}\right)$.

Denote

$$\mathbf{u} = \left(\mathbf{H}^{[n]}\right)^{-1} \mathbf{g}^{T \cap S}.$$

Our goal is to bound $\langle \mathbf{x}_j, \mathbf{u} \rangle$, and the key difficulty in getting a good bound is that $\mathbf{u}$ depends on the model $\hat{\boldsymbol{\theta}}$, which in turn depends on $\mathbf{x}_j$. Our strategy will be to first bound the inner product after removing this dependency and then bound the effect of $\mathbf{x}_j$ on $\mathbf{u}$.

In particular, we consider the model $\hat{\boldsymbol{\theta}}_{\backslash\{j\}}$ trained without this sample. Consider $\mathbf{H}_{\hat{\boldsymbol{\theta}}_{\backslash\{j\}}}^{\backslash\{j\}}$ – the Hessian of this model on all the samples except the $j$th sample. We can view the analogous version of the vector $\mathbf{u}$ for this new model

$$\mathbf{u}_{\backslash\{j\}} = \left(\mathbf{H}_{\hat{\boldsymbol{\theta}}_{\backslash\{j\}}}^{\backslash\{j\}}\right)^{-1} \mathbf{g}_{\hat{\boldsymbol{\theta}}_{\backslash\{j\}}}^{T \cap S}.$$

$\mathbf{x}_j$ is independent of $\mathbf{u}_{\backslash\{j\}}$, so $\langle \mathbf{u}_{\backslash\{j\}}, \mathbf{x}_j \rangle$ is normally distributed with mean 0 and variance $\left\|\mathbf{u}_{\backslash\{j\}}\right\|_2^2$. Therefore, with very high probability

$$\left|\langle \mathbf{u}_{\backslash\{j\}}, \mathbf{x}_j \rangle\right| = \widetilde{O}\left(\left\|\mathbf{u}_{\backslash\{j\}}\right\|_2\right) \leq \widetilde{O}\left(\left\|\left(\mathbf{H}_{\hat{\boldsymbol{\theta}}_{\backslash\{j\}}}^{\backslash\{j\}}\right)^{-1}\right\|_{\text{op}} \times \left\|\mathbf{g}_{\hat{\boldsymbol{\theta}}_{\backslash\{j\}}}^{T \cap S}\right\|_2\right) = \widetilde{O}\left(\frac{\sqrt{k}d}{n}\right),$$

where the last step, utilized Lemma D.1 (which tells us that $\left\|\mathbf{g}_{\hat{\boldsymbol{\theta}}_{\backslash\{j\}}}^{T \cap S}\right\|_2 = \widetilde{O}\left(\sqrt{kd}\right)$ wvhp) and Lemma H.11 (which tells us that the Hessian inverse contributed a $\frac{1}{n}$ factor).

Next, we define $\mathbf{u}^j = \left(\mathbf{H}^{\backslash\{j\}}\right)^{-1} \mathbf{g}^{T \cap S}$. Bounding $\langle \mathbf{x}_j, \mathbf{u}^j \rangle$ yields a bound on $\langle \mathbf{x}_j, \mathbf{u} \rangle$, since from the Sherman-Morrison formula,

$$\langle \mathbf{x}_j, \mathbf{u}^j \rangle = \frac{1}{1 - L_j} \langle \mathbf{x}_j, \mathbf{u} \rangle \Rightarrow |\langle \mathbf{x}_j, \mathbf{u} \rangle| \leq \left|\langle \mathbf{x}_j, \mathbf{u}^j \rangle\right|.$$

To bound $\langle \mathbf{x}_j, \mathbf{u}^j \rangle$, we will bound $\left\| \mathbf{u}^j - \mathbf{u}_{\backslash\{j\}} \right\|_2$ and apply the CS inequality to bound its inner product with $\mathbf{x}_j$. From Theorem B.3, we know that with very high probability

$$\hat{\boldsymbol{\theta}}_{\backslash\{j\}} = \hat{\boldsymbol{\theta}}^{\text{NS}}_{\backslash\{j\}} \pm \widetilde{O}\left(\frac{d}{n^2}\right) = \hat{\boldsymbol{\theta}} \pm \widetilde{O}\left(\frac{\sqrt{d}}{n}\right).$$

Therefore, utilizing our bound on the operator norm of the third order derivative of the loss (Lemma H.13), we know that wvhp over the randomness of the samples,

$$\left\| \left(\mathbf{H}^{\backslash\{j\}}_{\hat{\boldsymbol{\theta}}_{\backslash\{j\}}}\right)^{-1} - \left(\mathbf{H}^{\backslash\{j\}}_{\hat{\boldsymbol{\theta}}}\right)^{-1} \right\|_{\text{op}} = \widetilde{O}\left(\frac{1}{n^2} \times n \times \frac{\sqrt{d}}{n}\right) = \widetilde{O}\left(\frac{\sqrt{d}}{n^2}\right),$$

and that

$$\left\| \mathbf{g}^{T \cap S}_{\hat{\boldsymbol{\theta}}_{\backslash\{j\}}} - \mathbf{g}^{T \cap S}_{\hat{\boldsymbol{\theta}}_{\backslash\{j\}}} \right\| = \widetilde{O}\left(\frac{(k+d)\sqrt{d}}{n}\right).$$

Therefore,

$$\left\| \mathbf{u}^j - \mathbf{u}_{\backslash\{j\}} \right\|_2 \leq \left\| \left(\mathbf{H}^{\backslash\{j\}}_{\hat{\boldsymbol{\theta}}_{\backslash\{j\}}}\right)^{-1} \right\|_{\text{op}} \left\| \mathbf{g}^{S \cap T}_{\hat{\boldsymbol{\theta}}_{\backslash\{j\}}} - \mathbf{g}^{S \cap T}_{\hat{\boldsymbol{\theta}}} \right\|_2 + \tag{24}$$

$$+ \left\| \left(\mathbf{H}^{\backslash\{j\}}_{\hat{\boldsymbol{\theta}}_{\backslash\{j\}}}\right)^{-1} \right\|_{\text{op}} \left\| \mathbf{H}^{\backslash\{j\}}_{\hat{\boldsymbol{\theta}}_{\backslash\{j\}}} - \mathbf{H}^{\backslash\{j\}} \right\|_{\text{op}} \left\| \left(\mathbf{H}^{\backslash\{j\}}_{\hat{\boldsymbol{\theta}}}\right)^{-1} \right\|_{\text{op}} \left\| \mathbf{g}^{S \cap T}_{\hat{\boldsymbol{\theta}}} \right\|_2 \tag{25}$$

$$\leq \widetilde{O}\left(\frac{k+d}{n} \times \frac{\sqrt{d}}{n}\right) + \widetilde{O}\left(\frac{\sqrt{kd}}{n^2} \times n \times \frac{\sqrt{d}}{n}\right) = \widetilde{O}\left(\frac{(k+d)\sqrt{d}}{n^2}\right) \ll \widetilde{O}\left(\frac{\sqrt{k}}{n}\right). \tag{26}$$

Yielding

$$\left| \langle \mathbf{x}_j, \mathbf{u}_{\backslash\{j\}} - \mathbf{u}^j \rangle \right| \leq \|\mathbf{x}_j\|_2 \times \left\| \mathbf{u}_{\backslash\{j\}} - \mathbf{u}^j \right\|_2 = \widetilde{O}\left(\sqrt{d} \times \left(\frac{(k+d)\sqrt{d}}{n^2} + \frac{\sqrt{kd}}{n^2}\right)\right) = o\left(\frac{\sqrt{kd}}{n}\right),$$

from the CS inequality. $\qquad\qquad\qquad\qquad\qquad\qquad\qquad\qquad\qquad\qquad\qquad\qquad\qquad\qquad\qquad\qquad\qquad\quad\square$

**Proof of Lemma F.2**   We view

$$\sum_{j \in \overline{S}} \left(\mathbb{1}_{j \in T} - p\right) \mathbf{x}_j \left(\beta_j \langle \mathbf{x}_j, \mathbf{u} \rangle\right),$$

as the sum over $\leq n$ independent random variables, whose randomness we view as being a function of the randomness of $T \cap \overline{S}$ (note that $\mathbf{u}$ depends only on the randomness of $T \cap S$, which is independent of $T \cap \overline{S}$ due to the Poissonization step). From Lemma F.3, we know that with very high probability over the randomness of the samples and of $T \cap S$, $|\langle \mathbf{x}_j, \mathbf{u} \rangle| = \widetilde{O}\left(\frac{\sqrt{kd}}{n}\right)$ for all $j \in \overline{S}$, and this does not depend on the randomness of $T \cap \overline{S}$.

The sum over the covariance of the summands is given by

$$p(1-p) \sum_{j \in \overline{S}} \mathbf{x}_j \mathbf{x}_j^{\mathsf{T}} \left(\beta_j \langle \mathbf{x}_j, \mathbf{u} \rangle\right)^2 \preceq \widetilde{O}\left(p \times \frac{\sqrt{kd}}{n}\right) \times \mathbf{H}^{\overline{S}}_{\hat{\boldsymbol{\theta}}}.$$

Therefore, taking the trace, we get

$$\text{tr}\left[ p(1-p) \sum_{j \in \overline{S}} \mathbf{x}_j^{\mathsf{T}} \mathbf{x}_j \left(\beta_j \langle \mathbf{x}_j, \mathbf{u} \rangle\right)^2 \right] = \widetilde{O}\left(\frac{k}{n} \times \frac{kd}{n^2} \times n \times d\right) = \widetilde{O}\left(\frac{k^2 d^2}{n^2}\right).$$

**Concluding a bound on the norm of the decoupled summation**  Recall that each of the summands has mean $\mathbf{0}$, from our assumption that $\mathbb{1}_{j \in T}$ is an iid Bernoulli.

Therefore, applying the vector Bernstein inequality, we know that with very high probability

$$\left\| \sum_{j \in \overline{S}} \left(1_{j \in T} - p\right) \mathbf{x}_j \left(\beta_j \langle \mathbf{x}_j, \mathbf{u} \rangle\right) \right\|_2 = \widetilde{O}\left(\frac{kd}{n}\right),$$

as desired.

Therefore, the main term, which scales like $\frac{1}{n}$ times the norm of this summation is of norm at most $\widetilde{O}\left(\frac{kd}{n^2}\right)$, proving the average-case upper bound of Theorem B.4.

### F.3. Bounding the DRIF Error for Worst-Case Sample Drops

We now turn to bounding the DRIF error for worst-case sample drops. We will utilize the identity that

$$\hat{\boldsymbol{\theta}}_T^{\text{DRIF}} - \hat{\boldsymbol{\theta}}_T^{\text{NS}} = \sum_{i \in T} \left( \left( \frac{n-k}{n} \mathbf{H}^{\backslash\{i\}} \right)^{-1} - \left( \mathbf{H}^{\backslash T} \right)^{-1} \right) \mathbf{g}_i \tag{27}$$

$$= \frac{n}{n-k} \left( \mathbf{H}^{\backslash T} \right)^{-1} \sum_{i \in T} \left( p\mathbf{H}^{\backslash\{i\}} - \mathbf{H}^{T \backslash \{i\}} \right) \left( \mathbf{H}^{\backslash\{i\}} \right)^{-1} \mathbf{g}_i \tag{28}$$

$$= \underbrace{\frac{n}{n-k} \left( \mathbf{H}^{\backslash T} \right)^{-1} \sum_{i \in T} \left( p\mathbf{H}^{\backslash\{i\}} - \mathbf{H}^{T \backslash \{i\}} \right) \left( \mathbf{H}^{[n]} \right)^{-1} \mathbf{g}_i}_{\text{Main Term}} + \tag{29}$$

$$+ \underbrace{\frac{n}{n-k} \left( \mathbf{H}^{\backslash T} \right)^{-1} \sum_{i \in T} \frac{L_i}{1 - L_i} \left( p\mathbf{H}^{\backslash\{i\}} - \mathbf{H}^{T \backslash \{i\}} \right) \left( \mathbf{H}^{[n]} \right)^{-1} \mathbf{g}_i}_{\text{Higher Order Correction}}, \tag{30}$$

where $L_i = \beta_i \mathbf{x}_i^{\mathsf{T}} \left( \mathbf{H}^{[n]} \right)^{-1} \mathbf{x}_i$ is the $i$th leverage.

As before, we note that wvhp over the samples, for all $T \in \binom{[n]}{k}$, the higher order term is bounded due to submultiplicativity / triangle inequality

$$\|\text{Higher Order Correction}\|_2 = \widetilde{O}\left( \frac{d}{n} \times \frac{\widetilde{O}(k+d)}{n^2} \times k\sqrt{d} \right) = \widetilde{O}\left( \frac{kd\sqrt{d}(k+d)}{n^3} \right) = o\left( \frac{kd}{n^2} \right),$$

allowing us to focus on the main term.

We simplify the main term by using a decoupling argument

$$\text{Main Term} = \left( \mathbf{H}^{\backslash T} \right)^{-1} \sum_{i \in T} \left( p\mathbf{H}^{\backslash\{i\}} - \mathbf{H}^{T \backslash \{i\}} \right) \left( \mathbf{H}^{[n]} \right)^{-1} \mathbf{g}_i = \tag{31}$$

$$= \left( \mathbf{H}^{\backslash T} \right)^{-1} \sum_{\substack{i,j \in T \\ i \neq j}} \left( \mathbb{1}_{j \in T} - p \right) \beta_j \mathbf{x}_j \mathbf{x}_j^{\mathsf{T}} \left( \mathbf{H}^{[n]} \right)^{-1} \mathbf{x}_i \alpha_i = \tag{32}$$

$$= 4 \left( \mathbf{H}^{\backslash T} \right)^{-1} \mathop{\mathbb{E}}_{S \subseteq [n]} \left[ \sum_{\substack{i \in T \cap S \\ j \in \overline{S}}} \left( \mathbb{1}_{j \in T} - p \right) \beta_j \mathbf{x}_j \mathbf{x}_j^{\mathsf{T}} \left( \mathbf{H}^{[n]} \right)^{-1} \mathbf{x}_i \alpha_i \right] \tag{33}$$

From Lemma H.11, we know that wvhp over the samples, for all $T \in \binom{[n]}{k}$,

$$\left\| \left( \mathbf{H}^{\backslash T} \right)^{-1} \right\|_{\text{op}} = O\left( \frac{1}{n} \right).$$

Therefore, it suffices to bound the norm of

$$
\left\| \mathop{\mathbb{E}}_{S \subseteq [n]} \left[ \sum_{\substack{i \in T \cap S \\ j \in \overline{S}}} \left( \mathbb{1}_{j \in T} - p \right) \beta_j \mathbf{x}_j \mathbf{x}_j^\intercal \left( \mathbf{H}^{[n]} \right)^{-1} \mathbf{x}_i \alpha_i \right] \right\|_2 \leq
$$

$$
\leq \left\| \mathop{\mathbb{E}}_{S \subseteq [n]} \left[ \sum_{\substack{i \in T \cap S \\ j \in \overline{S}}} \mathbb{1}_{j \in T} \beta_j \mathbf{x}_j \mathbf{x}_j^\intercal \left( \mathbf{H}^{[n]} \right)^{-1} \mathbf{x}_i \alpha_i \right] \right\|_2 + \left\| \mathop{\mathbb{E}}_{S \subseteq [n]} \left[ \sum_{\substack{i \in T \cap S \\ j \in \overline{S}}} p \beta_j \mathbf{x}_j \mathbf{x}_j^\intercal \left( \mathbf{H}^{[n]} \right)^{-1} \mathbf{x}_i \alpha_i \right] \right\|_2 \leq
$$

$$
\leq 2 \max_{\substack{T, T' \in \binom{[n]}{\leq k} \\ T \cap T' = \emptyset}} \left\{ \left\| \sum_{\substack{i \in T \\ j \in T'}} \beta_j \mathbf{x}_j \mathbf{x}_j^\intercal \left( \mathbf{H}^{[n]} \right)^{-1} \mathbf{x}_i \alpha_i \right\|_2 \right\}
$$

Therefore, to complete the proof of Theorem B.4, it will suffice to prove the following claim:

**Lemma F.4.** *Under the assumptions of Theorem B.4, wvhp over the samples*

$$
\max_{\substack{T, T' \in \binom{[n]}{\leq k} \\ T \cap T' = \emptyset}} \left\{ \left\| \sum_{\substack{i \in T \\ j \in T'}} \beta_j \mathbf{x}_j \mathbf{x}_j^\intercal \left( \mathbf{H}^{[n]} \right)^{-1} \mathbf{x}_i \alpha_i \right\|_2 \right\} = \widetilde{O} \left( \frac{kd + k^2}{n} \right) .
$$

### F.3.1. PROOF STRATEGY

Lemma F.4 immediately implies the upper bound on the DRIF error in Theorem B.4. Lemma F.4 is relatively easy to prove for the case where $k \geq d$, since combining submultiplicativity, with Lemma H.9 (which can be used to show that, wvhp, the Hessian of any $\leq k$ samples is at most $\widetilde{O}\left(k + d\right)$) and equation (9) (which tells us that wvhp the maximal norm of the gradient of a set of $\leq k$ samples is at most $\widetilde{O}\left(\sqrt{kd} + k\right)$), we see that

$$
\left\| \sum_{\substack{i \in T \\ j \in T'}} \beta_j \mathbf{x}_j \mathbf{x}_j^\intercal \left( \mathbf{H}^{[n]} \right)^{-1} \mathbf{x}_i \alpha_i \right\|_2 \leq \left\| \mathbf{H}^T \right\|_{\mathrm{op}} \left\| \left( \mathbf{H}^{[n]} \right)^{-1} \right\|_{\mathrm{op}} \left\| \mathbf{g}^T \right\|_2 = \widetilde{O} \left( \frac{\left( \sqrt{kd} + k \right) \times (k + d)}{n} \right) =
$$

$$
= \widetilde{O} \left( \frac{k^2 + kd^{3/2}}{n} \right) = \widetilde{O} \left( \frac{k^2 + kd}{n} \right) ,
$$

where the last step utilized our assumption that $k \geq d$.

Therefore, it remains to prove Lemma F.4 for the $k < d$ regime. We split this proof into 3 sublemmas.

The first is a very technical concentration bound on the covariance of (sub)-gaussian random variables.

**Lemma F.5.** *Let $\mathbf{u}_1, \ldots, \mathbf{u}_k \sim \mathcal{U}$ be iid vectors drawn from a $1$-subgaussian distribution, and let $\mathbf{v} \in \mathbb{R}^d$ be any fixed vector that does not depend on $\mathbf{u}_1, \ldots, \mathbf{u}_k$. Then, for some constant $C > 0$, and for any $t \geq Ck$, with probability $1 - e^{-\Omega(t)}$,*

$$
\mathbf{v}^\intercal \left( \sum_{i=1}^k \mathbf{u}_i \mathbf{u}_i^\intercal \right) \mathbf{v} \leq t \left\| \mathbf{v} \right\|_2^2 .
$$

We apply Lemma F.5 with $t = \Omega\left(k \log(n)\right)$ large enough to combine it with a union bound on all disjoint pairs $T, T' \in \binom{[n]}{\leq k}$. We will set $\mathbf{u}_i = \mathbf{x}_i$, and want to set $\mathbf{v} \approx \left( \mathbf{H}^{[n]} \right)^{-1} \mathbf{g}^T$. The difficulty is that $\left( \mathbf{H}^{[n]} \right)^{-1} \mathbf{g}^T$ hides an inner dependence on the samples in $T'$ (both because these are included in the Hessian and because the model weights depend on these samples). Another key difficulty is that Lemma F.5 gives a bound on

$$
\text{Lemma F.5 LHS} = \mathbf{v}^\intercal \left( \sum_{i=1}^k \mathbf{u}_i \mathbf{u}_i^\intercal \right) \mathbf{v}
$$

while we need a bound on

$$\text{Target} = \left\|\left(\sum_{i=1}^{k}\mathbf{u}_i\mathbf{u}_i^\mathsf{T}\right)\mathbf{v}\right\|_2 = \sqrt{\mathbf{v}^\mathsf{T}\left(\sum_{i=1}^{k}\mathbf{u}_i\mathbf{u}_i^\mathsf{T}\right)^2\mathbf{v}}\,.$$

We first deal with the dependence of $\mathbf{H}$ and $\mathbf{g}$ on the model's dependence on the samples in $T'$ with the following lemma:

**Lemma F.6.** *Under the assumptions of Theorem B.4, wvhp over the samples,*

$$\left\|\hat{\boldsymbol{\theta}} - \hat{\boldsymbol{\theta}}_{T'}\right\|_2 = \widetilde{O}\left(\frac{\sqrt{kd}+k}{n}\right),$$

*yielding the bounds*

$$\max_{\substack{T,T'\in\binom{[n]}{k}\\T\cap T'=\emptyset}}\left\{\left\|\mathbf{g}_{\hat{\boldsymbol{\theta}}_{T'}}^{T} - \mathbf{g}_{\hat{\boldsymbol{\theta}}}^{T}\right\|_2\right\} = \widetilde{O}\left(\frac{k^2 + k^{1/2}d^{3/2}}{n}\right),$$

*and*

$$\max_{T\in\binom{[n]}{k}}\left\{\left\|\left(\mathbf{H}_{\hat{\boldsymbol{\theta}}_{T'}}^{[n]}\right)^{-1} - \left(\mathbf{H}_{\hat{\boldsymbol{\theta}}}^{[n]}\right)^{-1}\right\|_{\text{op}}\right\} = \widetilde{O}\left(\frac{\sqrt{kd}+k}{n^2}\right).$$

*Combining these with the triangle inequality and submultiplicativity yields*

$$\max_{\substack{T,T'\in\binom{[n]}{k}\\T\cap T'=\emptyset}}\left\{\left\|\mathbf{H}^{T'}\left(\mathbf{H}_{\hat{\boldsymbol{\theta}}}^{[n]}\right)^{-1}\mathbf{g}_{\hat{\boldsymbol{\theta}}}^{T} - \mathbf{H}^{T'}\left(\mathbf{H}_{\hat{\boldsymbol{\theta}}_{T'}}^{[n]}\right)^{-1}\mathbf{g}_{\hat{\boldsymbol{\theta}}_{T'}}^{T}\right\|_2\right\} = \widetilde{O}\left(\frac{k^3 + k^{1/2}d^{5/2}}{n^2}\right) = \widetilde{O}\left(\frac{k^2 + kd}{n}\right).$$

Finally, we deal with the dependence of $\left(\mathbf{H}^{[n]}\right)^{-1}\mathbf{g}^{T}$ on the inclusion of samples in $T'$ in the Hessian $\mathbf{H}^{[n]}$ and on the difference in targets. The following lemma bounds our value of interest if the main Hessian and the gradients were evaluated on the model optimized without the set of samples in $T'$:

**Lemma F.7.** *Under the assumptions of Theorem B.4, wvhp over the samples,*

$$\max_{\substack{T,T'\in\binom{[n]}{k}\\T\cap T'=\emptyset}}\left\{\left\|\mathbf{H}^{T'}\left(\mathbf{H}_{\hat{\boldsymbol{\theta}}_{T'}}^{[n]}\right)^{-1}\mathbf{g}_{\hat{\boldsymbol{\theta}}_{T'}}^{T}\right\|_2\right\} = \widetilde{O}\left(\frac{kd + k^2}{n}\right).$$

### F.3.2. Proof of Lemma F.5

Lemma F.5 follows almost immediately from the following bound on the tail of the distribution of the sum of subexponential random variables:

**Lemma F.8** (Corollary 2.9.2 of (Vershynin, 2018)). *Let $X_1,\ldots,X_N$ be independent, mean-zero, subexponential random variables, and let $a = (a_1,\ldots,a_N)\in\mathbb{R}^N$. Then, for every $t\geq 0$, we have*

$$\Pr\left[\left|\sum_{i=1}^{N}a_iX_i\right| \geq t\right] \leq 2\exp\left[-c\min\left(\frac{t^2}{K^2\|a\|_2^2}, \frac{t}{K\|a\|_\infty}\right)\right],$$

*where $c > 0$ is an absolute constant and $K \overset{\text{def}}{=} \max_i \|X_i\|_{\psi_1}$.*

*Lemma F.8 Implies Lemma F.5.* Define

$$z_i = \frac{\langle\mathbf{u}_i,\mathbf{v}\rangle^2}{\|\mathbf{v}\|_2^2}\,.$$

From our assumption that $\mathbf{v}$ is fixed and independent of $\mathbf{u}_i$, and that the $\mathbf{u}_i$ are iid subgaussian random variables, the $z_i$ are independent subexponential random variables, with mean

$$\mathbb{E}\left[z_i\right] = \frac{1}{\|\mathbf{v}\|_2^2}\mathbf{v}^\mathsf{T}\mathbb{E}\left[\mathbf{u}_i\mathbf{u}_i^\mathsf{T}\right]\mathbf{v} = O(1)\,.$$

Therefore, from Lemma F.8,

$$\Pr\left[\mathbf{v}^\intercal \left(\sum_{i=1}^{k} \mathbf{u}_i \mathbf{u}_i^\intercal\right) \mathbf{v} > t\,\|\mathbf{v}\|_2^2\right] = \exp\left(-\Omega\left(t\right)\right).$$

$\square$

### F.3.3. Proof of Lemma F.6

*Proof of Lemma F.6.* The first claim of Lemma F.6 follows from a combination of Lemma H.11 and equation (9) (which tell us that the Hessian inverse and gradient of the samples $[n] \setminus T'$ at $\hat{\boldsymbol{\theta}}$ are both bounded wvhp, implying that the Newton step approximation to dropping $T'$ is short), and Theorem B.3 (which tells us that wvhp over the samples, $\hat{\boldsymbol{\theta}}_{T'}$ is close to $\hat{\boldsymbol{\theta}}_{T'}^{\mathrm{NS}}$) to yield the bound

$$\left\|\hat{\boldsymbol{\theta}} - \hat{\boldsymbol{\theta}}_{T'}\right\|_2 \le \left\|\hat{\boldsymbol{\theta}} - \hat{\boldsymbol{\theta}}_{T'}^{\mathrm{NS}}\right\|_2 + \left\|\hat{\boldsymbol{\theta}}_{T'}^{\mathrm{NS}} - \hat{\boldsymbol{\theta}}_{T'}\right\|_2 \le \widetilde{O}\left(\frac{\sqrt{kd}+k}{n}\right) + \widetilde{O}\left(\frac{kd+k^2}{n^2}\right) = \widetilde{O}\left(\frac{\sqrt{kd}+k}{n}\right).$$

To conclude the bound on $\left\|\mathbf{g}_{\hat{\boldsymbol{\theta}}}^T - \mathbf{g}_{\hat{\boldsymbol{\theta}}_{T'}}^T\right\|_2$, we will use a global bound on the spectrum of $\mathbf{H}_{\boldsymbol{\theta}}^T$ implied by Lemma H.9. Let $\overline{T} = T \cup [\widetilde{O}\left(k+d\right)]$ be an extension of $T$ that is large enough for us to apply Lemma H.9. We have

$$\forall \|\boldsymbol{\theta}\|_2 \le C \quad \mathbf{0} \preceq \mathbf{H}_{\boldsymbol{\theta}}^T \preceq \mathbf{H}_{\boldsymbol{\theta}}^{\overline{T}} \Rightarrow \left\|\mathbf{H}_{\boldsymbol{\theta}}^T\right\|_{\mathrm{op}} \le \left\|\mathbf{H}_{\boldsymbol{\theta}}^{\overline{T}}\right\|_{\mathrm{op}} = \widetilde{O}\left(k+d\right).$$

Therefore

$$\left\|\mathbf{g}_{\hat{\boldsymbol{\theta}}_{T'}}^T - \mathbf{g}_{\hat{\boldsymbol{\theta}}}^T\right\|_2 \le \left\|\hat{\boldsymbol{\theta}} - \hat{\boldsymbol{\theta}}_{T'}\right\|_2 \times \max_{\boldsymbol{\theta} \in [\hat{\boldsymbol{\theta}}, \hat{\boldsymbol{\theta}}_{T'}]}\left\{\left\|\mathbf{H}_{\boldsymbol{\theta}}^T\right\|_{\mathrm{op}}\right\} = \widetilde{O}\left(\frac{k^2 + k^{1/2}d^{3/2}}{n}\right).$$

Finally, from Lemma H.13, we know that wvhp over the training set, the third order moment of the set of all samples is bounded

$$\forall \|\boldsymbol{\theta}\|_2 \le C \quad \left\|\mathbf{T}_{\boldsymbol{\theta}}^{[n]}\right\|_{\mathrm{op}} = \widetilde{O}\left(n + k^{3/2} + d^{3/2}\right) = \widetilde{O}\left(n\right),$$

where the last step utilized our assumption that $n \ge k^{3/2} + d^{3/2}$.

Therefore, we may bound the change in the Hessians by

$$\left\|\left(\mathbf{H}_{\hat{\boldsymbol{\theta}}_{T'}}^{[n]}\right) - \left(\mathbf{H}_{\hat{\boldsymbol{\theta}}}^{[n]}\right)\right\|_{\mathrm{op}} \le \left\|\hat{\boldsymbol{\theta}} - \hat{\boldsymbol{\theta}}_{T'}\right\|_2 \times \max_{\boldsymbol{\theta} \in [\hat{\boldsymbol{\theta}}, \hat{\boldsymbol{\theta}}_{T'}]}\left\{\left\|\mathbf{T}_{\boldsymbol{\theta}}^{[n]}\right\|_{\mathrm{op}}\right\} = \widetilde{O}\left(\sqrt{kd}+k\right).$$

Therefore, we may conclude the next claim of Lemma F.6 by submultiplicativity

$$\left\|\left(\mathbf{H}_{\hat{\boldsymbol{\theta}}_{T'}}^{[n]}\right)^{-1} - \left(\mathbf{H}_{\hat{\boldsymbol{\theta}}}^{[n]}\right)^{-1}\right\|_{\mathrm{op}} = \left\|\left(\mathbf{H}_{\hat{\boldsymbol{\theta}}_{T'}}^{[n]}\right)^{-1}\left(\left(\mathbf{H}_{\hat{\boldsymbol{\theta}}_{T'}}^{[n]}\right) - \left(\mathbf{H}_{\hat{\boldsymbol{\theta}}}^{[n]}\right)\right)\left(\mathbf{H}_{\hat{\boldsymbol{\theta}}}^{[n]}\right)^{-1}\right\|_{\mathrm{op}} \le$$

$$\le \left\|\left(\mathbf{H}_{\hat{\boldsymbol{\theta}}_{T'}}^{[n]}\right)^{-1}\right\|_{\mathrm{op}} \times \left\|\left(\left(\mathbf{H}_{\hat{\boldsymbol{\theta}}_{T'}}^{[n]}\right) - \left(\mathbf{H}_{\hat{\boldsymbol{\theta}}}^{[n]}\right)\right)\right\|_{\mathrm{op}} \times \left\|\left(\mathbf{H}_{\hat{\boldsymbol{\theta}}}^{[n]}\right)^{-1}\right\|_{\mathrm{op}} =$$

$$= \widetilde{O}\left(\frac{\sqrt{kd}+k}{n^2}\right).$$

Combining these bounds with the triangle inequality and submultiplicativity we may conclude that

$$\left\|\left(\mathbf{H}_{\hat{\boldsymbol{\theta}}}^{[n]}\right)^{-1}\mathbf{g}_{\hat{\boldsymbol{\theta}}}^T - \left(\mathbf{H}_{\hat{\boldsymbol{\theta}}_{T'}}^{[n]}\right)^{-1}\mathbf{g}_{\hat{\boldsymbol{\theta}}_{T'}}^T\right\|_2 \le \left\|\left(\left(\mathbf{H}_{\hat{\boldsymbol{\theta}}}^{[n]}\right)^{-1} - \left(\mathbf{H}_{\hat{\boldsymbol{\theta}}_{T'}}^{[n]}\right)^{-1}\right)\mathbf{g}_{\hat{\boldsymbol{\theta}}}^T\right\|_2 + \left\|\left(\mathbf{H}_{\hat{\boldsymbol{\theta}}_{T'}}^{[n]}\right)^{-1}\left(\mathbf{g}_{\hat{\boldsymbol{\theta}}}^T - \mathbf{g}_{\hat{\boldsymbol{\theta}}_{T'}}^T\right)\right\|_2 \le$$

$$\le \left\|\left(\mathbf{H}_{\hat{\boldsymbol{\theta}}}^{[n]}\right)^{-1} - \left(\mathbf{H}_{\hat{\boldsymbol{\theta}}_{T'}}^{[n]}\right)^{-1}\right\|_{\mathrm{op}}\left\|\mathbf{g}_{\hat{\boldsymbol{\theta}}}^T\right\|_2 + \left\|\left(\mathbf{H}_{\hat{\boldsymbol{\theta}}_{T'}}^{[n]}\right)^{-1}\right\|_{\mathrm{op}}\left\|\mathbf{g}_{\hat{\boldsymbol{\theta}}}^T - \mathbf{g}_{\hat{\boldsymbol{\theta}}_{T'}}^T\right\|_2 =$$

$$= \widetilde{O}\left(\frac{kd+k^2}{n^2} + \frac{1}{n} \times \frac{k^2 + k^{1/2}d^{3/2}}{n}\right) = \widetilde{O}\left(\frac{k^2 + k^{1/2}d^{3/2}}{n^2}\right).$$

Applying submultiplicativity again with the inequality that

$$\left\|\mathbf{H}^{T'}\right\|_{\mathrm{op}} \leq \left\|\mathbf{H}^{\overline{T'}}\right\|_{\mathrm{op}} = \widetilde{O}\left(k+d\right),$$

yields the desired result

$$\left\|\mathbf{H}^{T'}\left(\mathbf{H}_{\hat{\boldsymbol{\theta}}}^{[n]}\right)^{-1}\mathbf{g}_{\hat{\boldsymbol{\theta}}}^{T} - \left(\mathbf{H}_{\hat{\boldsymbol{\theta}}_{T'}}^{[n]}\right)^{-1}\mathbf{g}_{\hat{\boldsymbol{\theta}}_{T'}}^{T}\right\|_2 = \widetilde{O}\left(\frac{k^3 + k^{1/2}d^{5/2}}{n^2}\right) = \widetilde{O}\left(\frac{k^2 + kd}{n}\right),$$

where the last step also used our assumption that $n \geq d^{3/2}$.

$\square$

### F.3.4. PROOF OF LEMMA F.7

*Proof of Lemma F.7.* Recall that Lemma F.6 gave us a bound on

$$\left\|\mathbf{H}^{T'}\left(\mathbf{H}_{\hat{\boldsymbol{\theta}}_{T'}}^{[n]}\right)^{-1}\mathbf{g}_{\hat{\boldsymbol{\theta}}_{T'}}^{T} - \mathbf{H}^{T'}\left(\mathbf{H}_{\hat{\boldsymbol{\theta}}}^{[n]}\right)^{-1}\mathbf{g}_{\hat{\boldsymbol{\theta}}}^{T}\right\|_2.$$

Therefore, to conclude our desired bound on the norm of $\mathbf{H}^{T'}\left(\mathbf{H}_{\hat{\boldsymbol{\theta}}}^{[n]}\right)^{-1}\mathbf{g}_{\hat{\boldsymbol{\theta}}}^{T}$, it remains to show that Lemma F.5 also yields a bound on the norm of $\mathbf{H}^{T'}\left(\mathbf{H}_{\hat{\boldsymbol{\theta}}_{T'}}^{[n]}\right)^{-1}\mathbf{g}_{\hat{\boldsymbol{\theta}}_{T'}}^{T}$.

Lemma F.5 states that for any fixed $\mathbf{v}$ that does not depend on the samples in the set $T'$, we know that for any fixed $\mathbf{v}, T'$, with probability $1 - n^{-\Omega(k)}$,

$$\left\|\left(\sqrt{\beta_i}\langle\mathbf{x}_i, \mathbf{v}\rangle\right)_{i\in T'}\right\|_2^2 \leq \left\|\left(\langle\mathbf{x}_i, \mathbf{v}\rangle\right)_{i\in T'}\right\|_2^2 = \mathbf{v}^{\mathsf{T}}\left(\sum_{i\in T'}\mathbf{x}_i\mathbf{x}_i^{\mathsf{T}}\right)\mathbf{v} \leq \widetilde{O}\left(k \times \|\mathbf{v}\|_2^2\right).$$

We will apply this bound to

$$\mathbf{v}_{T,T'} \stackrel{\text{def}}{=} \left(\mathbf{H}_{\hat{\boldsymbol{\theta}}_{T'}}^{[n]\backslash T'}\right)^{-1}\mathbf{g}_{\hat{\boldsymbol{\theta}}_{T'}}^{T}.$$

By its definition $\mathbf{v}_{T,T'}$ does not depend on the samples in $T'$, and we will take a union bound over the $n^{O(k)}$ pairs of disjoint $T, T' \in \binom{[n]}{k}$.

To use this bound for our setting, we also need to utilize the Woodbury matrix identity on $\mathbf{H}^{[n]} = \mathbf{H}^{[n]\backslash T'} + \mathbf{X}_{T'}\operatorname{diag}\left(\boldsymbol{\beta}^{T'}\right)\mathbf{X}_{T'}$:

$$\operatorname{diag}\left(\sqrt{\boldsymbol{\beta}^{T'}}\right)\mathbf{X}_{T'}^{\mathsf{T}}\left(\mathbf{H}^{[n]}\right)^{-1} = (\mathbf{I} + \boldsymbol{\Delta})^{-1}\operatorname{diag}\left(\sqrt{\boldsymbol{\beta}^{T'}}\right)\mathbf{X}_{T'}^{\mathsf{T}}\mathbf{H}^{[n]\backslash T'},$$

where

$$\boldsymbol{\Delta} = \operatorname{diag}\left(\sqrt{\boldsymbol{\beta}^{T'}}\right)\mathbf{X}_{T'}^{\mathsf{T}}\left(\mathbf{H}_{\hat{\boldsymbol{\theta}}_{T'}}^{[n]}\right)^{-1}\mathbf{X}_{T'}\operatorname{diag}\left(\sqrt{\boldsymbol{\beta}^{T'}}\right) \succeq \mathbf{0}.$$

Combining all of these results, we have

$$\left\| \mathbf{H}^{T'} \left( \mathbf{H}_{\hat{\boldsymbol{\theta}}_{T'}}^{[n]} \right)^{-1} \mathbf{g}_{\hat{\boldsymbol{\theta}}_{T'}}^{T} \right\|_2 = \left\| \mathbf{X}_{T'} \operatorname{diag} \left( \boldsymbol{\beta}_{T'} \right) \mathbf{X}_{T'}^{\intercal} \left( \mathbf{H}_{\hat{\boldsymbol{\theta}}_{T'}}^{[n]} \right)^{-1} \mathbf{g}_{\hat{\boldsymbol{\theta}}_{T'}}^{T} \right\|_2 =$$

$$= \left\| \widetilde{\mathbf{X}}_{T'} \left( \mathbf{I} + \mathbf{X}_{T'}^{\intercal} \left( \mathbf{H}_{\hat{\boldsymbol{\theta}}_{T'}}^{[n]} \right)^{-1} \mathbf{X}_{T'} \right)^{-1} \widetilde{\mathbf{X}}_{T'}^{\intercal} \left( \left( \mathbf{H}_{\hat{\boldsymbol{\theta}}_{T'}}^{[n] \setminus T'} \right)^{-1} \right) \mathbf{g}_{\hat{\boldsymbol{\theta}}_{T'}}^{T} \right\|_2 \leq$$

$$\leq \left\| \widetilde{\mathbf{X}}_{T'} \right\|_{\mathrm{op}} \times \left\| \left( \mathbf{I} + \mathbf{X}_{T'}^{\intercal} \left( \mathbf{H}_{\hat{\boldsymbol{\theta}}_{T'}}^{[n]} \right)^{-1} \mathbf{X}_{T'} \right)^{-1} \right\|_{\mathrm{op}} \times \left\| \mathbf{X}_{T'}^{\intercal} \left( \left( \mathbf{H}_{\hat{\boldsymbol{\theta}}_{T'}}^{[n] \setminus T'} \right)^{-1} \right) \mathbf{g}_{\hat{\boldsymbol{\theta}}_{T'}}^{T} \right\|_2 \leq$$

$$\leq \sqrt{\left\| \mathbf{H}^{T'} \right\|_{\mathrm{op}}} \times \left\| \mathbf{I} \right\|_{\mathrm{op}} \times \left\| \mathbf{X}_{T'}^{\intercal} \left( \left( \mathbf{H}_{\hat{\boldsymbol{\theta}}_{T'}}^{[n] \setminus T'} \right)^{-1} \right) \mathbf{g}_{\hat{\boldsymbol{\theta}}_{T'}}^{T} \right\|_2 =$$

$$= \widetilde{O} \left( \frac{\sqrt{k+d} \times \sqrt{k} \times \left( \sqrt{kd} + k \right)}{n} \right) = \widetilde{O} \left( \frac{kd + k^2}{n} \right),$$

where $\widetilde{\mathbf{X}} \stackrel{\text{def}}{=} \mathbf{X} \operatorname{diag} \left( \sqrt{\boldsymbol{\beta}} \right)$. This concludes the proof of Lemma F.7, and by extension Lemma F.4 and Theorem B.4. $\quad\square$

## G. Asymptotic Analysis of Previous Results

In this section, we will prove Theorem B.1. In particular, we will show that with high probability over the training set, for all $T \in \binom{[n]}{k}$, Theorem 1.7 yields a bound that scales like

$$\text{Existing Bounds} = \widetilde{\Theta} \left( \frac{k^2 d}{n^2 \lambda^3} \right)$$

To prove this, we show that

**Lemma G.1.** *In the setting of Theorem B.1, with high probability over the training set, we have*

- $C_{\mathrm{Lip}} = \Theta(n)$
- $C_{\mathrm{op}} = \Theta \left( \frac{1}{\lambda n} \right)$
- $C_{\ell} = \Theta \left( \sqrt{d} \right)$

Since the bound in Theorem 1.7 scales like

$$\left\| \hat{\boldsymbol{\theta}}^T - \hat{\boldsymbol{\theta}}_T^{\mathrm{NS}} \right\|_2 = O(C_{\mathrm{Lip}} C_{\mathrm{op}}^3 k^2 C_{\ell}^2),$$

Theorem B.1 follows immediately from Lemma G.1.

*Proof of Lemma G.1.* By definition, $C_{\mathrm{Lip}}$ measures the Lipschitzness of the Hessian, which is in turn given by the third derivative of the loss $\mathbf{T}$. Lemma H.13 tells us that with high probability over this training set, this third moment converges to its expectation uniformly, and from our assumption that the features are normally distributed, we have

$$\mathbb{E} \left[ \mathbf{T}_{\boldsymbol{\theta}} \right] \simeq n \boldsymbol{\theta}^{\otimes 3}.$$

Therefore, with high probability $C_{\mathrm{Lip}} = \Theta(n)$ regardless of the choice of $T$.

Because our optimization domain $\Omega_{\theta} = \mathbb{R}^d$ contains limits where the Hessian of the unregularized logistic loss decays to $0$, the spectrum of the Hessian is globally lower-bounded only by the regularization term, yielding the scaling

$$C_{\mathrm{op}} = \Theta \left( \frac{1}{\lambda n} \right)$$

Finally, $C_\ell$ does not depend on the set of samples being removed and clearly concentrates around

$$C_\ell = \widetilde{\Theta}\left(\sqrt{d}\right)$$

$\square$

## H. Useful Concentration Bounds

### H.1. Expected Distance from Mean Lower Bounds Expected Norm

**Lemma H.1** (Expected Distance from Mean Lower Bounds Expected Norm). *Let $\|\cdot\|$ be any norm on $\mathbb{R}^d$ and let $\mathbf{v}$ be any random variable over the domain $\mathbb{R}^d$ with finite mean $\boldsymbol{\mu} = \mathbb{E}\left[\mathbf{v}\right]$. If $\mathbb{E}\left[\|\mathbf{v} - \boldsymbol{\mu}\|_2\right]$ is finite, then*

$$\mathbb{E}\left[\|\mathbf{v}\|\right] \geq \frac{1}{2}\mathbb{E}\left[\|\mathbf{v} - \boldsymbol{\mu}\|\right].$$

*Proof.* We prove the lemma by considering two cases. If $\|\boldsymbol{\mu}\| \geq \frac{1}{2}\mathbb{E}\left[\|\mathbf{v} - \boldsymbol{\mu}\|\right]$, then the claim follows from Jensen's inequality, since

$$\mathbb{E}\left[\|\mathbf{v}\|\right] \geq \left\|\mathbb{E}\left[\mathbf{v}\right]\right\| \geq \frac{1}{2}\mathbb{E}\left[\|\mathbf{v} - \boldsymbol{\mu}\|\right].$$

If $\|\boldsymbol{\mu}\| < \frac{1}{2}\mathbb{E}\left[\|\mathbf{v} - \boldsymbol{\mu}\|\right]$, then the claim follows from the triangle inequality

$$\mathbb{E}\left[\|\mathbf{v}\|\right] \geq \mathbb{E}\left[\|\mathbf{v} - \boldsymbol{\mu}\|\right] - \|\boldsymbol{\mu}\| \geq \frac{1}{2}\mathbb{E}\left[\|\mathbf{v} - \boldsymbol{\mu}\|\right].$$

$\square$

### H.2. Expectation of Third Order Derivative

**Lemma H.2** (Structure and Sign of the Expected Third Derivative). *Let $\mathbf{x} \sim \mathcal{N}(\mathbf{0}, \mathbf{I}_d)$ and define*

$$\gamma(z) \ \stackrel{\text{def}}{=}\ \frac{e^z\left(1 - e^z\right)}{(1 + e^z)^3}.$$

*For any fixed model $\boldsymbol{\theta} \in \mathbb{R}^d$ with $c < \|\boldsymbol{\theta}\|_2 < C$, write $t \stackrel{\text{def}}{=} \|\boldsymbol{\theta}\|_2$. Then*

$$\mathbb{E}_{\mathbf{x}}\left[\gamma(\langle\mathbf{x}, \boldsymbol{\theta}\rangle)\,\mathbf{x}^{\otimes 3}\right] \ = \ a(t)\,\boldsymbol{\theta}^{\otimes 3} \ + \ b(t)\,\mathrm{sym}(\mathbf{I} \otimes \boldsymbol{\theta}),$$

*where*

$$b(t) \ = \ \frac{1}{t}\,\mathbb{E}_{Z \sim \mathcal{N}(0,1)}\left[Z\,\gamma(tZ)\right], \qquad a(t) \ = \ \frac{1}{t^3}\left(\mathbb{E}_Z\left[Z^3\,\gamma(tZ)\right] - 3\,\mathbb{E}_Z\left[Z\,\gamma(tZ)\right]\right).$$

*Moreover, for any vector $\mathbf{v} \in \mathbb{R}^d$,*

$$\mathbb{E}_{\mathbf{x} \sim \mathcal{N}(\mathbf{0}, \mathbf{I}_d)}\left[\mathbf{T}_{\boldsymbol{\theta}}(\mathbf{v}, \mathbf{v}, \boldsymbol{\theta})\right] \ = \ -\Theta(1)\,\|\mathbf{v}\|_2^2\,\|\boldsymbol{\theta}\|_2^2.$$

*Proof of Lemma H.2.* By rotational symmetry of $\mathcal{N}(\mathbf{0}, \mathbf{I}_d)$, we may assume $\boldsymbol{\theta} = t\,\mathbf{e}_1$ (so $\mathbf{u} \stackrel{\text{def}}{=} \boldsymbol{\theta}/t = \mathbf{e}_1$) without loss of generality. Set

$$\mathbf{T} \ \stackrel{\text{def}}{=}\ \mathbb{E}_{\mathbf{x}}\left[\gamma(tx_1)\,\mathbf{x}^{\otimes 3}\right], \qquad \mathbf{x} = (x_1, \ldots, x_d).$$

**(1) Component-wise structure.** Let $T_{ijk} = \mathbb{E}\left[x_i x_j x_k\,\gamma(tx_1)\right]$. By independence of Gaussian coordinates and oddness of $\gamma$:

- If none of $i, j, k$ equals 1, then $T_{ijk} = 0$.

- If exactly one equals 1 and the other two are distinct (e.g., $i = 1$, but $1 \neq j \neq k \neq 1$), then $T_{ijk} = 0$ by $\mathbb{E}\left[x_j x_k\right] = 0$.

- If $j = k > 1$ and $i = 1$, then
$$T_{1jj} = \mathbb{E}\left[x_1 \gamma(tx_1)\right] =: b_\star(t).$$

- If $i = j = k = 1$, then
$$T_{111} = \mathbb{E}\left[x_1^3 \gamma(tx_1)\right] =: a_\star(t).$$

All other entries vanish by symmetry.

**(2) Invariant decomposition.** Rotational invariance implies $\mathbf{T}$ must lie in the span of $\boldsymbol{\theta}^{\otimes 3}$ and $\mathrm{sym}(\mathbf{I} \otimes \boldsymbol{\theta})$, hence

$$\mathbf{T} = \alpha(t)\,\mathbf{u}^{\otimes 3} + \beta(t)\,\mathrm{sym}(\mathbf{I} \otimes \mathbf{u}).$$

Matching components in the basis $\mathbf{u} = \mathbf{e}_1$ gives

$$T_{1jj} = \beta(t) \quad (j > 1), \qquad T_{111} = \alpha(t) + 3\,\beta(t).$$

Thus $\beta(t) = b_\star(t)$ and $\alpha(t) = a_\star(t) - 3b_\star(t)$. Rewriting in terms of $\boldsymbol{\theta} = t\mathbf{u}$ yields

$$\mathbf{T} = \frac{a_\star(t) - 3b_\star(t)}{t^3}\,\boldsymbol{\theta}^{\otimes 3} + \frac{b_\star(t)}{t}\,\mathrm{sym}(\mathbf{I} \otimes \boldsymbol{\theta}),$$

so $a(t)$ and $b(t)$ take the claimed form.

**(3) Signs and magnitude.** For every $t > 0$, note that $\gamma(z)$ is odd and strictly negative for $z > 0$. Hence both

$$b_\star(t) = \mathbb{E}\left[x_1 \gamma(tx_1)\right] < 0 \quad \text{and} \quad a_\star(t) = \mathbb{E}\left[x_1^3 \gamma(tx_1)\right] < 0.$$

**(4) Directional evaluation.** For arbitrary $\mathbf{v} \in \mathbb{R}^d$,

$$\mathbf{T}_{\boldsymbol{\theta}}(\mathbf{v}, \mathbf{v}, \cdot) = a(t)\,\langle\boldsymbol{\theta}, \mathbf{v}\rangle^2 \boldsymbol{\theta} + b(t)\left(\|\mathbf{v}\|_2^2\,\boldsymbol{\theta} + 2\,\langle\boldsymbol{\theta}, \mathbf{v}\rangle\mathbf{v}\right).$$

Since $a(t), b(t) = O(1)$ and $b(t) < 0$, the overall combination

$$\mathbf{T}_{\boldsymbol{\theta}}(\mathbf{v}, \mathbf{v}, \cdot) = a(t)\,\langle\boldsymbol{\theta}, \mathbf{v}\rangle^2 \boldsymbol{\theta} + b(t)\left(\|\mathbf{v}\|_2^2\,\boldsymbol{\theta} + 2\,\langle\boldsymbol{\theta}, \mathbf{v}\rangle\mathbf{v}\right)$$

has total scale $O(\|\mathbf{v}\|_2^2)$. It remains to establish that this expression has a *fixed negative sign* along $\boldsymbol{\theta}$.

To see this, we return to the behavior of the expected Hessian under a small movement in the $\boldsymbol{\theta}$ direction. Let $\mathbf{H}_{\boldsymbol{\theta}}^{\mathbf{x}}$ denote the sample Hessian at parameter $\boldsymbol{\theta}$ and $\mathbf{H}_{\boldsymbol{\theta}} = \mathbb{E}_{\mathbf{x}}\left[\mathbf{H}_{\boldsymbol{\theta}}^{\mathbf{x}}\right]$ its expectation. By definition of the third derivative tensor,

$$\mathbf{H}_{(1+\varepsilon)\boldsymbol{\theta}} \approx \mathbf{H}_{\boldsymbol{\theta}} + \varepsilon\,\mathbb{E}_{\mathbf{x}}\left[\mathbf{T}_{\boldsymbol{\theta}}(\cdot, \cdot, \boldsymbol{\theta})\right].$$

Hence, if moving along $\boldsymbol{\theta}$ makes the expected Hessian strictly smaller, then the contraction $\mathbb{E}_{\mathbf{x}}\left[\mathbf{T}_{\boldsymbol{\theta}}(\mathbf{v}, \mathbf{v}, \boldsymbol{\theta})\right]$ must be negative for all $\mathbf{v}$.

Now recall that the logistic second derivative $\beta(z) = \sigma(z)\sigma(-z)$ satisfies $\beta(z + \varepsilon) \leq e^{-\varepsilon/2}\beta(z)$ for $z > 1$. Splitting the Gaussian integral into $\{|\langle\mathbf{x}, \boldsymbol{\theta}\rangle| \leq 1\}$ and its complement gives

$$\begin{aligned}
\mathbb{E}_{\mathbf{x}}\left[\mathbf{H}_{(1+\varepsilon)\boldsymbol{\theta}}^{\mathbf{x}}\right] &= \mathbb{E}_{\mathbf{x}}\left[\beta((1+\varepsilon)\langle\mathbf{x}, \boldsymbol{\theta}\rangle)\,\mathbf{x}\mathbf{x}^\top \mathbb{1}_{|\langle\mathbf{x}, \boldsymbol{\theta}\rangle| \leq 1}\right] + \mathbb{E}_{\mathbf{x}}\left[\beta((1+\varepsilon)\langle\mathbf{x}, \boldsymbol{\theta}\rangle)\,\mathbf{x}\mathbf{x}^\top \mathbb{1}_{|\langle\mathbf{x}, \boldsymbol{\theta}\rangle| > 1}\right] \\
&\preceq \mathbb{E}_{\mathbf{x}}\left[\beta(\langle\mathbf{x}, \boldsymbol{\theta}\rangle)\,\mathbf{x}\mathbf{x}^\top \mathbb{1}_{|\langle\mathbf{x}, \boldsymbol{\theta}\rangle| \leq 1}\right] + e^{-\varepsilon/2}\mathbb{E}_{\mathbf{x}}\left[\beta(\langle\mathbf{x}, \boldsymbol{\theta}\rangle)\,\mathbf{x}\mathbf{x}^\top \mathbb{1}_{|\langle\mathbf{x}, \boldsymbol{\theta}\rangle| > 1}\right] \\
&\preceq (1 - \Omega(\varepsilon))\,\mathbb{E}_{\mathbf{x}}\left[\mathbf{H}_{\boldsymbol{\theta}}^{\mathbf{x}}\right].
\end{aligned}$$

Thus, $\mathbf{H}_{(1+\varepsilon)\boldsymbol{\theta}} \prec \mathbf{H}_{\boldsymbol{\theta}}$, confirming that the expected curvature decreases along $\boldsymbol{\theta}$. By the linearization above, this implies

$$\mathbb{E}_{\mathbf{x}} \left[ \mathbf{T}_{\boldsymbol{\theta}}(\mathbf{v}, \mathbf{v}, \boldsymbol{\theta}) \right] < 0 \qquad \text{for all } \mathbf{v},$$

and since its magnitude is $\Theta(\|\mathbf{v}\|_2^2 \, \|\boldsymbol{\theta}\|_2^2)$, we obtain

$$\mathbb{E}_{\mathbf{x}} \left[ \mathbf{T}_{\boldsymbol{\theta}}(\mathbf{v}, \mathbf{v}, \boldsymbol{\theta}) \right] = - \Theta(1) \, \|\mathbf{v}\|_2^2 \, \|\boldsymbol{\theta}\|_2^2 .$$

$\square$

## H.3. Concentration of the Sum of Sub-Gaussian Variables

**Lemma H.3** (Concentration of the Sum of Sub-Gaussian Variables). *Let $\mathbf{x}_1, \ldots, \mathbf{x}_n \sim \mathcal{X}$ be iid samples of some subgaussian distribution $\mathcal{X}$ on $\mathbb{R}^d$, with mean $\mathbf{0}$, and let $T \in \binom{[n]}{k}$ be a random set of these samples. Then with very high probability over the samples and over the choice of $T$*

$$\left\| \sum_{i \in T} \mathbf{x}_i \right\|_2 = O(\sqrt{kd}) .$$

*Moreover, with very high probability over the samples $\mathbf{x}_i$, the worst-case over choices of $T$ is also bounded*

$$\max_{T \in \binom{[n]}{k}} \left\| \sum_{i \in T} \mathbf{x}_i \right\| = O(\sqrt{kd} + k) .$$

*Proof of Lemma H.3.* For any fixed $T \subseteq [n]$ and unit vector $v \in \mathbb{R}^d$, $\sum_{i \in T} \langle v, \mathbf{x}_i \rangle$ is $O(k)$-subgaussian. So with probability at least $1 - \exp(-100d)$, we have $|\sum_{i \in T} \langle v, \mathbf{x}_i \rangle| \leq O(\sqrt{kd})$. Taking a union bound over a 0.01-net completes the proof for $\| \sum_{i \in T} \mathbf{x}_i \| = \sup_v \sum_{i \in T} \langle \mathbf{x}_i, v \rangle$. Using the same argument taking a union bound over all $T \in \binom{n}{k}$ proves the claimed bound on $\max_{T \in \binom{n}{k}} \| \sum_{i \in T} \mathbf{x}_i \|$. $\square$

## H.4. Concentration of Higher Order Moments of Subgaussian Random Variables

We prove a bound on the operator norm of the sum of fourth moments of independent subgaussian random variables.

**Lemma H.4** (Higher Order Empirical Moments of a Sub-Gaussian). *Let $\mathbf{x}_1, \ldots, \mathbf{x}_n \sim \mathcal{X}$ be iid samples drawn from a subgaussian distribution on $\mathbb{R}^d$ with mean $\mathbf{0}$ and bounded covariance $\|\mathbf{\Sigma}\|_{\mathrm{op}} = O(1)$. Then, for any fixed power $t \geq 2$, with probability $1 - e^{-\Omega(d)}$*

$$\max_{\mathbf{e} \in \mathbb{S}^{d-1}} \left\{ \sum_{i=1}^{n} |\langle \mathbf{x}_i, \mathbf{e} \rangle^t| \right\} = O\left( n + d^{t/2} \right) .$$

This result is a key technical component in our analysis of the Newton step data attribution method. Before the proof, we recall some definitions:

**Sub-Weibull random variables** A real-valued random variable $X$ is said to be *sub-Weibull* of order $\theta > 0$, written $X \sim \mathrm{subW}(\theta)$, if its Orlicz $\psi_\theta$-norm is finite:

$$\|X\|_{\psi_\theta} \stackrel{\text{def}}{=} \inf\{K > 0 : \mathbb{E}\exp\big((|X|/K)^\theta\big) \leq 2\} \; < \; \infty.$$

The smaller $\theta$, the heavier the tails; in particular, $\mathrm{subW}(2)$ corresponds to sub-Gaussian and $\mathrm{subW}(1)$ to sub-Exponential random variables. We will use the notation $X \sim \mathrm{subW}(\theta)$ and the standard inequalities $\|X\|_{\psi_p} \leq C \|X\|_{\psi_q}$ for $p > q$, and $\|X^r\|_{\psi_{p/r}} = \|X\|_{\psi_p}^r$ for any $r \geq 1$.

*Proof of Lemma H.4.* Let $S_{1/3}$ be a 1/3-net of the unit sphere $\mathbb{S}^{d-1}$ – that is, a set such that for every $\mathbf{u} \in \mathbb{S}^{d-1}$ there exists $\mathbf{v} \in S_{1/3}$ such that $\|\mathbf{u} - \mathbf{v}\| \leq 1/3$. We may assume that $|S| \leq \exp(O(d))$. First we observe that

$$\max_{\mathbf{e} \in \mathbb{S}^{d-1}} \left\{ \sum_{i=1}^{n} |\langle \mathbf{x}_i, \mathbf{e} \rangle^t| \right\} \leq O(2^t) \cdot \max_{\mathbf{e} \in S_{1/3}} \left\{ \sum_{i=1}^{n} |\langle \mathbf{x}_i, \mathbf{e} \rangle^t| \right\},$$

To see this, let $\mathbf{e} \in \mathbb{S}^{d-1}$ achieve the maximum on the left-hand side, and let $\mathbf{e}' \in S$ satisfy $\|\mathbf{e} - \mathbf{e}'\| \leq 1/3$. Then $|\langle \mathbf{x}_i, \mathbf{e} \rangle|^t = |\langle \mathbf{x}_i, \mathbf{e}' \rangle + \langle \mathbf{x}_i, \mathbf{e} - \mathbf{e}' \rangle|^t \leq 2^t (|\langle \mathbf{x}_i, \mathbf{e}' \rangle|^t + |\langle \mathbf{x}_i, \mathbf{e} - \mathbf{e}' \rangle|^t)$. Then we have

$$\sum_{i=1}^{n} |\langle \mathbf{x}_i, \mathbf{e} \rangle|^t \leq 2^t \max_{\mathbf{e} \in S_{1/3}} \left\{ \sum_{i=1}^{n} |\langle \mathbf{x}_i, \mathbf{e}' \rangle^t| \right\} + 2^t \cdot 3^{-t} \cdot \max_{\mathbf{e}'' \in \mathbb{S}^{d-1}} \left\{ \sum_{i=1}^{n} |\langle \mathbf{x}_i, \mathbf{e}'' \rangle^t| \right\},$$

which rearranges to what we wanted to show.

**Reduction to a fixed direction and sub-Weibullization.** Fix $\mathbf{e} \in S_{1/3}$ and put $Z_i \overset{\text{def}}{=} \langle \mathbf{x}_i, \mathbf{e} \rangle$. Since $\mathbf{x}_i$ are mean-zero subgaussian with $\|\mathbf{\Sigma}\| = O(1)$, the one-dimensional marginals are subgaussian with a (dimension-free) $\psi_2$-norm $\|Z_i\|_{\psi_2} = O(1)$. By the power–preserving property of sub-Weibull norms (Corollary 4 in the PDF), for any fixed integer $t \geq 2$,

$$Y_i \overset{\text{def}}{=} |Z_i|^t \sim \text{subW}\left(\frac{2}{t}\right) \quad \text{with} \quad \|Y_i\|_{\psi_{2/t}} = \|Z_i\|_{\psi_2}^t = O(1).$$

Moreover, $\mathbb{E} |Z_i|^t = O(1)$ (all constants may depend on $t$ and on the subgaussian and covariance constants but not on $n, d$).

**Concentration for a fixed net point.** Theorem 3.1 in (Hao et al., 2019) gives a Bernstein-style inequality for sums of independent sub-Weibull random variables. For any $u \geq 1$, with probability at least $1 - e^{-u}$,

$$\left| \frac{1}{n} \sum_{i=1}^{n} (Y_i - \mathbb{E}\, Y_i) \right| \leq C_1 \sqrt{\frac{u}{n}} + C_2 u^{t/2}/n,$$

where $C_1, C_2 = O(1)$ depend only on $t$ and the sub-Weibull constants. Combining this with a union bound finishes the proof. $\qquad \square$

**Theorem H.5.** *Let $x_1, \ldots, x_n$ be independent samples from a subgaussian distribution on $\mathbb{R}^d$ with mean zero and covariance $\mathbf{\Sigma}$ such that $\|\mathbf{\Sigma}\|_{\text{op}} \leq 1$. Let $n \geq d(\log d)^{O(1)}$. Then with probability at least $1 - n^{-\omega(1)}$,*

$$\left\| \sum_{i=1}^{n} (x_i \otimes x_i)(x_i \otimes x_i)^{\top} \right\|_{\text{op}} = O(nd).$$

To prove the theorem, we first seek to bound the operator norm of the expectation of a single term in the sum. Let $X$ be a random vector drawn from the same distribution as the $x_i$. We are interested in $\left\| \mathbb{E}(X \otimes X)(X \otimes X)^{\top} \right\|_{\text{op}}$. This is equivalent to finding the maximum of $\mathbb{E}\langle u, X \otimes X \rangle^2$ over all unit vectors $u \in \mathbb{R}^{d^2}$.

**Lemma H.6.** *Let $X$ be a subgaussian random vector in $\mathbb{R}^d$ with $\mathbb{E}\, X = 0$ and $\mathbb{E}\, XX^{\top} = \mathbf{\Sigma}$ with $\|\mathbf{\Sigma}\|_{\text{op}} \leq 1$. Then*

$$\left\| \mathbb{E}(X \otimes X)(X \otimes X)^{\top} \right\|_{\text{op}} = O(d).$$

*Proof.* Let $u \in \mathbb{R}^{d^2}$ be a unit vector. We can view $u$ as a $d \times d$ matrix $U$ such that $\|U\|_F^2 = \sum_{i,j} U_{ij}^2 = 1$. The expression $\langle u, X \otimes X \rangle$ is equivalent to the quadratic form $X^{\top}UX$. Let the singular value decomposition of $U$ be $U = \sum_{a=1}^{d} \sigma_a u_a v_a^{\top}$, where $\sigma_a$ are the singular values and $u_a, v_a$ are the left and right singular vectors, respectively. Since $\|U\|_F^2 = 1$, we have $\sum_{a=1}^{d} \sigma_a^2 = 1$.

The quadratic form can now be written as:

$$X^{\top}UX = \sum_{a=1}^{d} \sigma_a (X^{\top}u_a)(v_a^{\top}X).$$

We want to bound $\mathbb{E}[(X^\top U X)^2]$:

$$\mathbb{E}[(X^\top U X)^2] = \mathbb{E}\left[\left(\sum_{a=1}^{d} \sigma_a (X^\top u_a)(v_a^\top X)\right)^2\right]$$

$$= \sum_{a,b=1}^{d} \sigma_a \sigma_b \, \mathbb{E}[(X^\top u_a)(v_a^\top X)(X^\top u_b)(v_b^\top X)]$$

$$\leq \sum_{a,b=1}^{d} \sigma_a \sigma_b (\mathbb{E}(X^\top u_a)^4)^{1/4} (\mathbb{E}(X^\top v_a)^4)^{1/4} (\mathbb{E}(X^\top u_b)^4)^{1/4} (\mathbb{E}(X^\top v_b)^4)^{1/4}$$

$$\leq O(1) \cdot \sum_{a,b=1}^{d} \sigma_a \sigma_b$$

$$\leq O(d) \cdot \sum_{a=1}^{d} \sigma_a^2$$

$$= O(d) \,.$$

$\square$

Now we can prove Theorem H.5.

*Proof.* Given Lemma H.6, we can prove Theorem H.5 by applying the matrix Bernstein inequality. We bound the deviation of the sum from its expectation. Let $Z_i = (x_i \otimes x_i)(x_i \otimes x_i)^\top$. We want to bound $\left\|\sum_i (Z_i - \mathbb{E}\, Z_i)\right\|_{\mathrm{op}}$. We will use the matrix Bernstein inequality. A key challenge is that the $Z_i$ are not bounded. We address this by truncation.

Let $C = O(\sqrt{d}\log n)$. For a subgaussian vector $x_i$, we have $\Pr(\|x_i\|_2 > t\sqrt{d}) \leq \exp(-\Omega(t^2))$ (Jin et al., 2019). Let $\mathcal{E}_i$ be the event that $\|x_i\|_2 \leq C$. By a union bound, $\Pr(\forall i, \mathcal{E}_i) \geq 1 - n \cdot n^{-\omega(1)} = 1 - n^{-\omega(1)}$. Let $\tilde{x}_i = x_i \cdot \mathbb{I}(\mathcal{E}_i)$ and $\tilde{Z}_i = (\tilde{x}_i \otimes \tilde{x}_i)(\tilde{x}_i \otimes \tilde{x}_i)^\top$. With high probability, $\sum_i Z_i = \sum_i \tilde{Z}_i$. It suffices to bound $\left\|\sum_i (\tilde{Z}_i - \mathbb{E}\, \tilde{Z}_i)\right\|_{\mathrm{op}}$.

The truncated variables $\tilde{Z}_i$ are bounded: $\left\|\tilde{Z}_i\right\|_{\mathrm{op}} \leq \|\tilde{x}_i\|_2^4 \leq C^4 = O(d^2 \log^4 n)$. This is our parameter $R$ in the matrix Bernstein inequality.

Next, we need to bound the variance parameter $\sigma^2 = \left\|\sum_i \mathbb{E}(\tilde{Z}_i - \mathbb{E}\, \tilde{Z}_i)^2\right\|_{\mathrm{op}} \leq \left\|\sum_i \mathbb{E}\, \tilde{Z}_i^2\right\|_{\mathrm{op}}$.

$$\mathbb{E}\, \tilde{Z}_i^2 = \mathbb{E}[(x_i \otimes x_i)(x_i \otimes x_i)^\top (x_i \otimes x_i)(x_i \otimes x_i)^\top \cdot \mathbb{I}(\mathcal{E}_i)] \preceq \mathbb{E}[\|x_i\|_2^4 (x_i \otimes x_i)(x_i \otimes x_i)^\top] \,.$$

We need to bound the operator norm of this matrix. As in Lemma H.6, we test it with a unit vector $u \in \mathbb{R}^{d^2}$, which we view as a matrix $U$ with $\|U\| = 1$.

$$\mathbb{E}[\|x_i\|_2^4 (x_i^\top U x_i)^2] \leq \sqrt{\mathbb{E}\, \|x_i\|_2^8 \, \mathbb{E}(x_i^\top U x_i)^4} \,.$$

Since $x_i$ is subgaussian, $\mathbb{E}\, \|x_i\|_2^p = O(d^{p/2})$. So $\mathbb{E}\, \|x_i\|_2^8 = O(d^4)$. Also, $\mathbb{E}(x_i^\top U x_i)^4 = O(d^2)$ by extending the logic of Lemma H.6. This gives a variance parameter for a single term of order $O(d^3)$. Summing over $n$ terms, $\sigma^2 = O(nd^3)$.

The matrix Bernstein inequality states that for $t > 0$,

$$\Pr\left(\left\|\sum_i (\tilde{Z}_i - \mathbb{E}\, \tilde{Z}_i)\right\|_{\mathrm{op}} \geq t\right) \leq 2d^2 \exp\left(\frac{-t^2/2}{\sigma^2 + Rt/3}\right) \,.$$

Plugging in $R = O(d^2 \mathrm{polylog}(n))$ and $\sigma^2 = O(nd^3 \mathrm{polylog}(n))$, and setting $t = O(nd)$ while using $n \geq d(\log d)^{O(1)}$, we find that the deviation is bounded by $t$ with probability $1 - n^{-\omega(1)}$. The lemma follows by combining the bound on the expectation and the deviation. $\square$

## H.5. Parameter Learning for Logistic Regression and Local Strong Convexity

**Lemma H.7.** *Let $(X, y)$ be a random variable over $\mathbb{R}^d \times \{0, 1\}$ such that $X$ is a subgaussian random vector with covariance* **I**. *Let $\theta \in \mathbb{R}^d$ be an interior minimizer of the population logistic risk*

$$L_{\text{pop}}(\alpha) \overset{\text{def}}{=} \underset{(X,y)}{\mathbb{E}} [\ell(y, \langle \alpha, X \rangle)]$$

*with $c \leq \|\theta\|_2 \leq C$ for constants $0 < c < C < \infty$. Let $\hat{\theta}$ be the empirical risk minimizer given independent draws $(X_1, y_1), \ldots, (X_n, y_n)$. Suppose $n \geq d(\log d)^{O(1)}$. Then with very high probability,*

$$\left\| \hat{\theta} - \theta \right\|_2 \leq \tilde{O} \left( \sqrt{\frac{d}{n}} \right).$$

We will use this lemma to derive another useful one:

**Lemma H.8.** *Under the same assumptions as Lemma H.7, if $\hat{\theta}$ is the empirical risk minimizer, then with very high probability,*

$$\nabla^2 L_n(\hat{\theta}) \succeq \Omega(n) \cdot \mathbf{I}$$

*so long as $n \geq d(\log d)^{O(1)}$. Furthermore, the same is true if we consider the Hessian only on a subset of samples: there is a constant $c > 0$ such that with high probability all $S \subseteq [n]$ with $|S| \leq cn/\log n$,*

$$\nabla^2 L_{[n] \setminus S}(\hat{\theta}) \succeq \Omega(n) \cdot \mathbf{I}.$$

We give both proofs at the end of this section after building the requisite lemmas.

### H.5.1. UNIFORM CONVERGENCE OF THE HESSIAN

We establish uniform convergence of the Hessian. While this uses standard techniques, we include it for completeness.

To prove the lemma we start with uniform convergence of the Hessian.

**Lemma H.9.** *Under the assumptions of Lemma H.7, for any fixed constant $C = O(1)$, if $n \geq d(\log d)^{O(1)}$, then with probability at least $1 - \beta$ for all $\alpha \in \mathbb{R}^d$ with $\|\alpha\| \leq C$, the Hessian satisfies*

$$\left\| \nabla^2 L_n(\alpha) - \mathbb{E} \nabla^2 L_n(\alpha) \right\|_{op} \leq \tilde{O}(\sqrt{n(d + \log(1/\beta))} + \log(1/\beta)).$$

*Proof.* Let $x_1, \ldots, x_n \in \mathbb{R}^d$ be the covariates. We aim to bound

$$\max_{\|\alpha\| \leq C} \left\| \nabla^2 L_n(\alpha) - \mathbb{E} \nabla^2 L_n(\alpha) \right\|_{op}.$$

By a symmetrization argument, losing a constant factor it will be enough to prove a very-high-probability bound on

$$\underset{\varepsilon_1, \ldots, \varepsilon_n}{\mathbb{E}} \max_{\|\alpha\| \leq C} \left\| \sum_{i \leq n} \varepsilon_i \beta_i(\alpha) x_i x_i^\top \right\|$$

where $\beta_i(\alpha)$ is the variance of the prediction of the logistic model $\alpha$ on $x_i$ and $\varepsilon_1, \ldots, \varepsilon_n$ are independent Rademacher random variables. This in turn is precisely

$$\underset{\varepsilon_1, \ldots, \varepsilon_n}{\mathbb{E}} \sup_{\|\alpha\| \leq C, \|u\| \leq 1} \sum_{i \leq n} \varepsilon_i \beta_i(\alpha) \langle x_i, u \rangle^2$$

Let $S \subseteq \mathbb{R}^{d+d}$ be a $\delta$-net of the set $\{(a, u) : \|a\| \leq 10, \|u\| \leq 1\}$; we may assume $|S| \leq \delta^{-O(d)}$. Consider a fixed $(a, u) \in S$. Each $\varepsilon_i \beta_i(a) \langle x_i, u \rangle^2$ is a mean-zero, $O(1)$-subexponential random variable with variance $O(1)$. By composition

of subexponential random variables ((Podkopaev & Rinaldo, 2019)), $\sum_{i\leq n} \varepsilon_i \beta_i(a)\langle x_i, u\rangle^2$ is $O(1)$-subexponential with variance $O(n)$. Taking the supremum over all $|S|$ such random variables, we get that with probability at least $1 - \beta$,

$$\sup_{(a,u)\in S}\left|\sum_{i\leq n}\varepsilon_i\beta_i(a)\langle x_i, u\rangle^2\right| \leq O(\sqrt{n(d\log(1/\delta)^2 + \log(1/\beta))} + d\log(n/\delta)^2) + \log(1/\beta).$$

Now let $(a, u)$ be arbitrary, and let $(a_0, u_0) \in S$ such that $\delta_a = a_0 - a, \delta_u = u_0 - u$ with $\|\delta_a\|, \|\delta_u\| \leq \delta$. We have

$$\sum_{i\leq n}\varepsilon_i\beta_i(a_0 + \delta_a)\langle x_i, u_0 + \delta_u\rangle^2 = \sum_{i\leq n}\varepsilon_i(\beta_i(a_0) \pm O(\|x_i\|\|\delta_a\|) \cdot \langle x_i, u_0 + \delta_u\rangle^2$$

$$= \sum_{i\leq n}\varepsilon_i(\beta_i(a_0) \pm O(\|x_i\|\delta))(\langle x_i, u_0\rangle^2 \pm O(\|x_i\|^2\delta))$$

$$= \sum_{i\leq n}\varepsilon_i\beta_i(a_0)\langle x_i, u_0\rangle^2 \pm O(\|x_i\|^3\delta).$$

via Lipschitzness of the sigmoid function. Hence for any $\varepsilon_1, \ldots, \varepsilon_n$ and $x_1, \ldots, x_n$,

$$\sup_{\|a\|\leq 10, \|u\|\leq 1}\left|\sum_{i\leq n}\varepsilon_i\beta_i(a)\langle x_i, u\rangle^2\right| \leq \sup_{(a,u)\in S}\left|\sum_{i\leq n}\varepsilon_i\beta_i(a)\langle x_i, u\rangle^2\right| + O(\delta)\sum_{i\leq n}\|x_i\|^3$$

With very high probability, the latter is at most $O(\sqrt{n(d\log(n/\delta)^2 + \log(1/\beta))} + d\log(n/\delta)^2) + \log(1/\beta) + \tilde{O}(\delta d^{3/2}n)$. Picking $\delta = (nd)^{-O(1)}$ finishes the proof. $\qquad\square$

Next we need a lower bound on the population Hessian.

**Lemma H.10.** *Under the assumptions of Lemma H.7, the population Hessian satisfies*

$$\mathbb{E}\,\nabla^2 L_n(\alpha) \succeq n \cdot e^{-O(\|\alpha\|)} \cdot \mathbf{I}.$$

*Proof.* Let $x$ be a single sample from the covariate distribution. Let $\beta(\alpha)$ be the variance of the prediction of the logistic model $\alpha$ on $x$. Let $v$ be a fixed unit vector. It suffices to prove a lower bound on $\mathbb{E}\,\beta(\alpha) \cdot \langle v, x\rangle^2$. Note that $\langle v, x\rangle$ is 1-subgaussian with mean 0 and variance 1. By Paley-Zygmund, we have for any $\varepsilon \in (0, 1)$ that $\Pr(|\langle v, x\rangle| \geq \varepsilon\|v\|) \geq 1 - O(\varepsilon^2)$ So there exists a constant $C$ such that $\Pr(|\langle v, x\rangle| \geq \|v\|/C) \geq 0.99$. Now, $\langle x, \alpha\rangle$ is also mean zero and $\|\alpha\|$-subgaussian. So, with probability at least 0.99, we have $|\langle x, \alpha\rangle| \leq \|\alpha\|/C$. Putting these together, we have that

$$\mathbb{E}\,\beta(\alpha)\langle v, x\rangle^2 \geq e^{-O(\|\alpha\|)}.$$

$\qquad\square$

Putting together Lemma H.10 and Lemma H.9 immediately proves the following lemma.

**Lemma H.11.** *Under the assumptions of Lemma H.7, with very high probability, for every $\alpha \in \mathbb{R}^d$ we have $\nabla^2 L_n(\alpha) \succeq (n \cdot e^{-O(\|\alpha\|)} - \tilde{O}(\sqrt{dn})) \cdot \mathbf{I}$.*

### H.5.2. GRADIENT AT $\theta$

We also need to show that the gradient of $L_n$ has small norm at $\theta$.

**Lemma H.12.** *Under the assumptions of Lemma H.7, with very high probability,*

$$\|\nabla L_n(\theta)\| \leq \tilde{O}\left(\sqrt{dn}\right)$$

*Proof.* We start by expanding the gradient of the logistic loss explicitly. For the logistic regression loss the gradient at $\theta$ is

$$\nabla L_n(\theta) = \sum_{i=1}^{n}(\sigma(\langle\theta, x_i\rangle) - y_i)x_i,$$

where $\sigma(z) = \frac{1}{1+e^{-z}}$ is the sigmoid function.

Let $\alpha_i = \sigma(\langle \theta, x_i \rangle) - y_i$ and $\mathbf{z}_i = \alpha_i x_i$. Since $\theta$ is an interior minimizer of the population logistic risk, first-order optimality gives

$$\mathbb{E}[\mathbf{z}_i] = \mathbb{E}[(\sigma(\langle \theta, x_i \rangle) - y_i) x_i] = \mathbf{0}.$$

Moreover, $|\alpha_i| \leq 1$, so for every fixed unit vector $\mathbf{v} \in \mathbb{S}^{d-1}$,

$$|\langle \mathbf{z}_i, \mathbf{v} \rangle| \leq |\langle x_i, \mathbf{v} \rangle|.$$

Since $x_i$ is subgaussian with covariance $\mathbf{I}$, this implies that $\mathbf{z}_i$ is a mean-zero $O(1)$-subgaussian random vector. The vectors $\mathbf{z}_1, \ldots, \mathbf{z}_n$ are iid, and therefore for any fixed unit vector $\mathbf{v}$,

$$\langle \sum_{i=1}^{n} \mathbf{z}_i, \mathbf{v} \rangle$$

is $O(n)$-subgaussian.

Taking a union bound over a $1/2$-net of $\mathbb{S}^{d-1}$, we get that with very high probability

$$\left\| \sum_{i=1}^{n} \mathbf{z}_i \right\|_2 = \tilde{O}(\sqrt{nd}).$$

Since $\nabla L_n(\theta) = \sum_{i=1}^{n} \mathbf{z}_i$, this proves the claim. $\qquad\square$

### H.5.3. PROOFS OF LEMMAS H.8 AND H.7

Now we can prove our parameter learning statement.

*Proof of Lemma H.7.* Suppose the very high probability events of Lemma H.9 and Lemma H.12 hold. Then $\nabla^2 L_n(\alpha) \succeq \Omega(n) \cdot \mathbf{I}$ for all $\alpha$ such that $\|\alpha - \theta\| \leq 1$, and $\|\nabla L_n(\theta)\| \leq \tilde{O}(\sqrt{dn})$. Hence, $\|\hat{\theta} - \theta\| \leq \tilde{O}(\sqrt{dn})/n = \tilde{O}(\sqrt{d/n})$. $\qquad\square$

*Proof of Lemma H.8.* The first part follows immediately from Lemma H.9 and Lemma H.7 using that $\|\hat{\theta}\| \leq 2$.

For the second statement, it will be enough to show that with high probability all $S \subseteq [n]$ with $|S| \leq O(n/\log n)$ satisfy $\sum_{i \in S} x_i x_i^\top \leq O(|S| \log n + d)$. Putting together (Vershynin, 2012), Theorem 5.39, and a union bound over all $S$ of size $O(n/\log n)$ completes the result. $\qquad\square$

## H.6. Concentration of Third-Order Derivatives

**Lemma H.13.** *Under the assumptions of Lemma H.7, for any fixed constant $C = O(1)$, with high probability, for every $\|\alpha\| \leq C$,*

$$\sup_{\|e\|=1} |\langle \nabla^3 L_n(\alpha), e \otimes e \otimes e \rangle - \mathbb{E}\langle \nabla^3 L_n(\alpha), e \otimes e \otimes e \rangle| \leq \tilde{O}(\sqrt{nd} + d^{3/2}).$$

*Furthermore, with high probability, all $T \in \binom{n}{k}$ simultaneously satisfy*

$$\sup_{\|e\|=1} |\langle \nabla^3 L_{[n] \setminus T}(\alpha), e \otimes e \otimes e \rangle - \mathbb{E}\langle \nabla^3 L_{[n] \setminus T}(\alpha), e \otimes e \otimes e \rangle| \leq \tilde{O}(\sqrt{n(d+k)} + (d+k)^{3/2}).$$

We omit the proof of Lemma H.13 because it follows the same outline as Lemma H.9, substituting the sub-Weibull concentration bound of (Hao et al., 2019)(Theorem 3.1) for composition of subexponential random variables, as in the proof of Lemma H.4.

