# OpenReview forum: "On the Accuracy of Newton Step and Influence Function Data Attributions"
_ICML.cc/2026/Conference — ICML 2026 spotlight_

### Official Review · Reviewer_sY4H · 2026-03-10

**Soundness:** 4
**Presentation:** 3
**Significance:** 3
**Originality:** 4
**Overall Recommendation:** 5
**Confidence:** 4

**Summary:**

The authors derive theoretical bounds on data attribution methods -- mainly Newton Step (NS) and Influence Functions (IF), as well as its variant Rescaled Influence Functions (RIF) -- for convex ERM problems in the large $n$ regime. Their bound on the accuracy of NS requires fewer assumptions than previous work, namely only Lipschitz Hessians on the path from the old optimum to the NS approximation, and strong convexity in a ball around the NS approximation). This theory enables to explain prior empirical observations about the delta between IF and NS.

For logistical regression, the authors derive almost asymptotically tight bounds and show that NS is more accurate than IF in the $d>>k$ regime. Finally, they also introduce DRIF, a variant of RIF, motivated by the derived theory.

**Compliance With Llm Reviewing Policy:**

Affirmed.

**Final Justification:**

I believe the paper to be a worthy addition to the field, and find its theoretical findings very interesting. Therefore, I would like to see this paper accepted to the conference.

**Key Questions For Authors:**

1. Could you expand on where convexity of the ERM problem is needed for the presented theory, and what the challenges are in extending the results to non-convex settings?
2. Do you have some small scale experiments on the differences between RIF and DRIF, especially for comparing small to large drop-sets?
3. You state on lines 152-156 (left) that you "expect that the upper bounds of Theorem 1.1 [are] similarly close to tight for any $k,d \leq const \cdot n$". Do you have heuristic, theoretical or empirical reason for this belief?

**Limitations:**

Yes

**Strengths And Weaknesses:**

**In Short**
Strengths:
- The paper is mathematically precise, correct, and easily readable (with some minor exceptions outlined below). It's results are interesting and menaingful to the community
- The paper's theoretical contributions explain empirically observed differences between the gap of IF and NS accuracy. Furthermore, an improvement do RIF called DRIF eminates organically from the theory.
- The theoretical results are linked to potential downstream applications in machine unlearning.

Weaknesses:
- The paper only considers (weakly) convex ERM problems. While many data attribution methods build their theory on an assumption of convexity, this is not the case for most deep learning problems and there exists recent work which aims to build coherent theory for non-convex problem [1].
- The theoretically derived bounds hold only in the large-$n$ regime ($n >> d^2, k^2). Thus, such bounds hold neither for overparametrized models, nor for large-scale removal of samples. This obstructs the applicability of such bounds to overparametrized models (aside from their non-convexity) and to large-scale machine unlearning.
- The additional rescaling of RIF to derive DRIF is motivated entirely theoretically. Some (small scale) experiments to show the effect of additional rescaling would strengthen the need for additional rescaling and make readers more interested in the practical application of the presented theory.

**Points**
- I believe the presentation of Section 3 can be improved:
  - A clearer idea of the general proof strategy should be given at the start of the section, in order to help the reader follow the proof and give some better understanding as to why the previous assumptions are indeed required. To my understanding, the proof proceeds as follows: In order to bound the norm of $\mathbf{d}$, we aim to bound the inner produce between $\mathbf{d}$ and the gradient evaluated at the NS approximation from above and below. By applying the Lipschitz assumption on the segment $[\hat{\theta}, \hat{\theta}_T^{NS}]$, we bound the gradient (and thus the inner product) from above. By applying strong convexity in the ball to the intersection between $\mathcal{B}$ and the line segment $[\hat{\theta}_T, \hat{\theta}_T^{NS}]$, we bind the inner product from below. Finally, we use Assumption 1.10 to show that the derived inequality can hold only if the entire line segment $[\hat{\theta}_T, \hat{\theta}_T^{NS}]$ lies in the ball, thus deriving an inequality only involving $\| \mathbf{d} \|$, which gives us the desired result.
  - A figure similar to Figure 1 can probably help to visualise to what segement we apply what assumption.
  - 3.3 repeats a lot of previously made arguments (almost all inequalities, occasionally replacing $r$ with $\|\mathbf{d}\|$, and Figure 2 being almost equal to Figure 1). This is not terrible, but I think it makes it a bit bloated and harder for the reader to follow. Maybe 3.2 and 3.3 can be a bit reorganised to not repeat the same arguments?
- I am intrigued where exactly the convexity of the ERM problem is vital for the proof and how we could potentially extend it to non-convex problems, where $\hat{\theta}$ may be a mere saddlepoint and not a (global) optimum.To my understanding, the proof mainly fails because of the assumption of weak convexity in Lemma 3.2, but maybe I have also overlooked other uses in the argumentation. Can you elaborate where and why global weak convexity is required, and what the hurdles are to extending the results to non-convex settings?
- I found the new notation introduced in lines 268-274 (left) to be confusing: From this point on, H refers to the Hessian of the loss of the retained samples, while before (e.g. in the definition of IF in 1.1.1), H refers to the Hessian of the full loss. This has confused me at some points, e.g. in the NS cancellation step in the proof proof of Lemma 3.1.
- The proof of Lemma 3.2 relies on weak convexity of the retained loss (line 375), which to the best of my knowledge is *technically* not guaranteed or mentioned as an assumption (although normally given in most ERM settings). As the paper otherwise goes to great length to be mathematically precise, I would suggest adding this requirement.

----------------------------------
[1] Wei, D., Padhi, I., Ghosh, S., Dhurandhar, A., Ramamurthy, K. N., & Chang, M. (2024). Final-Model-Only Data Attribution with a Unifying View of Gradient-Based Methods. arXiv preprint arXiv:2412.03906.

---

> ### Author Rebuttal · Authors · 2026-03-31
>
> We thank the reviewer for the careful reading and helpful comments. We will implement the reviewer’s feedback to ensure that the final version of the paper is more clearly written.
>
> 1. **Can the main results be extended to non-convex settings?** Yes -- the proof techniques in our paper can be extended to losses that are not globally convex. In particular, as we sketch below, given Assumptions 1.8 and 1.9, and a mild tightening of Assumption 1.10, we are able to show that $B$ contains a local minimum of the loss even if the loss is not globally convex. The key limitation of such a result is that even if $B$ contains a local minimum of the loss, without convexity we are not guaranteed that gradient descent will converge to this local minimum. Given the reviewer’s interest, we will include this result and a discussion of its limitations in the final version of our paper.
>
> 2. **Empirical validation.** Since submission, we have run additional experiments on both synthetic data and an IMDB sentiment analysis benchmark comparing IF, NS, RIF, and DRIF. Anonymous figures are available here: Synthetic Gaussian: https://figshare.com/s/ed16ac414dd4eed35136?file=63311983 ; IMDB: https://figshare.com/s/c40666f8d8bfbd0bb515 . The qualitative behavior matches our theory: for small drop-sets, DRIF and RIF match the accuracy of NS, while for larger removals DRIF is substantially more accurate than IF/RIF. We will include these experiments in the final version of the paper.
>
> 3. **On the regime $k,d \ll \sqrt{n}$.** Thank you for your question. We indeed did not make sufficiently clear what we expect to happen when this assumption is violated. For the most part, we expect the bounds in Theorem 1.2 to be unchanged, with the exception of the NS approximation error for adversarial drop-sets, where we expect both our upper bounds and the error achievable by an adversary to include an additional term
>
> $$\Vert \theta^{\mathrm{NS}} - \hat{\theta}\Vert = \widetilde{\Theta}\left( \frac{kd + k^2}{n^2} + \frac{k^2 d^{3/2}}{n^3}\right)$$
>
> We explain our reasoning in our response to reviewer nRbY and do not repeat it here due to space limitations in our rebuttal.
>
> **Beyond globally convex losses:**
>
> Recall that the main result from our paper was:
>
> **Theorem 1.11:** (rephrased)
> If the loss is globally weakly convex and we are given Assumption 1.9, $H_{\theta} \succeq C_{op}^{-1} \Sigma$ (Assumption 1.8), and that $C_{op} C_h < r$ (Assumption 1.10), then *every* local minimum of the loss is within radius $C_{op} C_h$ of $\theta^{NS}$.
>
> **Theorem 1.11':** (extension to non-convex losses; not stated in the paper)
> Even if the loss is not weakly convex everywhere, but we are given the local assumptions above, and that $\mathbf{2} C_{op} C_h < r$, then *there exists* a local minimum of the loss within radius $C_{op} C_h$ of $\theta^{NS}$.
>
> The proof of this extension is similar to the proof of Theorem 1.11 from the paper, and we give a sketch here.
>
> **Proof Sketch:**
> Let $\theta^{NS}$ denote the end of the first Newton step.
> Lemma 3.1 bounds the gradient of the loss at this point and it still holds without global convexity (it depends only on the length of the Newton step and Lipschitzness of the Hessian).
>
> The key is extending the proof of Lemma 3.2 to the non-convex case.
>
> Let $I = L(B)$ be the image of our loss on the ball $B$.
> Because $L$ is continuous in $B$ and $B$ is compact, we know that its image must also be compact and therefore bounded, and from the fact that $B$ is compact by definition, $I$ must also be compact.
>
> In particular, $I$ must have a minimal value and there must be some $\theta \in B$ that attains it.
> To show that $\theta$ is a local minimum of the loss, we also need to show that $\theta$ is in the interior of $B$ (this is because if the minimum on $B$ is obtained in the interior then by definition $B$ contains a neighborhood of the minimum, meaning that it is indeed a local minimum, but we need to rule out the case that the argmin of the loss on $B$ is on the boundary, in which case it might not be a local minimum of the loss when viewed over the entire domain).
>
> Assume to the contrary that $\theta$ is on the boundary of $B$, and apply the fundamental theorem of calculus twice to show that
> $$
> L(\theta) = L(\theta^{NS}) + \langle (\theta - \theta^{NS}), g_{\theta^{NS}} \rangle + \iint \nabla^2 L \geq L(\theta^{NS}) - r \Vert g_{\theta^{NS}} \Vert_{\Sigma^{-1}} + \frac{1}{2} r^2 / C_{op}
> $$
> where the last step combined the Cauchy-Schwartz inequality on the 1st order term with our assumption of strong convexity in $B$ for the 2nd order term.
>
> From our assumption that $C_{op} C_h < r/2$, we conclude that $L(\theta) > L(\theta^{NS})$, contradicting our assumption that $\theta$ minimizes $L$ on the ball $B$.
>
> Therefore, $B$ contains a local minimum of the loss.
> From here, we conclude the $C_h C_{op}$ bound as in Lemma 3.3 (both $\theta$ and $\theta^{NS}$ lie inside of $B$ where we know that the loss is strongly convex).

---

> > ### Author Rebuttal · Reviewer_sY4H · 2026-04-04
> >
> > I thank the authors for their considerations and care in proving additional results.
> >
> > I believe the empirical validation shared in the rebuttal should be added to the paper as it helps readers see the theoretically motivated advantages of DRIF over RIF. Furthermore, I would be delighted if the authors added their proof for the setting of non-convexity into the appendix of the paper; I believe these results would be of interest to the community.
> >
> > In general, I believe the paper to be a worthy addition to the field, and want to explicitly highlight the importance of theoretical work at conferences such as ICML. Therefore, I would urge the AC to accept this paper.

---

### Official Review · Reviewer_1K4c · 2026-03-11

**Soundness:** 3
**Presentation:** 2
**Significance:** 3
**Originality:** 2
**Overall Recommendation:** 4
**Confidence:** 5

**Summary:**

This paper studies the theoretical properties of influence functions (IF) and the single Newton step (NS) method. In particular, this study observes that prior work typically relies on a strong global convexity assumption, and replaces it with a weaker local convexity assumption. Additionally, it introduces two alternative methods with convergence rates in Section 1.5.

**Compliance With Llm Reviewing Policy:**

Affirmed.

**Final Justification:**

Please see Rebuttal Acknowledgement and discussions. The reviewer had discussions with the authors on data attributions.

**Key Questions For Authors:**

please see section above.

**Limitations:**

yes

**Strengths And Weaknesses:**

- (S1) This is a very intriguing, well-written, and solid paper. Some results are somewhat straightforward, but it does not necessarily mean that they are trivial. Many results are definitely nontrivial. I personally examined the same estimator in Section 1.5 a while ago, but I could not develop any theoretical results myself. Seeing them presented here was truly enjoyable. I believe this paper is informative and helpful to broad ML/AI communities.

- (W1) My main concern with this paper is that its connection to data attribution seems less relevant. As mentioned in the introduction, data attribution aims to explain how an estimator depends on the training data. Typically, **this dependency is measured at the level of individual data points**. In other words, we aim for computing a single number for each point that represents the dependency (i.e., the papers the author cited (Ilyas et al. and Park et al.) considered individual-level values). However, the paper primarily studies how closely the NS approximation matches $\theta_T$ compared to IF for a specific subset $T$, which is not straightforward as a data attribution method. I agree with the paper’s main claim that it characterizes when NS can better approximate $\theta_T$ than IF. However, the implications for data attribution remain somewhat unclear. (Even assumptions are specific to T).
  - In fact, I think the theory in this paper could be more relevant to machine unlearning (the authors did mention it in section 1.3).
  - Please correct me if I am missing something trivial. This misalignment was one of the main reasons why I didn't select "Accept" because I expected to see theoretical results in data attribution.
- (W2) The assumptions/implications of Theorem 1.2 could be better explained. For instance, why do we need $k, d<< \sqrt{n}$?, Is NS always more accurate than IF across all different values of $d$ and $k$? When $d$ and $k$ are of similar order in terms of $n$, shouldn’t the IF and NS behave similarly? I think there are many interesting direct conclusions from the current theorems.
  - This comment applies to Theorems 1.12 and 1.13 as well.
  - Moreover, for a direct comparison, don't we need $\mathbb{E} \| \theta_{T} - \theta_{T} ^{IF} \|_2$?
  - I think the authors can add more explanations in the manuscript and the proofs (section 3) can be moved or summarized.
- (Q1) Given that Assumptions 1.4, 1.5, and 1.6 are stronger than Assumptions 1.8, 1.9, and 1.10, it seems counterintuitive that the result in Theorem 1.7 is weaker than that in Theorem 1.11. Do you have any insight into why this occurs? Could it be due to a loose bound proved in Theorem 1.7?
- (Q2) Assumption 1.10 sounds a bit strong but essential for the local-convexity. Is this assumption reasonable for various $n$, $d$, and $k$? How practical is this?
- (Q3) In page 2, the authors mentioned that ```While NS is typically much slower to compute than IF (because of the matrix inversion $H_w ^{-1}$)```. Doesn't IF also need the matrix inversion for $H ^{-1}$? Could you clarify whether their computational complexity is significantly different?

---

> ### Author Rebuttal · Authors · 2026-03-31
>
> We thank the reviewer for the careful reading and thoughtful comments.
>
> 1. **On the paper’s connection to data attribution.** We appreciate this comment, and we suspect there may be a slight mismatch in terminology. In the literature, "data attribution" is used in at least two related senses.
>
> In our paper, we study **predictive data attribution**: given a subset $T$, the goal is to accurately predict the retrained parameter $\hat{\theta}_T$ (or downstream quantities derived from it). This definition is consistent with recent literature, including the ICML 2024 tutorial *Data Attribution at Scale*.
>
> At the same time, there are also widely used **additive / credit-assignment** notions of data attribution.
>
> In credit assignment, one assigns each training sample a scalar contribution to some value of the trained model (e.g., Data Shapley).
> These notions are closely related: methods such as Data Shapley are defined by aggregating the effect of adding/removing points across many subsets, so they rely on the same subset-level quantities that we study, together with an additional aggregation step that produces per-point scores.
> Our results therefore do not directly analyze additive Shapley-based attribution scores, but they do provide guarantees for a key primitive that such methods rely on.
>
> Moreover, our analysis of non-additive methods like NS is also what enables our analysis of additive methods like IF / RIF that directly produce single-sample influence estimates.
>
> Accordingly, we view our contribution as most directly about predictive data attribution, with potential applications to machine unlearning and to credit-assignment. We agree that making this distinction more explicit would improve clarity, and we will revise the paper to better situate our contributions relative to both interpretations of "data attribution."
>
> 2. **Why compare IF to NS rather than directly to $\hat{\theta}_T$?** Similar to previous analyses, we prove our bounds on the IF error via a comparison with NS because of the similarity between the IF and NS formulas. By showing bounds on the difference between IF and NS, and on the NS error itself, we immediately get a bound on the IF error due to the triangle inequality.
> $$ \|\hat{\theta}^{\mathrm{IF}}_T-\hat{\theta}_T\| \in (\|\hat{\theta}^{\mathrm{IF}}_T-\hat{\theta}^{\mathrm{NS}}_T\| - \|\hat{\theta}^{\mathrm{NS}}_T-\hat{\theta}_T\|, \|\hat{\theta}^{\mathrm{IF}}_T-\hat{\theta}^{\mathrm{NS}}_T\| + \|\hat{\theta}^{\mathrm{NS}}_T-\hat{\theta}_T\|).$$
>
> In many regimes, we show that the NS error is significantly smaller than the difference between IF and NS, making the interval above very narrow.
> As we state in the last line of Theorem 1.2, NS is more accurate than IF as long as $d \gg k$.
>
> While we do not prove our asymptotic analysis for $k, d = \Theta(n)$, in the regime we do cover, we show that the NS error is of a similar order to IF whenever $k \approx d$.
>
> 3. **Why do we assume $k,d \ll \sqrt{n}$?** This question was also raised by Reviewer nRbY and we do not repeat the full response here due to space limitations. The main goal of our asymptotic analysis is to show that the bounds produced by our theorems are tight in some regimes, and we focus on $k,d \ll \sqrt{n}$ because extending beyond this would require very delicate concentration bounds that we felt were beyond the scope of this paper. We expect the bounds of Thms 1.2, 1.12 and 1.13 to hold for larger $k, d$, with the one exception that for adversarial drop-sets we expect the NS accuracy to scale with:
> $$
> \Vert\theta^{NS} - \hat{\theta}\Vert\approx \frac{kd+k^2}{n^2}+\frac{k^2 d^{3/2}}{n^3}
> $$
>
> 4. **Why does Theorem 1.11 end up stronger than Theorem 1.7 despite requiring weaker assumptions?** This is due to two limitations of previous analyses. First, they indeed used loose bounds, which resulted in a sub-optimal scaling wrt $k, d$. The main reason we introduced an asymptotic analysis was to verify that unlike these previous results, our analysis was not loose. However, our more fundamental contribution was to show that even just local conditions suffice to derive bounds on NS accuracy, which was previously not known.
>
> 5. **How reasonable is Assumption 1.10?** We proved our asymptotic upper bounds on the NS error by showing that Assumption 1.10 holds in the cases we analyze. We carry out this asymptotic analysis only in the regime $k,d \ll \sqrt{n}$ because of the technical difficulties discussed above. We expect this assumption to hold up to $k, d = \Theta(n)$ for random drop-sets and $k, d \leq n^{6/7}$ for adversarial drop-sets (at which point the $k^2 d^{3/2} / n^3$ terms in the NS error scaling would cause our analysis to break down).
>
> 6. **Why is NS slower than IF if both involve Hessian inversion?** NS is slower than IF in query complexity, since IF can be computed with a single Hessian inversion, whereas NS requires inverting the Hessian separately for each drop-set $T$.

---

> > ### Author Rebuttal · Reviewer_1K4c · 2026-04-03
> >
> > Thank you for the responses. I believe there may be some confusion regarding the definition of "predictive data attribution." To clarify, let me first quote the authors’ statement: **given a subset $T$, the goal is to accurately predict the retrained parameter $\hat{\theta}_T$ (or downstream quantities derived from it).**
> >
> > The ICML 2024 tutorial "Data Attribution at Scale," mentioned by the authors, actually refers to a paper (https://arxiv.org/abs/2202.00622) for the formal definition of "predictive data attribution" (In fact, this paper is written by the tutorial organizers). In that paper, the final output (Equation (9)) is a \(d\)-dimensional vector, where \(d\) corresponds to the number of training samples, *assigning an attribution score to each individual training point.* Another paper (https://arxiv.org/abs/2303.14186), authored by many of the same researchers, defines the data attribution (Definition 2.1) at the level of individual training data points. The objective in these works is to assign a contribution score to each data point, rather than to predict the retrained parameter \(\hat{\theta}_T\).
> >
> > In addition, the term "predictive" used in the data attribution literature refers to a prediction with respect to a specific test input \(x\), not a loss value. I believe the authors' statement **given a subset $T$, the goal is to accurately predict the retrained parameter $\hat{\theta}_T$ (or downstream quantities derived from it).** is not necessarily correct, if it were to define "predictive data attribution."
> >
> > I believe the other points make sense to me. While I am confident that the authors are confused regarding the definition of "data attribution" and, indeed, when considering the potential impact of this good quality paper, I would otherwise be inclined to oppose its acceptance, I have nonetheless concluded that this paper offers more values to the ML community; therefore, I will maintain my current score.

---

> > > ### Author Response · Authors · 2026-04-04
> > >
> > > Thank you for the clarification. We agree that the literature uses *data attribution* in a few closely related ways, and we would be happy to make this more explicit in revision. Our only point here is that the subset-level formulation we use is not unusual, but rather a standard predictive data attribution / datamodeling formulation.
> > >
> > > For example, the ICML tutorial [*Data Attribution at Scale*](https://ml-data-tutorial.org/chapter-1) states:
> > >
> > > > The goal of a predictive data attribution method is to output a function $\hat f$ (called a datamodel) such that, for any possible training dataset $S \subset \mathcal U$,
> > > > $$
> > > > \hat f(S) \approx \ell(\theta(S)).
> > > > $$
> > >
> > > Likewise, Definition 1 of the first paper you cite, [*Datamodels: Predicting Predictions from Training Data*](https://arxiv.org/abs/2202.00622), is explicitly subset-based:
> > >
> > > > For any set $S' \subset S$, let $f_{\mathcal A}(x;S')$ be the (stochastic) output of training a model on $S'$ using $\mathcal A$, and evaluating on $x$. A datamodel for $x$ is a parametric function $g_\theta$ optimized to predict $f_{\mathcal A}(x;S_i)$ from training subsets $S_i \sim \mathcal D_S$.
> > >
> > > The linear form in Eq. (9) of that paper appears later, in Section 2 (“Constructing (linear) datamodels”), as a simplification of this more general subset-based setup.
> > >
> > > We agree that in the second paper you cite, [*TRAK: Attributing Model Behavior at Scale*](https://arxiv.org/abs/2303.14186), the formal definition focuses on single-sample removals, and we are happy to note this ambiguity in terminology in revision. At the same time, that paper also motivates the problem in broader counterfactual terms:
> > >
> > > > To this end, we adopt the view that a data attribution method is useful insofar as it can make accurate *counterfactual predictions*, i.e., answer questions of the form “what would happen if I trained the model on a given subset $S^\prime$ of my training set?”
> > >
> > > More broadly, we think the literature contains several closely related formalizations of data attribution, all centered on predicting counterfactual model behavior under changes to the training set. We use the subset-level formulation because it naturally subsumes the single-point setting: per-instance removals are recovered by taking the removed set to be a singleton, and output-level quantities can be viewed as downstream functions of the retrained weights. We would be happy to clarify this terminology more explicitly in revision.

---

### Official Review · Reviewer_nRbY · 2026-03-11

**Soundness:** 3
**Presentation:** 4
**Significance:** 3
**Originality:** 4
**Overall Recommendation:** 4
**Confidence:** 3

**Summary:**

The paper analyzes existing knowns about the Newton Step(NS) and the influence function(IF) for Data attribution, and addresses a missing piece regarding "sufficient conditions" to explain the performance gap between NS and IF for remove-k-out effect for logistic regression with Gaussian covariates, and provides asymptotic bounds depending on training samples size $n$, size of removal $k$ as well as dimension $d$.

**Compliance With Llm Reviewing Policy:**

Affirmed.

**Key Questions For Authors:**

1. What would fundamentally break under more realistic regime $n\lesssim d$?
2. What would be a meaningful comparison between $\hat{\theta}^{\text{IF}}\_{T}$ and $\hat{\theta}\_{T}$?
3. How significant are bounds in Theorem 1.2? For example, for k-fold cross validation, the gap follows roughly $d/n$. It seems trivial for large models $n\lesssim d$?

**Limitations:**

- Theorem statements can be improved (see weaknesses).
- Direct comparison between IF estimator and $\hat{\theta}_{T}$ is missing.
- Discussion on realistic large model case is missing.

**Strengths And Weaknesses:**

Strengths:
- The paper highlights a notable difference between NS and IF, and quantitively addresses the missing piece in theoretically understanding their gaps.
- The paper is well-written and easy to follow. The explanatory figure for the main theorem is demonstrative.
- The literature review is complete to provide a comprehensive picture.

Weaknesses:
- Theorem statements can be improved. For example, probabilistic statement can be more complete by explicitly stating probability depending on $n,k,d$ for Theorem 1.2. Theorem 1.11 can also have a simple remark to note its independence of logistic regression with Gaussian covariates setup (it is mixing with other theorems that depend the specific setup).
- The paper states to address why NS is more accurate than IF in Theorem 1.2, but the IF estimator is compared with NS estimator rather than $\hat{\theta}_{T}$.
- The results highly depend on the sufficient data regime where $n\ll d^2,k^2$. In realty, the parameter size could be way greater than the sample size, and this is not well-aligned with posed assumptions.

---

> ### Author Rebuttal · Authors · 2026-03-31
>
> We thank the reviewer for the careful reading and helpful comments.
>
> 1.  **On the regime $k,d \ll \sqrt{n}$.** We focus on the small $k, d$ regime, not because we expect things to fundamentally break down for larger $k, d$, but because of the difficulty in proving concrete concentration bounds in these regimes. Our main motivation for the asymptotic analysis was to check that Theorem 1.1 was tight, so we felt that extending our bounds to the high $k, d$ regime was beyond the scope of the current paper.
>
> The only portion of our analysis that we expect to fundamentally change when $k, d$ are allowed to be larger is that, when $d$ is greater than $n^{2/3}$, the asymptotic formula for the NS error on adversarial drop-sets would change. This is because the NS error is driven by the third-order derivative $\mathbf{T}_{\theta} = \sum \mathbf{x}_i^{\otimes 3} \gamma(\theta,\mathbf{x}_i)$, and because this is a third-order tensor, we expect it to have $d^{3/2}$ "spikes" -- i.e., if we take the inner product of this tensor with the unit tensor $e^{\otimes 3}$, for $e = x_i / \Vert x_i \Vert$, then we see that it has a singular value of order $d^{3/2}$. We do not expect random drop-sets to have a high inner product with any of these spikes, but for adversarial drop-sets in this regime, we do expect an error of:
>
> $$\Vert \theta^{\mathrm{NS}} - \hat{\theta}\Vert = \widetilde{\Theta}\left( \frac{kd + k^2}{n^2} + \frac{k^2 d^{3/2}}{n^3}\right)$$
>
> We expect this scaling to be tight (i.e., that there exists an adversary attaining this error), since the adversary that we analyze in the submission (which currently drops samples aligned with an arbitrary direction) can be easily adapted to drop samples aligned with a fixed retained sample.
>
> Note that the scaling above would still preserve the high-level claim that the NS error is smaller than that of IF for "small" adversarial drop-sets (though the condition for being "small" would now be $k \ll \min(d, n^{2/3})$ instead of $k \ll d$).
> Again, these scalings for large $k, d$ are conjectural and this regime is beyond the scope of our submission, since proving these results would require substantially more complicated concentration bounds, which were not the main focus of our result.
>
> 2. **Why compare IF to NS rather than directly to $\hat{\theta}_T$?** Similar to previous analyses, we prove our bounds on the IF error via a comparison with NS because of the similarity between the IF and NS formulas. By showing bounds on the difference between IF and NS, and on the NS error itself, we immediately get a bound on the IF error due to the triangle inequality.
> $$ \|\hat{\theta}^{\mathrm{IF}}_T-\hat{\theta}_T\| \in  (\|\hat{\theta}^{\mathrm{IF}}_T-\hat{\theta}^{\mathrm{NS}}_T\| - \|\hat{\theta}^{\mathrm{NS}}_T-\hat{\theta}_T\|, \|\hat{\theta}^{\mathrm{IF}}_T-\hat{\theta}^{\mathrm{NS}}_T\| + \|\hat{\theta}^{\mathrm{NS}}_T-\hat{\theta}_T\|).$$
>
> Our key observation is that $\|\hat{\theta}^{\mathrm{NS}}_T-\hat{\theta}_T\| \lesssim \|\hat{\theta}^{\mathrm{IF}}_T-\hat{\theta}^{\mathrm{NS}}_T\|$, and in several regimes even a $\ll$ relationship holds. Using the triangle inequality above, this implies a corresponding separation between IF and the retrained parameter $\hat{\theta}_T$. We agree that this implication should be stated much more explicitly in the paper, and we will amend this in the final version of the paper.
>
>
> 3. **Interpreting Theorem 1.2.** To understand the bounds in Theorem 1.2, we consider the inherent scales involved. We derived our bounds for a PAC setting with a model of norm of order $1$ (in some sense, this is the baseline prior without seeing data), and from classical statistics, we know that training on $n$ samples should put our model in a ball of radius $\sqrt{d/n}$ around the ground truth model (the space of all "reasonable" models).
>
> For $k$-fold cross-validation, we are considering a removal of a random subset of the samples, and from the reviewer’s phrasing, it seems like they are asking about the case of $O(1)$ folds. In this case, $k$ (as defined in our submission to be the number of samples removed) would be of order $\Theta(n)$, which puts it beyond the guarantees of Theorem 1.2. But if we are willing to assume that the asymptotics of Theorem 1.2 hold for large $k$ as we conjecture, then the resulting bounds would tell us that the IF approximation error is of order $\sqrt{d/n}$ (i.e., we could be anywhere in the ball of all reasonable models), and the NS approximation error is of order $d/n$ (i.e., in this case NS would be much more accurate).

---

### Official Review · Reviewer_oyZE · 2026-03-16

**Soundness:** 3
**Presentation:** 2
**Significance:** 2
**Originality:** 3
**Overall Recommendation:** 4
**Confidence:** 4

**Summary:**

Prior guarantees for IF/NS depended on global strong convexity of the leave-out objective, which is often unrealistic for logistic regression on $\mathbb{R}^d$. This paper instead proves that local strong convexity in a neighborhood of the first Newton step, with a directional Hessian-Lipschitz condition, is enough to bound NS error. It then instantiates the framework in Gaussian logistic regression and derives nearly tight scaling laws for average-case and worst-case drop sets. The authors also analyze RIF/DRIF and argue DRIF can approach NS accuracy while preserving additivity.

**Compliance With Llm Reviewing Policy:**

Affirmed.

**Key Questions For Authors:**

See weakness.

**Limitations:**

The main limitation should be the limited practical impact beyond theoretical interests.

**Strengths And Weaknesses:**

# Strengths.
1. Tackles an important problem in influence function with a sound theoretical contribution.
2. Provides nearly tight bounds for logistic regression and explains why Newton-based approximations outperform IF in high dimensions.
3. Extends the analysis to RIF/DRIF, which improves practical relevance.

# Weaknesses.
1. Purely theoretical; no experiments to validate the claims. For pure theoretical interest, this might be interesting.
2. The local assumptions may still be hard to verify in practice.
3. Limited impact: NS is rarely used in practice, and the theoretical results provide limited insight that guides practitioners, and its broader practical impact is suggested more than demonstrated.

---

> ### Author Rebuttal · Authors · 2026-03-31
>
> We thank the reviewer for the careful reading and helpful comments.
>
> 1. **Empirical validation.** Since submission, we have run additional experiments on both synthetic data and an IMDB sentiment analysis benchmark comparing IF, NS, RIF, and DRIF. Anonymous figures are available here: Synthetic Gaussian: https://figshare.com/s/ed16ac414dd4eed35136?file=63311983 ; IMDB: https://figshare.com/s/c40666f8d8bfbd0bb515 . The qualitative behavior matches our theory: for small drop-sets, DRIF and RIF match the accuracy of NS, while for larger removals DRIF is substantially more accurate than IF/RIF. We will include these experiments in the final version of the paper.
>
> 3. **Practical relevance of NS.** We agree that exact NS is often too computationally expensive for deep networks and are not advocating for its adoption in that setting. However, analyzing NS is still useful because: (i) many practical data-attribution methods are motivated by NS -- for instance TRAK is derived via a series of approximation to NS, (ii) it is a useful proxy through which to study influence functions (both in our submission and in previous analyses of IF), and (iii) it motivates our definition of **DRIF**, which preserves the efficiency and additivity of IF while approaching NS accuracy.

---

> > ### Author Rebuttal · Reviewer_oyZE · 2026-04-01
> >
> > Thanks for the response. For 1, I'm happy to see we have some experimental results. I can see how this can fit into the paper. My (nitpicking) concern is that this, as an illustration of the theory, still provides limited suggestions to practitioners/audiences of ICML.
> >
> > For 2, I think a way to improve the paper is to carefully discuss these nuances and explain the rationale of why one should care about approximating NS and studying NS's accuracy in a standalone paragraph.
> >
> > Overall, I'll leave the decision to AC. I'm leaning towards acceptance since I believe that for audiences who enjoy theoretical results like myself, this paper is interesting. But it might just not be suited best for ICML's theme.

---

### Decision · Program_Chairs · 2026-04-30

**Decision:**

Accept (spotlight)

**Comment:**

Overall, all reviewers agree with the strong theoretical contribution made in this paper to extend the analysis of the influence function beyond the strongly convex case. This is highly non-trivial contribution and clearly above the threshold of ICML acceptance. Although the practical implication may be limited, this weakness is not significant compared with the strength of the contributions.

As requested by the reviewers, it is a good idea to expand the discussions on the assumptions and add the experiments to the final version.